# Aurora B controls anaphase onset and error-free chromosome segregation in trypanosomes

Daniel Ballmer[1,2] , Hua Jane Lou[3] , Midori Ishii[1,2] , Benjamin E. Turk[3] , and Bungo Akiyoshi[1,2] 

**Kinetochores form the interface between chromosomes and spindle microtubules and are thus under tight control by a complex regulatory circuitry. The Aurora B kinase plays a central role within this circuitry by destabilizing improper kinetochore–microtubule attachments and relaying the attachment status to the spindle assembly checkpoint. Intriguingly, Aurora B is conserved even in kinetoplastids, a group of early-branching eukaryotes which possess a unique set of kinetochore proteins. It remains unclear how their kinetochores are regulated to ensure faithful chromosome segregation. Here, we show in *Trypanosoma brucei* that Aurora B activity controls the metaphase-to-anaphase transition through phosphorylation of the divergent Bub1-like protein KKT14. Depletion of KKT14 overrides the metaphase arrest resulting from Aurora B inhibition, while expression of non-phosphorylatable KKT14 delays anaphase onset. Finally, we demonstrate that re-targeting Aurora B to the outer kinetochore suffices to promote mitotic exit but causes extensive chromosome missegregation in anaphase. Our results indicate that Aurora B and KKT14 are involved in an unconventional circuitry controlling cell cycle progression in trypanosomes.**

## Introduction

During cell division, the duplicated genetic material must be faithfully distributed from mother to daughter cells. To ensure this, sister chromatids that are held together by cohesin complexes need to form stable end-on attachments with microtubules emanating from opposite spindle poles, a process referred to as biorientation (Musacchio and Desai, 2017). Kinetochores, which assemble onto centromeric chromatin, act as the interface between chromosomes and the spindle apparatus. In most studied eukaryotes, kinetochore assembly is scaffolded by a centromere-specific histone H3 variant, CENP-A (Westhorpe and Straight, 2013; Maddox et al., 2012; Hori and Fukagawa, 2012; Allshire and Karpen, 2008; Black and Cleveland, 2011). A collection of "inner kinetochore" protein complexes called the constitutive centromere-associated network (CCAN) interacts with centromeric CENP-A chromatin and provides a platform for the "outer kinetochore" KNL1/Mis12 complex/Ndc80 complex (KMN) network (Cheeseman et al., 2006; Okada et al., 2006; Izuta et al., 2006; Foltz et al., 2006), which captures spindle microtubules during mitosis.

The kinetochore–microtubule (KT–MT) interface is under tight regulatory control by a complex circuitry of kinases and phosphatases. A key player is the chromosomal passenger complex (CPC), comprising the Aurora B kinase (the catalytic subunit), INCENP, Survivin, and Borealin in humans (Honda et al., 2003; Gassmann et al., 2004; Sampath et al., 2004; Kim et al., 1999; Nakajima et al., 2009). The CPC concentrates at centromeres during early mitosis, where it releases improper KT-MT attachments that lack tension by phosphorylating outer kinetochore proteins, a process termed "error correction" (Musacchio and Desai, 2017). Unattached kinetochores activate the spindle assembly checkpoint (SAC), a feedback control system that delays the onset of anaphase (Foley and Kapoor, 2012; Musacchio, 2015; Sacristan and Kops, 2015). SAC components include the kinases Mps1 and Bub1, as well as BubR1 (Mad3), Bub3, Mad1, Mad2, and Cdc20, which are widely conserved among eukaryotes (Kops et al., 2020). It is thought that unattached kinetochores catalyze the production of a diffusible "wait anaphase" signal, the mitotic checkpoint complex (MCC; composed of Mad2, Cdc20, BubR1, and Bub3), which inhibits the anaphase-promoting complex/cyclosome (APC/C) (Herzog et al., 2009; Izawa and Pines, 2015; Alfieri et al., 2016; Yamaguchi et al., 2016; Sudakin et al., 2001; Chao et al., 2012). The APC/C is a multisubunit E3 ubiquitin ligase that promotes anaphase onset and sister chromatid separation by marking securin and cyclin B for proteasomal degradation (Pines, 2011; Alfieri et al., 2017). Thus, the timing of anaphase onset in metazoa and yeast is

[1]Department of Biochemistry, University of Oxford, Oxford, UK;   [2]The Wellcome Centre for Cell Biology, Institute of Cell Biology, School of Biological Sciences, University of Edinburgh, Edinburgh, UK;   [3]Department of Pharmacology, Yale School of Medicine, New Haven, CT, USA.

Correspondence to Bungo Akiyoshi: bungo.akiyoshi@ed.ac.uk.



controlled by the rate of MCC production, which depends on the phosphorylation status of the KMN network at each kinetochore, governed by the local activity of Aurora B, the checkpoint kinase Mps1, and antagonizing phosphatases. Upon anaphase onset, the CPC translocates to the central spindle and is degraded as cells enter G1 (Cooke et al., 1987).

Similar to SAC components, key players of the CCAN and KMN network are widely conserved among eukaryotes (Drinnenberg and Akiyoshi, 2017; Meraldi et al., 2006; Tromer et al., 2019; van Hooff et al., 2017). However, none are found in the kinetoplastid phylum, a group of evolutionarily divergent flagellated protists, which include parasitic Trypanosomatida (e.g., *Trypanosoma brucei*, *Trypanosoma cruzi*, and *Leishmania* spp.). Instead, a unique set of proteins called kinetoplastid kinetochore proteins (KKTs) and KKT-interacting proteins (KKIPs) are present in *T. brucei* (Akiyoshi and Gull, 2014; Nerusheva and Akiyoshi, 2016; Nerusheva et al., 2019; D'Archivio and Wickstead, 2017). Based on the finding that some KKT proteins have similarities to components of synaptonemal complexes (zipper-like structures that assemble between homologous chromosomes and promote genetic exchange during meiosis) or homologous recombination machinery, we have hypothesized that a kinetoplastid ancestor repurposed parts of its meiotic machinery to assemble unique kinetochores (Tromer et al., 2021). Indeed, like synaptonemal complexes, sister kinetochores are closely paired in trypanosomes (Ogbadoyi et al., 2000), which stands in sharp contrast with canonical kinetochores that have a significant space (~1 µm) in between sister kinetochores (called inner centromeres) (Bloom, 2014). Due to the proximity between sister kinetochores, most YFP-tagged KKT proteins appear as single dots (rather than pairs of dots) under conventional microscopes (Akiyoshi and Gull, 2014), while N-terminally YFP-tagged KKT24 and many KKIP proteins appear as pairs of dots that are separated by ~340 nm in metaphase (Brusini et al., 2021; Nerusheva et al., 2019). Our recent study using 3D-SIM super-resolution microscopy has revealed pairs of dots for KKT4, KKT14, and KKT15 that are separated by ~140 nm, while other tested KKT proteins and Aurora B[AUK1] still appear as diffraction-limited dots (Hayashi and Akiyoshi, unpublished data). In traditional model eukaryotes, the term "outer kinetochore" refers to the KMN network that has microtubule-binding activity (Musacchio and Desai, 2017). In trypanosomes, this term was previously used to refer to the KKIP1 protein based on weak similarity to outer kinetochore proteins Ndc80/Nuf2 in coiled-coil regions (D'Archivio and Wickstead, 2017; Brusini et al., 2021). However, AlphaFold-based structural predictions do not support the possibility that KKIP1 is a divergent Ndc80/Nuf2, and currently, there is no evidence that KKIP1 has microtubule-binding activity. Instead, KKT4 remains the only kinetochore protein that has been shown to bind microtubules (Llauró et al., 2018). We therefore suggest that the term "outer kinetochore" is used for the microtubule-binding KKT4 protein and KKT14/15, "kinetochore periphery" for those (e.g., KKIP1, KKIP2, KKIP3, KKT24) whose N- or C-terminal ends are located farther away from KKT4, and "inner kinetochore" for those proteins that appear as single dots at the resolution of 3D-SIM (Brusini et al., 2021;

D'Archivio and Wickstead, 2017; Nerusheva et al., 2019) (Hayashi and Akiyoshi, unpublished data). The "inner kinetochore" includes the protein kinases KKT2 and KKT3, which possess kinase domains classified as unique among known eukaryotic kinase subfamilies (Parsons et al., 2005; Akiyoshi and Gull, 2014). KKT2 and KKT3 localize to centromeres throughout the cell cycle using unique zinc-binding central domains and have divergent polo boxes required for the localization of other kinetochore proteins (Marcianò et al., 2021; Ishii et al., 2022). Hence, they are thought to form the base of the kinetochore assembly hierarchy in *T. brucei*.

Intriguingly, trypanosomes are unable to halt their cell cycle in response to spindle defects, and it is thought that they do not possess a canonical SAC system (Robinson et al., 1995; Ploubidou et al., 1999; Hayashi and Akiyoshi, 2018). However, despite the large number of chromosomes in *T. brucei* (11 homologous pairs of megabase chromosomes with regional centromeres and ~100 minichromosomes without centromeres), their mis-segregation rate is very low (~1%, e.g., comparable with human cells [Ishii and Akiyoshi, 2020; Santaguida and Amon, 2015; Wickstead et al., 2003]). It remains unknown how kinetoplastids ensure error-free chromosome segregation. Interestingly, kinetoplastids have a conserved Aurora B kinase (Tu et al., 2006; Li et al., 2008a, 2008b). We recently demonstrated that the CPC in *T. brucei* is a pentameric complex comprising the Aurora B[AUK1] kinase, INCENP[CPC1], CPC2, and two orphan kinesins KIN-A and KIN-B (Ballmer and Akiyoshi, 2024). Whether the CPC is involved in error correction and/or some form of mitotic checkpoint signaling in trypanosomes is not known. Previous studies showed that knockdown of any of the five CPC subunits prevents cells from completing nuclear division (Tu et al., 2006; Li et al., 2008a, 2008b; Ballmer and Akiyoshi, 2024), suggesting that the Aurora B[AUK1] kinase functions as a key regulator of mitosis in *T. brucei*. Yet, in the absence of canonical substrates, the molecular principles and mode of action of the trypanosome CPC remain elusive.

Here, using an analog-sensitive approach, we show that Aurora B[AUK1] activity controls the metaphase-to-anaphase transition and promotes chromosome biorientation in the procyclic form *T. brucei*. Aurora B[AUK1] phosphorylates several kinetochore components, including the microtubule-binding protein KKT4 and the Bub1/BubR1-like protein KKT14. Several sites matching the Aurora B[AUK1] consensus motif within the N-terminal region (NTR) but not the C-terminal pseudokinase domain of KKT14 are phosphorylated by Aurora B[AUK1]. Depletion of KKT14 results in a partial rescue of the cell cycle arrest caused by Aurora B[AUK1] inhibition, while expression of phosphodeficient KKT14 mutants results in a prominent delay in the metaphase-to-anaphase transition, suggesting that KKT14 antagonizes APC/C activation. Finally, ectopic tethering of the catalytic module of the CPC to the outer kinetochore using a GFP nanobody-based system is sufficient to promote mitotic exit but causes massive lagging chromosomes in anaphase. We propose that the CPC and KKT14 are involved in a regulatory circuit controlling error-free chromosome segregation and cell cycle progression in trypanosomes.

## Results

### Aurora B[AUK1] controls the metaphase-anaphase transition in trypanosomes

As reported previously (Jones et al., 2014; Tu et al., 2006; Ballmer and Akiyoshi, 2024), siRNA-mediated depletion of Aurora B[AUK1] caused a pronounced cell cycle defect, with cells arresting in G2/M phase after 16 h (Fig. S1, A and B). These cells exhibited elongated and aberrantly shaped nuclei (Fig. S1 C) that were positive for cyclin B[CYC6] (data not shown), indicating that they were unable to progress into anaphase. The distance between the segregated kinetoplasts ("K") can be used to estimate the progression of cytoplasmic/flagellar division cycle, which is uncoupled from nuclear division ("N") in *T. brucei* (Hayashi and Akiyoshi, 2018; Ploubidou et al., 1999; Robinson et al., 1995) (Fig. 1 B). The average interkinetoplast distance in 2K1N cells significantly increased upon the knockdown of Aurora B[AUK1] (Fig. S1 C), consistent with a delay in the metaphase–anaphase transition in the nucleus.

To test whether the kinase activity of Aurora B[AUK1] regulates entry into anaphase and to implement a more rapid loss-of-function system, we generated cell lines harboring analog-sensitive Aurora B[AUK1] alleles (Aurora B[AUK1-as1]) (Bishop et al., 2000). Treatment of Aurora B[AUK1-as1] cells with 2 µM of PP1 analogs (1NM-PP1, 1NA-PP1, or 1MB-PP1) resulted in a prominent growth defect (Fig. 1 A and Fig. S1 D). Remarkably, after just 4 h of treatment with 1NM-PP1 (corresponding to half a cell cycle), 40% of cells were in a 2K1N state (Fig. 1, C and D), which is comparable with the cell cycle arrest observed upon treatment with the proteasome inhibitor MG132 (Fig. S1, E and F) or expression of non-degradable cyclin B[CYC6] (Hayashi and Akiyoshi, 2018). Morphologically, these cells possessed elongated nuclei with a mitotic spindle (marked by tdTomato-MAP103) and were positive for cyclin B[CYC6], indicative of a metaphase arrest (Fig. 1, D–H). After 16 h (approximately two cell cycles) of Aurora B[AUK1] inhibition, most cells had reached a 4K1N state (Fig. S1, G and H), consistent with two rounds of kinetoplast replication having occurred in the absence of nuclear division. Thus, inhibition of Aurora B[AUK1] kinase activity using our analog-sensitive system efficiently halts anaphase entry within the first cell cycle. Our results are consistent with a previous study showing cell cycle arrest of *T. brucei* upon treatment with high doses of a small-molecule Aurora kinase inhibitor (Li et al., 2009).

Even though spindle assembly was observed upon 4-h inhibition of Aurora B[AUK1] activity, we noticed that the fraction of metaphase cells with an intact spindle progressively declined upon prolonged 1NM-PP1 treatment (Fig. S1 I). To test whether Aurora B[AUK1] is required for spindle stability, we arrested cells in metaphase by MG132 treatment followed by a brief pulse of ansamitocin to depolymerize the mitotic spindle and then monitored spindle reformation in the presence of 1NM-PP1 or MG132 as a control (Fig. 1, H and I). We found that spindle reformation was inefficient in 1NM-PP1 treated cells, suggesting that Aurora B[AUK1] activity is important for spindle assembly and/or stability. This could explain why the mitotic spindle was previously reported to be lost in trypanosomes depleted for CPC components for 2 days (Li et al., 2008a, 2008b). Together, these data show that Aurora B[AUK1] activity controls the metaphase-to-anaphase transition in trypanosomes and is required for preserving the integrity of the mitotic spindle.

### Aurora B[AUK1] activity is required for the establishment of stable KT-MT attachments

We next aimed to examine the biorientation status after inhibiting Aurora B[AUK1] activity. Kinetochore periphery proteins (e.g., KKIP2, KKIP3) appear as two dots in metaphase cells even under a conventional microscope (Brusini et al., 2021). We therefore reasoned that they may serve as a biorientation marker in trypanosomes. Indeed, the number of metaphase kinetochores double positive for KKIP2 was significantly reduced when spindle microtubules were disrupted or Aurora B[AUK1] activity was inhibited for 4 h (Fig. 2, A and B). Similar defects were observed even after 1-h inhibition of Aurora B[AUK1]. Furthermore, the distance between KKIP2 foci labeling bioriented kinetochores was markedly decreased in 1NM-PP1 treated cells (Fig. 2 C), indicative of a reduced level of tension across the inter-sister kinetochore axis. These results show that Aurora B[AUK1] activity is important for chromosome biorientation in trypanosomes.

YFP-tagged Aurora B[AUK1-as1] localized at kinetochores in the presence of 1NM-PP1, suggesting that its kinase activity is not essential for targeting the CPC at the centromeric region (Fig. 1 D). Moreover, Aurora B[AUK1] inhibition had only a moderate impact on the recruitment of most KKT proteins (Fig. S2 A). In general, inner kinetochore proteins showed a modest increase in signal intensity at metaphase kinetochores, whereas outer kinetochore and kinetochore periphery proteins were largely unaffected in 1NM-PP1-treated cells, except for KKIP3 whose levels were significantly reduced.

To gain further insights into the ultrastructure of kinetochores and the mode of KT-MT attachments upon Aurora B[AUK1] inhibition, we performed transmission electron microscopy (TEM) on glutaraldehyde-fixed samples. As described previously (Ogbadoyi et al., 2000), metaphase kinetochores in *T. brucei* appear as electron-dense plaques that contain two "outer layers" (Fig. 2, D and F; and Fig. S2 C). Spindle microtubules appear to terminate in these outer layers, therefore possibly representing the outer kinetochore in trypanosomes. In addition, another electron-dense structure was detected distal to the outer kinetochore, which was particularly visible in detergent-extracted samples fixed with a combination of glutaraldehyde and tannic acid, which improves contrast of certain subcellular structures such as microtubules (Fig. S2 B) (Ogbadoyi et al., 2000; Fujiwara and Linck, 1982). Because the position of this structure corresponds to that of the N-terminus of kinetochore periphery proteins, we propose to call it the "kinetochore periphery" (Fig. 2 F).

We detected at least one bioriented kinetochore in 27 out of 36 (75%) imaged metaphase nuclei, defined by their elongated spindle-like shape, in the MG132 treatment condition. In contrast, only 14 out of 39 (∼36%) metaphase nuclei had clearly identifiable bioriented kinetochores in 1NM-PP1-treated samples. Fig. 2 E and Fig. S2 D show two examples of apparent KT-MT attachment defects upon inhibition of Aurora B[AUK1].

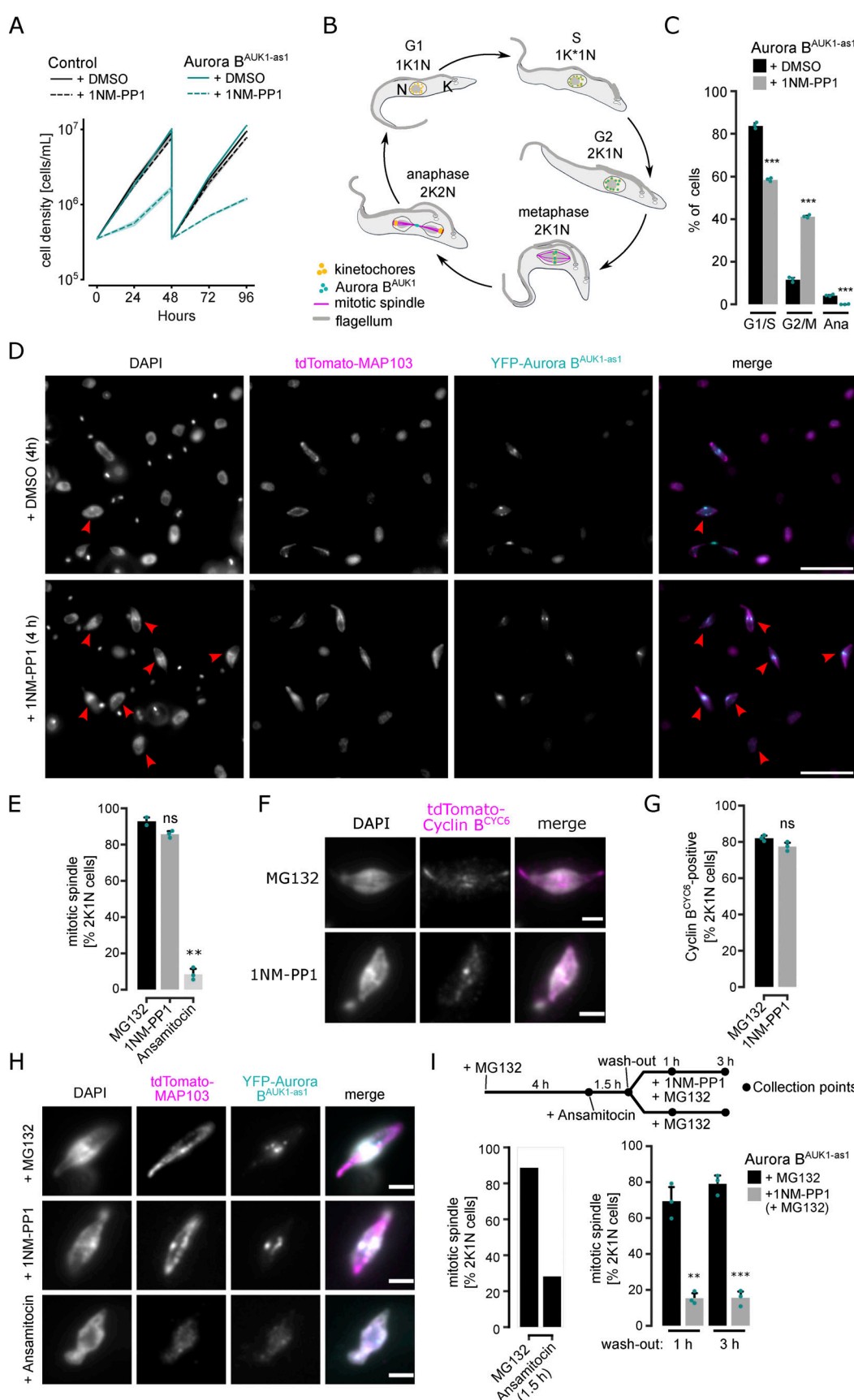

Figure 1.   **Inhibition of Aurora B<sup>AUK1</sup> using an analog-sensitive approach arrests cells in metaphase. (A)** Growth curves upon treatment of control and Aurora B<sup>AUK1-as1</sup> cells with 2 µM 1NM-PP1 or an equal volume of DMSO. The control cell line is heterozygous for the Aurora B<sup>AUK1-as1</sup> allele. Cell densities were

measured at 0, 24, 48, 72, and 96 h. Cultures were diluted at 48 h. Data are presented as the mean ± SD of three replicates. Cell lines: BAP2169, BAP2198. **(B)** Cartoon depicting the kinetoplast (K)/nucleus (N) configuration throughout the cell cycle in procyclic *T. brucei*, with K* denoting an elongated kinetoplast (adapted from Ballmer and Akiyoshi [2024]). The kinetoplast is an organelle found uniquely in kinetoplastids, which contains the mitochondrial DNA. It replicates and segregates prior to nuclear division, so the KN configuration serves as a cell cycle marker (Woodward and Gull, 1990; Siegel et al., 2008). Aurora B$^{AUK1}$ localizes to kinetochores from the S phase until metaphase and translocates to the central spindle in anaphase. **(C)** Cell cycle profile of Aurora B$^{AUK1-as1}$ cells upon treatment with 2 μM 1NM-PP1 or an equal volume of DMSO for 4 h. All graphs depict the means (bar) ± SD of three replicates. A minimum of 450 cells per replicate were quantified. Cell line: BAP2281. **(D)** Representative fluorescence micrographs showing YFP-Aurora B$^{AUK1-as1}$ cells expressing tdTomato-MAP103 (spindle marker) treated with 2 μM 1NM-PP1 or an equal volume of DMSO for 4 h. DNA was stained with DAPI. Red arrowheads indicate 2K1N cells. Cell line: BAP2281. Scale bars, 10 μm. **(E)** Quantification of 2K1N Aurora B$^{AUK1-as1}$ cells that possess a mitotic spindle (marked by tdTomato-MAP103) upon treatment with 10 μM MG132, 2 μM 1NM-PP1 or 5 nM ansamitocin for 4 h. All graphs depict the means (bar) ± SD of at least two replicates (shown as dots). A minimum of 40 cells per replicate were quantified. **(F)** Representative fluorescence micrographs showing the localization of tdTomato-cyclin B$^{CYC6}$ in Aurora B$^{AUK1-as1}$ cells arrested in metaphase upon treatment with 2 μM 1NM-PP1 or 10 μM MG132 for 4 h. Scale bars, 2 μm. Cell line: BAP2356. **(G)** Quantification of Aurora B$^{AUK1-as1}$ 2K1N cells that are positive for tdTomato-cyclin B$^{CYC6}$ upon treatment with 2 μM 1NM-PP1 or 10 μM MG132 for 4 h. All graphs depict the means (bar) ± SD of two replicates (shown as dots). A minimum of 35 cells per replicate were quantified. **(H)** Representative fluorescence micrographs showing the localization of the spindle marker tdTomato-MAP103 and YFP-Aurora B$^{AUK1}$ upon treatment of Aurora B$^{AUK1-as1}$ cells with 10 μM MG132, 2 μM 1NM-PP1, or 5 nM ansamitocin for 4 h. Cell line: BAP2281. Scale bars, 2 μm. **(I)** Upper: Schematic describing experimental design. Aurora B$^{AUK1-as1}$ cells were treated with 10 μM MG132 for 4 h to enrich for cells in metaphase, followed by a 1.5-h treatment with 5 nM ansamitocin to depolymerize the mitotic spindle. Ansamitocin was then washed out and the cells were allowed to recover and reform a spindle with or without 2 μM 1NM-PP1. Collection points are indicated with black dots. Lower: Quantification of 2K1N Aurora B$^{AUK1-as1}$ cells that possess a mitotic spindle under indicated conditions. All graphs depict the means (bar) ± SD of three replicates (dots). A minimum of 90 cells per replicate were quantified. *P < 0.05, **P ≤ 0.01, ***P ≤ 0.001 (two-sided, unpaired *t* test).

Moreover, we found a moderate reduction in the distance between the edges of outer layers (d2) and that between the kinetochore peripheries (d3) (Fig. 2 F) on biorieted kinetochores in Aurora B$^{AUK1}$-inhibited cells, suggesting that tension is indeed reduced across the inter-sister kinetochore axis. In summary, we conclude that Aurora B$^{AUK1}$ activity is required for the establishment of stable biorieted KT–MT attachments.

### Profiling Aurora B$^{AUK1}$ substrates at kinetoplastid kinetochores

We next aimed to identify CPC targets at the trypanosome kinetochore. Substrates of Aurora B kinases in other eukaryotes typically conform to the consensus [RK]-[RK]-X-[ST] (where X is any residue) (Meraldi et al., 2004). We determined the substrate-recognition motif of *T. brucei* Aurora B$^{AUK1}$ by performing a positional scanning peptide array analysis (Hutti et al., 2004) using recombinant Aurora B$^{AUK1}$ bound to its activator INCENP$^{CPC1}$ (Fig. 3, A–C and Fig. S3 A). The consensus motif for trypanosome Aurora B$^{AUK1}$ closely matched that of its homologs in other eukaryotes: selectivity for basic residues N-terminal to the phosphorylation site (with a particularly strong preference for Arg at the −2 position) and a preference for Ser over Thr as the phosphorylation site residue. There also appears to be a unique, though modest, preference for basic residues (R, K, H) at the +2 position, which has not been observed in any of the human Aurora kinases (Johnson et al., 2023). Moreover, Aurora B$^{AUK1}$ strongly deselected peptides containing Pro at position +1.

We next performed in vitro kinase assays using active or kinase-dead (K58R [Li and Wang, 2006]) Aurora B$^{AUK1}$/INCENP$^{CPC1}$ complexes on recombinant CPC or kinetochore proteins as substrates (Fig. 3, D and E; and Fig. S3 B). Among CPC components, we detected strong autophosphorylation of Aurora B$^{AUK1}$ and moderate phosphorylation of INCENP$^{CPC1}$. Interestingly, the C-terminal unstructured tail of KIN-A, which directs kinetochore targeting of the CPC in *T. brucei* (Ballmer and Akiyoshi, 2024), was heavily phosphorylated (Fig. S3 B), raising the possibility that Aurora B$^{AUK1}$ activity may finetune the affinity of KIN-A for its kinetochore receptor(s). The motor

domain of KIN-A was weakly phosphorylated. By contrast, the motor domain of KIN-B was not phosphorylated by Aurora B$^{AUK1}$.

Aurora B$^{AUK1}$ also phosphorylated various KKT proteins in vitro (Fig. 3, D–F), including the inner kinetochore members KKT1, KKT7, and KKT8. KKT7 and KKT8 are components of the KKT7–KKT8 complex, which serves as the main kinetochore receptor of the CPC in *T. brucei* (Ballmer and Akiyoshi, 2024). The outer kinetochore protein KKT4, the only microtubule tip-coupling protein so far identified at the kinetoplastid kinetochore (Llauró et al., 2018), was also phosphorylated. Among the KKT4 fragments tested, KKT4$^{115-343}$ was most strongly phosphorylated by Aurora B$^{AUK1}$ (Fig. S3 C). The fact that KKT4$^{115-343}$ contains the microtubule-binding domain hints at a potential involvement of the CPC in modulating the interaction of the outer kinetochore with microtubules. Indeed, MT cosedimentation assays using a 6HIS-KKT4$^{115-343}$ construct containing phosphomimetic (PM; S/T to D) mutations for putative Aurora B$^{AUK1}$ sites (T234, T266, T267, T268, T316, and S334) had reduced affinity for MTs, whereas the corresponding phosphodeficient (PD; S/T to A) mutant behaved similar to the wild-type control (Fig. S3 D). To test whether phosphorylation of these residues is important for chromosome segregation, we replaced one allele of KKT4 with C-terminally YFP-tagged KKT4$^{PD}$ or KKT4$^{PM}$ constructs and used a previously validated RNAi construct directed against the 3′UTR of KKT4 to deplete the untagged allele (Llauró et al., 2018). Both KKT4$^{PD}$-YFP and KKT4$^{PM}$-YFP were able to rescue the growth defect caused by KKT4 depletion (Fig. S3 E), suggesting that Aurora B$^{AUK1}$-dependent phosphorylation of KKT4 does not significantly affect its MT-binding activity in vivo or that loss of its MT-binding activity can be compensated through other, yet to be identified, MT-binding proteins in trypanosomes.

### Phosphorylation of KKT14 by Aurora B$^{AUK1}$ promotes anaphase entry

Intriguingly, the most robustly phosphorylated kinetochore component was KKT14, an outer kinetochore protein of

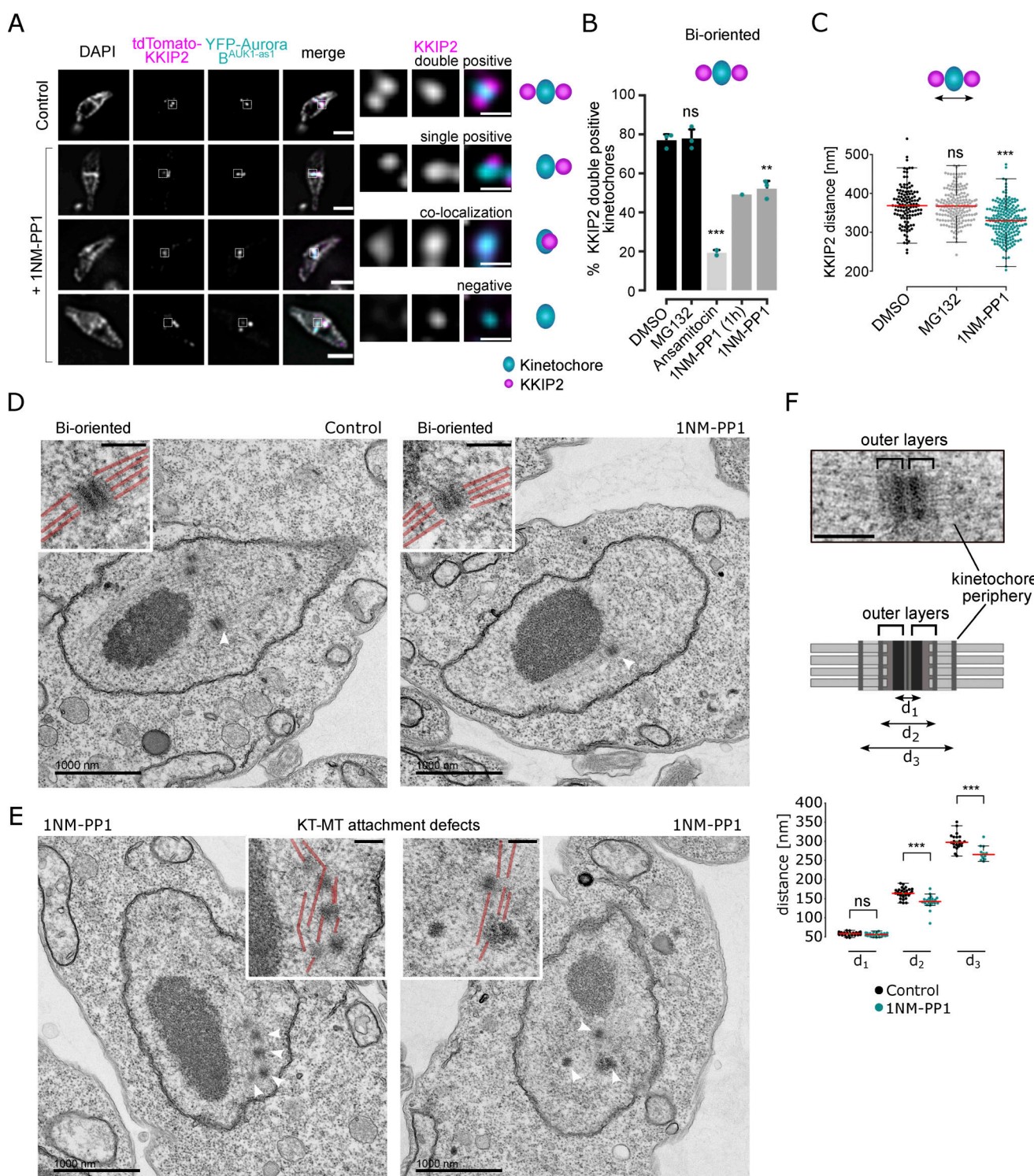

Figure 2. **Aurora B<sup>AUK1</sup> activity is required for the establishment of stable KT-MT attachments. (A)** Representative fluorescence micrographs showing the configuration of tdTomato-KKIP2 (kinetochore periphery, magenta) and YFP-Aurora B$^{AUK1}$ (inner kinetochore, cyan) in Aurora B$^{AUK1-as1}$ cells arrested in metaphase upon treatment with 2 μM 1NM-PP1 or 10 μM MG132 (control) for 4 h, with a schematic guide for each configuration. Note that the kinetochore periphery component KKIP2 undergoes displacement upon biorientation and forms two foci across the inter-sister kinetochore axis ("double positive"). Scale bars, 2 μm. The insets show the magnification of the boxed region (scale bars, 0.5 μm). Cell line: BAP2312. **(B)** Quantification of bioriented kinetochores (as defined in A) in Aurora B$^{AUK1-as1}$ cells in metaphase. Cells were treated with 2 μM 1NM-PP1, 10 μM MG132, and 5 nM ansamitocin or DMSO for 4 h unless otherwise stated. *P < 0.05, **P ≤ 0.01, ***P ≤ 0.001 (two-sided, unpaired $t$ test). **(C)** Quantification of the distance between tdTomato-KKIP2 foci at bioriented kinetochores in Aurora B$^{AUK1-as1}$ cells arrested in metaphase upon treatment with DMSO (black), 10 μM MG132 (gray) or 2 μM 1NM-PP1 (cyan) for 4 h. At least 120 kinetochores (shown as dots) from three replicates were analyzed per condition. The median is indicated in red. *P < 0.05, **P ≤ 0.01, ***P ≤ 0.001 (Mann–Whitney U). **(D and E)** Representative transmission electron micrographs showing bioriented (D) and improperly attached (E) kinetochores in Aurora

$B^{AUK1-as1}$ cells arrested in metaphase upon treatment with 10 µM MG132 or 2 µM 1NM-PP1 for 4 h. Scale bars, 1 µm. White arrowheads indicate the kinetochores shown in magnified insets (scale bars, 200 nm). Microtubules are marked in red in insets. Raw images without microtubules highlighted are shown in Fig. S2, C and D. Cell line: BAP2198. **(F)** Upper: Representative transmission electron micrograph and schematic of a bioriented kinetochore (scale bar, 200 nm). Lower: Quantification of the distance between $d_1$, $d_2$, and $d_3$ at bioriented kinetochores in Aurora $B^{AUK1-as1}$ cells arrested in metaphase upon treatment with 10 µM MG132 (control, black) or 2 µM 1NM-PP1 (cyan) for 4 h. At least 20 kinetochores (shown as dots) from two replicates were analyzed per condition. The median is indicated in red. *P < 0.05, **P ≤ 0.01, ***P ≤ 0.001 (Mann–Whitney U).

unknown function, recently identified to be a distant homolog of the Bub1/BubR1 checkpoint components (Ballmer et al., 2024). We next tested the possibility that KKT14 is a key substrate of Aurora $B^{AUK1}$ that controls the metaphase-to-anaphase transition and/or chromosome segregation in trypanosomes. Interestingly, KKT14 depletion using a previously established RNAi construct (Marcianò et al., 2021; Ballmer et al., 2024) partially rescued the cell cycle arrest caused by Aurora $B^{AUK1}$ inhibition, with some cells progressing into anaphase (Fig. 4, A and B). Nevertheless, these anaphase cells displayed lagging chromosomes and some were negative for Aurora $B^{AUK1}$ (Fig. 4 A), suggesting that they re-entered G1 despite being unable to complete mitosis.

KKT14 consists of an N-terminal region (NTR) harboring an ABBA motif and a C-terminal pseudokinase domain (Fig. S4 A) (Ballmer et al., 2024). We found that Aurora $B^{AUK1}$ strongly phosphorylated the NTR (KKT14$^{2-357}$) but not the pseudokinase domain (KKT14$^{358-685}$) in vitro (Fig. 5 A). 7 out of 11 phosphosites identified by mass spectrometry (MS) match the consensus motif for Aurora $B^{AUK1}$ (e.g., R in position –1, –2, or –3) (Fig. 4 D). To test whether these sites were also phosphorylated in vivo, we performed immunoprecipitation coupled to MS analysis (IP-MS) of GFP-KKT14$^{NTR}$ in Aurora $B^{AUK1-as1}$ cells treated with 1NM-PP1 or MG132 as a control. Many phosphosites in the NTR were downregulated upon Aurora $B^{AUK1}$ inhibition, including three residues (T333, S25, and S113) which were also phosphorylated in vitro and matched the consensus motif for Aurora $B^{AUK1}$ (Fig. S4 B).

To examine the importance of the CPC-dependent phosphorylation of KKT14, we ectopically expressed GFP-KKT14$^{NTR}$ constructs containing wild-type (WT), phosphodeficient (PD; S/T to A), or phosphomimetic (PM; S/T to D/E) with 1 µg/ml doxycycline and monitored cell cycle distribution. We first mutated the phosphorylation sites detected by mass spectrometry (referred to as PD1/PM1). These GFP-KKT14$^{NTR}$ constructs were expressed at similar levels (Fig. S4 C) and localized to kinetochores (Fig. S4 D). We then created additional PD and PM constructs for which either all or only highly conserved $R(x)_{0-2}S/T$ sites (S25, S107, S113, T333) (Ballmer et al., 2024) were targeted (referred to as PD2/PM2 and PD3/PM3, respectively) (Fig. S4 A). We found that expression of WT, PD1, PD2, and to a lesser extent PD3 constructs caused an increase in 2K1N cells (Fig. S4, E, F, and H), and thus partially phenocopied Aurora $B^{AUK1}$ inhibition. In contrast, expression of phosphomimetic KKT14$^{NTR}$ or the C-terminal pseudokinase domain did not delay anaphase onset (Fig. S4, G and H). These results suggest that the expression of KKT14$^{NTR}$ affects cell cycle progression, which is regulated by Aurora $B^{AUK1}$.

We next extended our analysis by inducing GFP-fusions of full-length KKT14 PD and PM constructs in which all $R(x)_{0-2}S/T$

sites in the NTR were targeted with either 0.01 or 1 µg/ml doxycycline (Fig. 5, B and C) and assessed cell growth, cell cycle progression, and chromosome segregation fidelity. Expression of KKT14$^{PD}$ and KKT14$^{PM}$ and to a lesser extent KKT14$^{WT}$ slowed down cell growth and caused lagging kinetochores in anaphase (Fig. 5, D–F and Fig. S5, A–C). Overall, we found that the effect of KKT14$^{PD}$ and KKT14$^{PM}$ was more severe compared with KKT14$^{WT}$, with growth defects and lagging kinetochores observed even when the expression was induced with 0.01 µg/ml doxycycline. To test whether the failure to accurately segregate chromosomes upon KKT14 expression was preceded by a defect in the formation of stable KT-MT attachments in (pro)metaphase, we counted the number of kinetochores double positive for the kinetochore periphery protein KKIP1 in cells treated with MG132. Like KKIP2, KKIP1 undergoes tension-dependent displacement upon proper biorientation (Llauró et al., 2018). Similarly to Aurora $B^{AUK1}$ inhibition, expression of KKT14 was accompanied by a significant reduction in tdTomato-KKIP1 double positive kinetochores (Fig. S5, D and E), suggesting that the lagging kinetochores observed in anaphase indeed stem from a failure to form stable KT-MT attachments and/or properly biorient kinetochores in (pro)metaphase.

Importantly, the expression of full-length KKT14$^{PD}$ but not KKT14$^{WT}$ caused a prominent delay in the metaphase–anaphase transition (Fig. 5, G–I). However, when combined with Aurora $B^{AUK1}$ inhibition, KKT14$^{PD}$ did not further impair cell cycle progression, indicating that KKT14 operates downstream of Aurora $B^{AUK1}$ in regulating the timing of anaphase onset. In line with this, expression of KKT14$^{PM}$ rescued the cell cycle arrest caused by Aurora $B^{AUK1}$ inhibition to a similar extent as observed upon KKT14 depletion (Fig. 5, G and J; and Fig. 4, A and B). Many cells were able to enter an early stage of anaphase, albeit in the presence of lagging kinetochores (Fig. 5 G and Fig. S5 C). The absence of a full rescue upon depletion of KKT14 or expression of the phosphomimetic mutant suggests that Aurora $B^{AUK1}$ phosphorylates additional kinetochore proteins to promote error-free entry into anaphase, and/or that continued activity of Aurora $B^{AUK1}$ at the anaphase central spindle is required for completion of nuclear division.

Taken together, these data demonstrate that phosphorylation of KKT14 by Aurora $B^{AUK1}$ modulates the timing of anaphase entry in trypanosomes.

### Localization of Aurora $B^{AUK1}$ to the inner kinetochore is important for error-free chromosome segregation

The prevailing model explaining how Aurora B recognizes and corrects improper KT–MT attachments relies on proximity between outer kinetochore proteins and centromeric pools of Aurora B, which become spatially separated as tension builds up

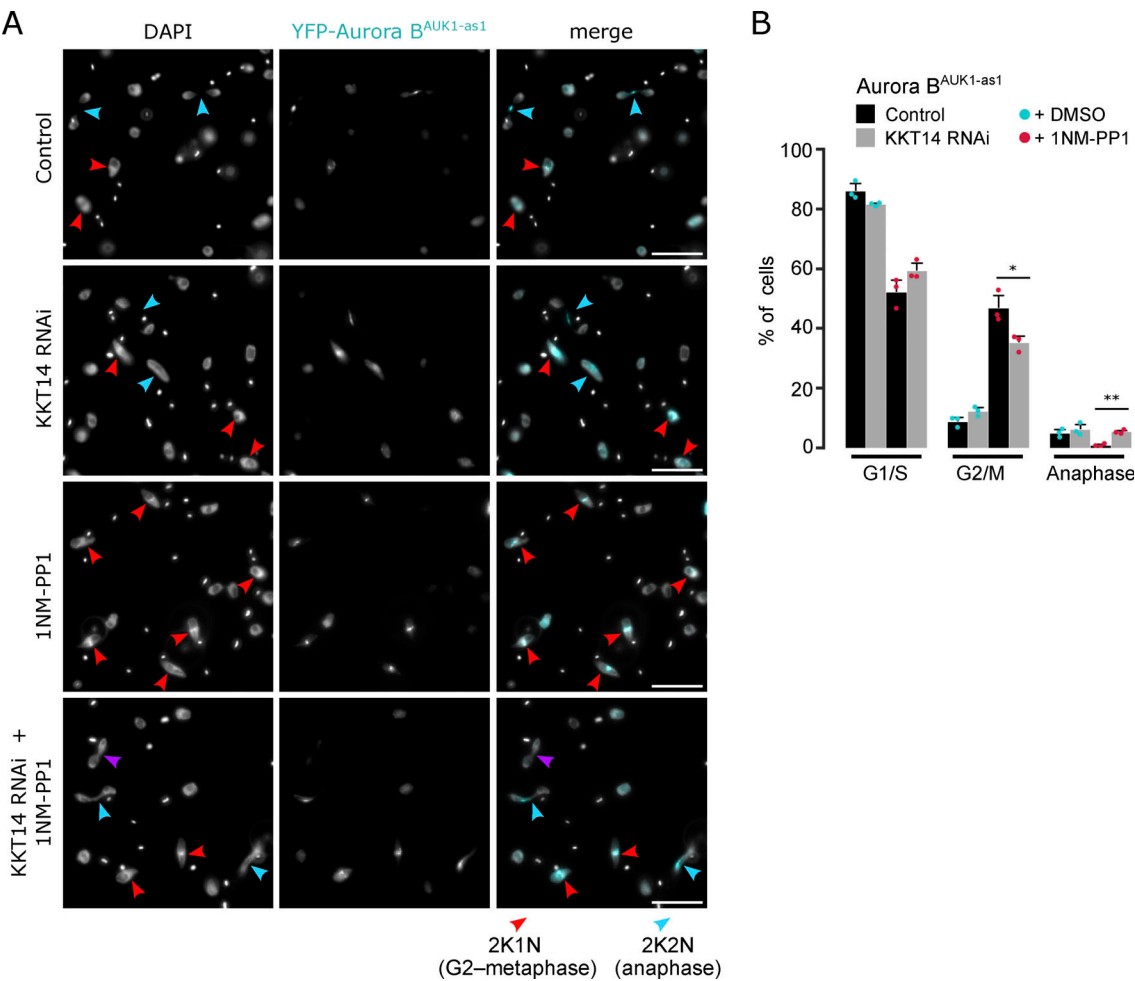

Figure 3. **Profiling Aurora B$^{AUK1}$ in vitro substrates. (A)** Positional scanning peptide array image of recombinant 3FLAG-Aurora B$^{AUK1}$/INCENP$^{CPC1}$. Darker spots indicate preferred residues. The second run is shown in Fig. S3. **(B)** Quantification of A. Spot intensities were normalized so that the average value within a position was equal to one. The heatmap shows the log$_2$ transformed data (averaged from the two separate runs) with positive selections shown in red and negative selections shown in blue. **(C)** Aurora B$^{AUK1}$ substrate motif logo. **(D and E)** Aurora B$^{AUK1}$ in vitro kinase assay using the indicated recombinant kinetochore proteins as substrates. The left panel (input) shows the Coomassie Brilliant Blue staining. Substrates are marked with red dots. Phosphorylation was detected by autoradiography. Note that KKT2, KKT3, and KKT10$^{CLK1}$ are kinases themselves and display autophosphorylation. **(F)** Normalized $^{32}$P signal intensities for indicated proteins relative to Aurora B$^{AUK1}$ autophosphorylation. The $^{32}$P signal intensity for each substrate was normalized to the total protein amount (estimated by measuring the intensity of Coomassie-stained bands). To correct for non-Aurora B$^{AUK1}$-dependent phosphorylation, the normalized intensities from the kinase-dead controls were subtracted from these values. Source data are available for this figure: SourceData F3.

**Figure 4. Depletion of KKT14 partially rescues the cell cycle arrest caused by Aurora B[AUK1] inhibition. (A)** Representative fluorescence micrographs showing Aurora B[AUK1-as1] cells treated with 2 µM 1NM-PP1 or an equal volume of DMSO for 4 h. Prior to that, RNAi-mediated knockdown of KKT14 was induced with 1 µg/ml doxycycline for 20 h. DNA was stained with DAPI. Arrowheads indicate 2K1N (red) and 2K2N (light blue) cells. Purple arrowhead indicates a 2K2N cell that is negative for Aurora B[AUK1], suggesting a re-entry into G1 despite failure to complete nuclear division. Cell line: BAP2469. Scale bars, 10 µm. **(B)** Cell cycle profile for indicated conditions as in A. All graphs depict the means (bar) ± SD of three replicates. A minimum of 300 cells per replicate were quantified. Cell line: BAP2469. *P < 0.05, **P ≤ 0.01, ***P ≤ 0.001 (two-sided, unpaired *t* test).

across the inter-sister kinetochore axis in properly bioriented kinetochores (Liu et al., 2009; Tanaka et al., 2002). However, this spatial separation model has been challenged by the observations that the inner centromere localization of Aurora B is dispensable for chromosome biorientation in various model organisms (Hengeveld et al., 2017; Campbell and Desai, 2013; Yue et al., 2008). A revised spatial separation model posits that localization of Aurora B at centromeres or inner kinetochores is required for biorientation (Fischböck-Halwachs et al., 2019; García-Rodríguez et al., 2019; Li et al., 2023). Trypanosome kinetochores offer an interesting opportunity to test this model because trypanosomes intrinsically lack inner centromeres (Tromer et al., 2021) and their CPC localizes primarily via inner kinetochore proteins (Ballmer and Akiyoshi, 2024). To test the importance of its spatial regulation in trypanosomes, we established a nanobody-based assay to target Aurora B to the inner or outer kinetochore (Fig. 6, A and B). We fused the catalytic module of the CPC (Aurora B[AUK1] + INCENP[CPC1 148–263]) to tdTomato and a nanobody recognizing GFP or YFP (VhhGFP4

[Ishii and Akiyoshi, 2022; Saerens et al., 2005]) (CPC[cat]-tdTomato-vhhGFP4) (Fig. 6 B), enabling the fusion protein to be tethered to YFP-tagged inner (KKT3, KKT9) or outer (KKT4, KK14) kinetochore components. Importantly, CPC[cat]-tdTomato-VhhGFP4 itself is not expected to localize at inner kinetochores because it lacks the N-terminal domain of INCENP[CPC1] that binds the KIN-A:KIN-B scaffold (Ballmer and Akiyoshi, 2024). Indeed CPC[cat]-tdTomato-VhhGFP4 failed to localize at kinetochores in the absence of YFP-tagged kinetochore proteins (Fig. 6 C). In contrast, CPC[cat]-tdTomato-VhhGFP4 colocalized with YFP-tagged inner or outer KKT proteins (Fig. 6, D–G). By incorporating this system into our Aurora B[AUK1-as1] background cell lines, we selectively inhibited the endogenous kinase, ensuring that the only active Aurora B[AUK1] molecule was derived from the fusion protein (Fig. 6 B). Following the induction of the fusion constructs, we treated cells with 1NM-PP1 or DMSO for 4 h and scored cell cycle distribution and lagging kinetochores in anaphase (Fig. 6 B). As expected, untethered CPC[cat]-tdTomato-VhhGFP4 failed to rescue the effects of Aurora B[AUK1] inhibition

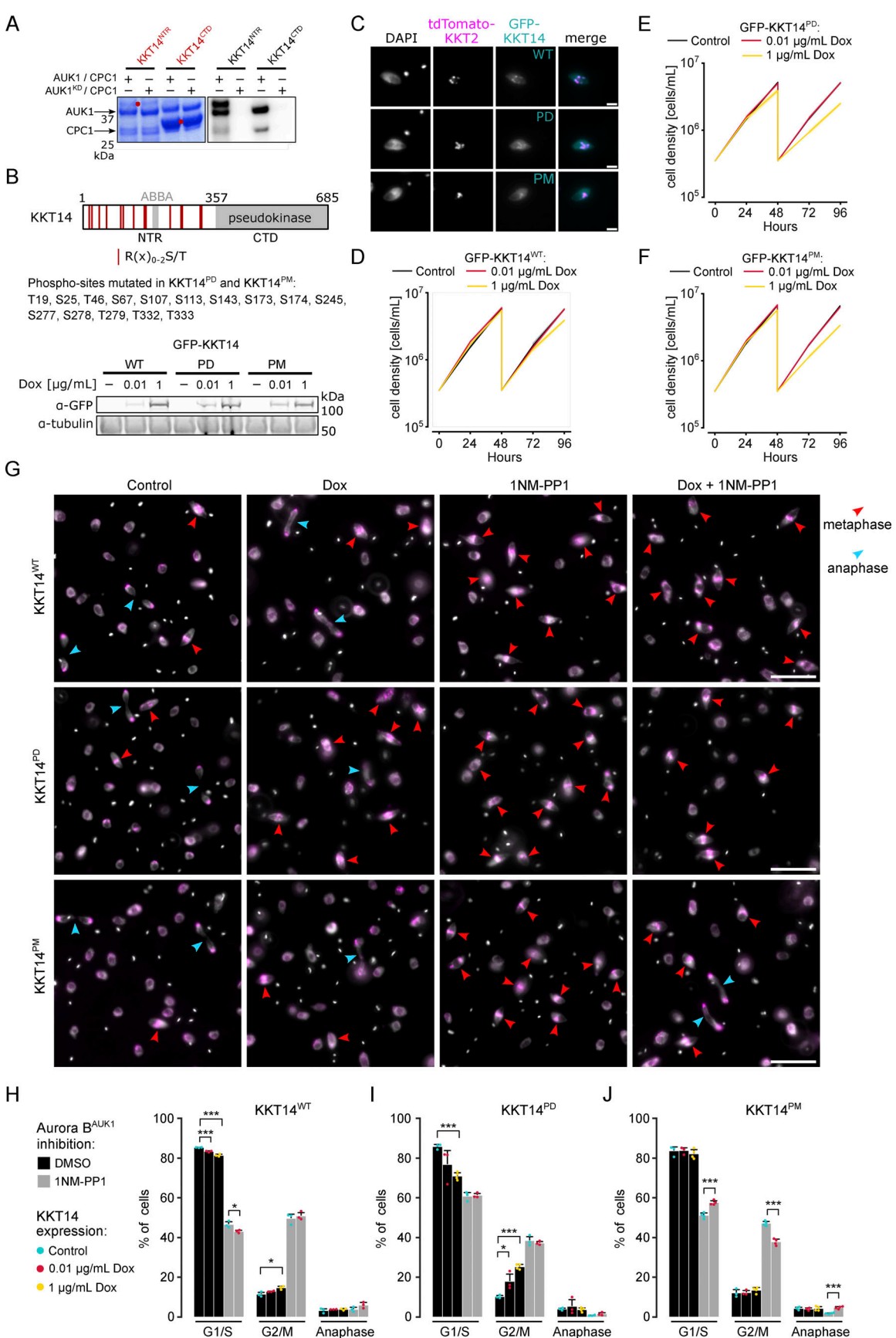

Figure 5. **Phosphorylation of KKT14 by Aurora B$^{AUK1}$ regulates anaphase entry. (A)** Aurora B$^{AUK1}$ in vitro kinase assay using the indicated recombinant KKT14 constructs as substrates. The left panel (input) shows the Coomassie Brilliant Blue staining. Substrates are marked with red dots. Phosphorylation was

detected by autoradiography. **(B)** Upper: Schematic representation of KKT14 showing NTR and C-terminal pseudokinase domain (CTD). NTR phosphorylation sites targeted in phosphodeficient (PD) and phosphomimetic (PM) constructs correspond to the Aurora B$^{AUK1}$ consensus motif (R(x)$_{0-2}$S/T) and are indicated by red lines. Lower: Western blot showing protein levels of indicated GFP-KKT14 constructs, induced with 0.01 or 1 μg/ml doxycycline for 24 h. Tubulin was used as a loading control. Cell lines: BAP3228, BAP3212, and BAP3213. **(C)** Representative fluorescence micrographs showing localization of indicated GFP-KKT14 constructs and tdTomato-KKT2 (kinetochore marker). Expression of fusion proteins was induced with 0.01 μg/ml doxycycline and cells were fixed at 24 h. Cell lines: BAP3228, BAP3212, BAP3213. Scale bars, 2 μm. **(D–F)** Growth curves upon expression of GFP-KKT14$^{WT}$ (D), -KKT14$^{PD}$ (E) and -KKT14$^{PM}$ (F) induced with 0.01 (red) or 1 μg/ml (yellow) doxycycline. Cell densities were measured at 0, 24, 48, 72, and 96 h. Cultures were diluted at 48 h. Data are presented as the mean ± SD of three replicates. Cell lines: BAP3228, BAP3212, BAP3213. **(G)** Representative fluorescence micrographs showing cell cycle distribution upon indicated treatment conditions. Expression of KKT14 constructs ("Dox") was induced with 1 μg/ml doxycycline for 24 h. Cells were treated with 2 μM 1NM-PP1 for 4 h to inhibit Aurora B$^{AUK1}$ kinase activity ("1NM-PP1"). In the rescue condition ("Dox + 1NM-PP1"), expression of indicated KKT14 constructs was induced with 0.01 μg/ml doxycycline for 24 h. TdTomato-KKT2 marks kinetochores and DNA was stained with DAPI. Arrowheads indicate metaphase (red) and anaphase (light blue) cells. Cell lines: BAP3228, BAP3212, and BAP3213. Scale bars, 10 μm. **(H–J)** Cell cycle profiles upon indicated treatment conditions as in G. All graphs depict the means (bar) ± SD of at least three replicates (dots). A minimum of 300 cells per replicate were quantified. Cell lines: BAP3228, BAP3212, BAP3213. *P < 0.05, **P ≤ 0.01, ***P ≤ 0.001 (two-sided, unpaired t test). Source data are available for this figure: SourceData F5.

(Fig. 6 H), while tethering the fusion protein to the inner kinetochore proteins KKT3 or KKT9 (a key interaction partner of the CPC [Ballmer and Akiyoshi, 2024]) did (Fig. 6 H). Intriguingly, targeting CPC$^{cat}$-tdTomato-VhhGFP4 to the outer kinetochore components KKT4 or KKT14 partially restored cell cycle progression but resulted in a massive increase of lagging chromosomes in anaphase (Fig. 6, H and I). This is unlikely to be an artifact of impairing KT-MT attachments due to steric hindrance (e.g., by physically blocking access of MTs to binding sites at the kinetochore) because tethering of CPC$^{cat}$-tdTomato-VhhGFP4 to these outer kinetochore proteins without inhibiting the endogenous Aurora B$^{AUK1}$ did not affect cell cycle progression and sustained error-free chromosome segregation (Fig. 6, H and I). Nevertheless, we cannot exclude that tethering may have additional effects independently of target phosphorylation. In summary, our results imply that localization of Aurora B$^{AUK1}$ to either the inner or outer kinetochore is sufficient to promote anaphase entry in trypanosomes.

## Discussion

From yeast to humans, Aurora kinases impart fidelity to cell division by regulating various processes including kinetochore assembly, chromosome biorientation, and SAC signaling (Krenn and Musacchio, 2015). Our results provide the first in-depth analysis of the role of Aurora B$^{AUK1}$ in regulating chromosome segregation in kinetoplastids, a group of flagellated protists proposed to be among the earliest branching eukaryotes (Allen et al., 2008; Cavalier-Smith, 2010; Akiyoshi and Gull, 2013). Like in other eukaryotes, kinetochores fail to form proper bioriented attachments upon Aurora B$^{AUK1}$ inhibition. In vitro, Aurora B$^{AUK1}$ strongly phosphorylated the MT-binding domain of KKT4, and MT co-sedimentation assays using a phosphomimetic KKT4 mutant found a reduced affinity for MTs. This suggests that the trypanosome CPC may be involved in an error correction process analogous to that described in other eukaryotes. Consistent with this notion, artificial targeting of Aurora B$^{AUK1}$ to the inner kinetochore (but not to the outer kinetochore) allowed proper chromosome segregation in anaphase. Thus, the overall mechanistic principles of error correction, e.g., tension-dependent regulation of "outer" MT-binding proteins that are spatially separated from a main "inner" pool of the CPC may be evolutionarily conserved.

Remarkably, inhibition of Aurora B$^{AUK1}$ in *T. brucei* arrests cells in metaphase, a phenotype that has not been reported in traditional model eukaryotes (Biggins and Murray, 2001; Hauf et al., 2003). Our data suggest that the divergent Bub1-homolog KKT14 is a main target of Aurora B$^{AUK1}$. The KKT14 NTR contains an ABBA motif (a conserved CDC20-interaction motif) (Ballmer et al., 2024), suggesting that this domain might be involved in regulating APC/C activity. Although direct binding to CDC20 and/or APC/C subunits remains to be demonstrated, it is possible that the KKT14 NTR in its unphosphorylated state might prevent premature APC/C activation by sequestering CDC20 or certain subunits of the APC/C, and that this inhibition could be relieved by Aurora B$^{AUK1}$ activity. Alternatively, the NTR in its phosphorylated state may act as a scaffold that promotes APC/C–CDC20 interaction and its subsequent activation. A third possibility is that kinetoplastids evolved an APC/C- or CDC20-independent mechanism to regulate the metaphase–anaphase transition.

Apart from KKT14, Aurora B$^{AUK1}$ also phosphorylated several CPC subunits in vitro, including its activator INCENP$^{CPC1}$ and the C-terminal tail of KIN-A. We recently demonstrated that the C-terminal unstructured tail of KIN-A plays a key role in targeting the CPC to kinetochores by interacting with the KKT8 complex at the inner kinetochore. We speculate that Aurora B$^{AUK1}$-dependent phosphorylation of KIN-A may modulate its kinetochore-binding affinity. Importantly, inhibition of Aurora B$^{AUK1}$ using our analog-sensitive approach did not impair kinetochore localization of the CPC. Rather, phosphorylation of the KIN-A C-terminal tail could serve to weaken the association of KIN-A with its kinetochore receptors and facilitate the release of the CPC from kinetochores onto spindle microtubules upon metaphase–anaphase transition. An intriguing possibility is that the interaction of the KIN-A motor domain with microtubules in (pro)metaphase, coupled to the establishment of proper kinetochore-microtubule attachments, may trigger conformational changes within the CPC (such as tension-dependent stretch of the KIN-A C-terminus) and/or full activation of Aurora B$^{AUK1}$, thereby allowing temporally and spatially regulated phosphorylation of the KIN-A C-terminal tail. Whether the timely release of KIN-A from kinetochores may be a prerequisite for mitotic exit in addition to KKT14 phosphorylation will require further testing in the future.

Further potential targets of the trypanosome CPC identified through our in vitro kinase assays include the inner kinetochore

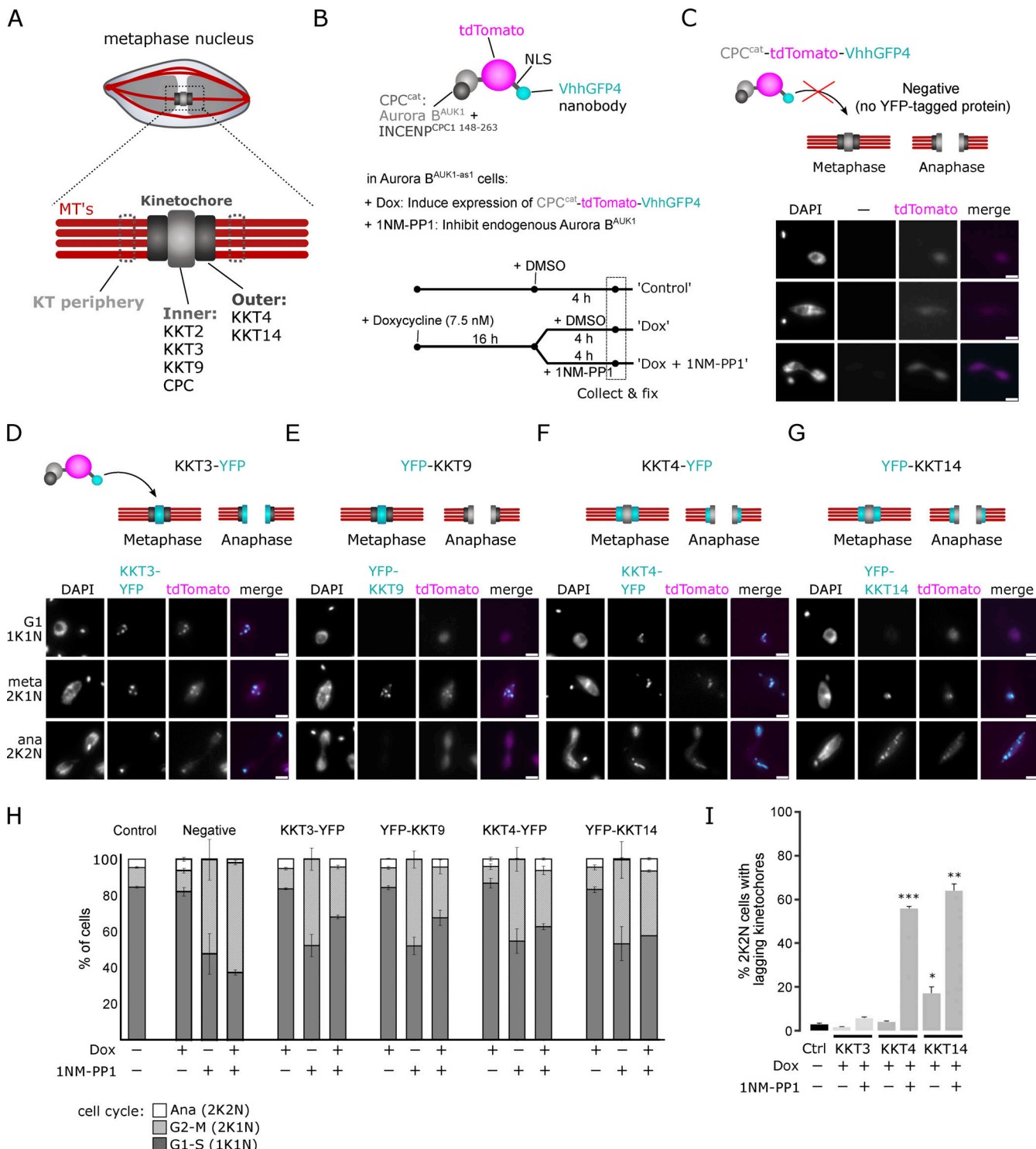

Figure 6. **Nanobody-based targeting of Aurora B$^{AUK1}$ to the inner or outer kinetochore. (A)** Schematic illustration of the trypanosome kinetochore, indicating proteins localizing to the inner or outer kinetochore. **(B)** Top: Schematic illustration of CPC$^{cat}$-tdTomato-VhhGFP4. Aurora B$^{AUK1}$ is fused to the C-terminal domain of INCENP$^{CPC1}$, which binds to Aurora B$^{AUK1}$ and contains the IN-box required for full Aurora B$^{AUK1}$ activity but lacks the regions required to interact with endogenous KIN-A:KIN-B at the inner kinetochore (Ballmer and Akiyoshi, 2024). The fusion construct also contains a nuclear localization signal (NLS) between tdTomato and VhhGFP4. Bottom: Schematic of Aurora B$^{AUK1}$ targeting experiment. Expression of CPC$^{cat}$-tdTomato-VhhGFP4 was induced for 16 h using 7.5 ng/ml doxycycline in cell lines harboring YFP-tagged inner (KKT3, KKT9) or outer (KKT4, KKT14) kinetochore proteins, followed by addition of either DMSO (control) or 2 μM 1NM-PP1 for 4 h to inhibit the endogenous Aurora B$^{AUK1}$ kinase. Cells were then fixed and cell cycle distribution and lagging kinetochores were scored. **(C)** Fluorescence micrographs showing diffuse nuclear localization of CPC$^{cat}$-tdTomato-VhhGFP4 induced with 7.5 ng/ml doxycycline in a cell line lacking YFP-tagged kinetochore proteins. Cell line: BAP2671. Scale bars, 2 μm. **(D–G)** Representative fluorescence micrographs showing the colocalization of CPC$^{cat}$-tdTomato-vhhGFP4 with YFP-tagged KKT3 (D), KKT9 (E), KKT4 (F), and KKT14 (G). The localization dynamics of the YFP-tagged kinetochore proteins (marked in cyan) in metaphase and anaphase are schematically depicted on top. Cell lines: BAP2673, BAP2990, BAP2991, and BAP2992.

Scale bars, 2 μm. **(H)** Cell cycle profiles for indicated treatment regimes. "Control" cells were treated with DMSO for 4 h. "Negative" control corresponds to a cell line that does not express any YFP-tagged protein (as shown in C). All graphs depict the means (bar) ± SD of at least two replicates. A minimum of 500 cells per replicate were quantified. **(I)** Quantification of lagging kinetochores in 2K2N cells under indicated treatment regimes. "Control" cells were treated with DMSO for 4 h. Note that lagging kinetochores could not be assessed in the cell line expressing YFP-KKT9, because KKT9 is not present at kinetochores in anaphase. All graphs depict the means (bar) ± SD of at least two replicates (dots). A minimum of 35 cells per replicate were quantified. *P < 0.05, **P ≤ 0.01, ***P ≤ 0.001 (two-sided, unpaired $t$ test).

components KKT1, KKT7, and KKT8, which are also substrates of the two functionally redundant KKT10/19 kinases (also called CLK1/2) (Ishii and Akiyoshi, 2020). Interestingly, KKT10 and KKT19 localize to the inner kinetochore by binding to the N-terminus of KKT7, which in conjuncture with the KKT8 complex also serves as the main recruitment arm for the trypanosome CPC (Ishii and Akiyoshi, 2020; Ballmer and Akiyoshi, 2024). Moreover, depletion of KKT10/19 causes a delay in the metaphase–anaphase progression and lagging kinetochores in anaphase, raising the possibility that these kinases may be involved in some form of error correction process and/or form part of the regulatory circuitry controlling anaphase onset. Contrary to Aurora B[AUK1], however, KKT10/19 phosphorylates the C-terminal domain of KKT4 rather than its MT-binding domain (Ishii and Akiyoshi, 2020), and the mitotic spindle appears to be hyperstabilized rather than destabilized in these mutants (unpublished observations). Thus, it is conceivable that KKT10/19 and Aurora B[AUK1] may play opposing roles for spindle stability despite their close spatial association.

We previously found that retargeting the CPC from kinetochores onto the mitotic spindle in metaphase was still able to support entry into anaphase (Ballmer and Akiyoshi, 2024). Here, we demonstrate that this is also true upon tethering of a CPC[cat]-tdTomato-VhhGFP4 fusion protein to the outer kinetochore while inhibiting endogenous Aurora B[AUK1-as1], which localizes at the inner kinetochore. Together, these results imply that the ability of Aurora B[AUK1] to trigger mitotic exit is not strictly dependent on its recruitment to the inner kinetochore and hence may not be spatially regulated. Rather, the trypanosome CPC may sense conformational changes when the kinetochore comes under tension upon forming stable, bioriented attachments to the spindle, which would allow conditional phosphorylation of substrates (such as KKT14) involved in the metaphase–anaphase transition. Identification and functional characterization of additional CPC substrates at kinetochores/centromeres will be key to understanding how anaphase onset is controlled in kinetoplastids.

Our results indicate that Aurora B[AUK1] regulates two key processes at the kinetochore: the stability of KT–MT attachments at each kinetochore and the entry into anaphase. Thus, the CPC acts as a master regulator of chromosome segregation in *T. brucei*. We envisage that the CPC may be involved in various pathways at the kinetoplastid kinetochore. Disentangling the contributions of Aurora B[AUK1] to chromosome segregation in kinetoplastids and exploring its regulatory crosstalk with the other kinetochore-localized kinases (KKT10/19, KKT2, KKT3, CDK[CRK3]) will prove to be a challenging task but is bound to provide important insights into the evolution of the mitotic circuitry governing eukaryotic cell division.

## Materials and methods
### Cloning
All primers, plasmids, bacmids, and synthetic DNA used in this study as well as their source or construction details are described in Table S1. All constructs were sequence-verified.

### Trypanosome culture
All trypanosome cell lines used in this study were derived from *T. brucei* SmOxP927 procyclic form cells (TREU 927/4 expressing T7 RNA polymerase and the tetracycline repressor to allow inducible expression [Poon et al., 2012]) and are described in Table S1. Cells were grown at 28°C in SDM-79 medium supplemented with 10% (vol/vol) heat-inactivated fetal calf serum, 7.5 μg/ml hemin (Brun and Schönenberger, 1979), and appropriate selection drugs. Cell growth was monitored using a CASY cell counter (Roche). PCR products or plasmids linearized by NotI were transfected into cells by electroporation (Biorad). Transfected cells were selected by the addition of 30 μg/ml G418 (Sigma-Aldrich), 50 μg/ml hygromycin (Sigma-Aldrich), 5 μg/ml phleomycin (Sigma-Aldrich), or 10 μg/ml blasticidin S (Insight Biotechnology). To obtain endogenously tagged clonal strains, transfected cells were selected by the addition of appropriate drugs and cloned by dispensing dilutions into 96-well plates. Endogenous YFP tagging was performed using the pEnT5-Y vector (Kelly et al., 2007) or a PCR-based method (Dean et al., 2015). Endogenous tdTomato tagging was performed using pBA148 (Akiyoshi and Gull, 2014) and its derivatives. For doxycycline-inducible expression of head-to-head (pBA3-based) and hairpin (pBA310-based) RNAi constructs, as well as GFP-NLS (pBA310-based), the linearized plasmids were integrated into 177-bp repeats on minichromosomes. Expression of GFP-KKT14 fusions was induced for 24 h by the addition of 1 μg/ml or 10 ng/ml doxycycline as indicated. Expression of RNAi constructs was induced by the addition of 1 μg/ml doxycycline for indicated time periods. Expression of CPC[cat]-tdTomato-vhhGFP4 was induced by the addition of 7.5 ng/ml doxycycline.

### Fluorescence microscopy
Cells were washed once with PBS, settled onto glass slides, and fixed with 4% paraformaldehyde in PBS for 5 min (Nerusheva and Akiyoshi, 2016). Cells were then permeabilized with 0.1% NP-40 in PBS for 5 min and embedded in mounting media (1% wt/vol 1,4-diazabicyclo[2.2.2]octane, 90% glycerol, 50 mM sodium phosphate, pH 8.0) containing 100 ng/ml DAPI. Images were captured at room temperature on a Zeiss Axioimager.Z2 microscope (Zeiss) installed with ZEN using a Hamamatsu ORCA-Flash4.0 camera with 63× objective lenses (1.40 NA). Typically, ~20 optical slices spaced 0.24 μm apart were collected. Images were analyzed in ImageJ/Fiji (Schneider et al.,

2012). The mean intensity of segmented kinetochore foci for each cell was calculated as follows using a custom macro (Data S1). First, metaphase cells were manually selected from the images. Then, nuclei regions were segmented in 3D with a default threshold value in the ImageJ threshold tool at the focus plane. Kinetochores were detected in the nuclei regions by the 3D object counter tool in ImageJ with the threshold determined by the default threshold value at the focus plane. The total fluorescent signal intensity of the kinetochore protein in a nucleus was calculated by summing up the intensity detected by a 3D object counter, which was standardized using z-scores. For analysis of kinetochore biorientation using tdTomato-KKIP2, images were captured on a DeltaVision OMX V3 or SR housed in the Oxford Micron facility. Fluorescent images were captured at room temperature using 60x objective lenses (1.42 NA) (typically 16–21 z sections at 0.25-μm steps) and deconvoluted using softWoRx. Bioriented kinetochores were scored manually by quantifying the number of kinetochores (marked by YFP-Aurora B$^{AUK1}$) that were double positive for tdTomato-KKIP2. Images shown in the figure are central slices. The distance between KKIP2 dots was measured in ImageJ using the "plot profile" function across the inter-sister kinetochore axis and measuring the distance between the two peaks.

**Transmission electron microscopy (TEM)**
Whole cells were fixed in 2% glutaraldehyde and 2% formaldehyde in PEME buffer (100 mM PIPES-NaOH, pH 6.9, 2 mM EGTA, 1 mM MgSO$_4$, and 0.1 mM EDTA) for 1 h at RT. Fixed samples were then centrifuged at 800 $g$ for 10 min at RT and the pellet was resuspended in PIPES buffer (0.1 M PIPES at pH 7.2). After several washes in PIPES buffer (five changes of fresh buffer, each followed by 5 min incubation at RT, rotating), a quenching step (50 mM glycine in PIPES buffer for 15 min at RT, rotating), and a final wash (10 min incubation at RT, rotating), samples were postfixed with 1% osmium tetroxide and 1.5% potassium ferrocyanide in PIPES buffer for 1 h at 4°C. Samples were washed five times with Milli-Q H$_2$O and embedded in 4% LMP agarose. The agarose-embedded samples were kept at 4°C for 15 min to allow the agarose to set and were then cut into small blocks, which were then stained with 0.5% aqueous uranyl acetate at 4°C in the dark o/n. Following several wash steps in H$_2$O, samples were dehydrated through an ethanol series and gradually infiltrated with Agar low viscosity resin. The samples were then transferred into embedding capsules, and the resin was polymerized at 60°C for 24 h. Ultrathin (90 nm) sections were taken with a Diatome diamond knife on a Leica UC7 ultramicrotome and mounted onto 200 mesh copper grids, which were then poststained in lead citrate for 5 min at RT. Grids were imaged in a Tecnai FEI T12 transmission electron microscope (TEM) operated at 120 kV with a Gatan Oneview digital camera. For detailed visualization of kinetochores and MTs, cells were extracted in 1% NP-40 in PEME for 5 min at RT and centrifuged at 1,800 $g$ for 15 min at RT. The pellet was resuspended in a fixing buffer containing 4% glutaraldehyde and 1% tannic acid in PEME. All subsequent steps were performed as described above.

**Immunoprecipitation followed by mass spectrometry (IP-MS)**
400 ml cultures were grown to ∼5–10 million cells/ml. Expression of GFP-KKT14$^{NTR}$ was induced with 10 ng/ml doxycycline for 24 h. After 4 h of treatment with 10 μM of MG132 (control) or 2 μM 1NM-PP1, cells were pelleted by centrifugation (800 $g$, 10 min), washed once with PBS, and extracted in PEME (100 mM PIPES-NaOH, pH 6.9, 2 mM EGTA, 1 mM MgSO$_4$, and 0.1 mM EDTA) with 1% NP-40, protease inhibitors (10 μg/ml leupeptin, 10 μg/ml pepstatin, 10 μg/ml E-64, and 0.2 mM PMSF), and phosphatase inhibitors (1 mM sodium pyrophosphate, 2 mM Na-β-glycerophosphate, 0.1 mM Na$_3$VO$_4$, 5 mM NaF, and 100 nM microcystin-LR) for 5 min at RT, followed by centrifugation at 1,800 $g$ for 15 min. Samples were kept on ice from this point on. The pelleted fraction containing kinetochore proteins was resuspended in modified buffer H (BH0.15: 25 mM Hepes, pH 8.0, 2 mM MgCl$_2$, 0.1 mM EDTA, pH 8.0, 0.5 mM EGTA, pH 8.0, 1% NP-40, 150 mM KCl, and 15% glycerol) with protease and phosphatase inhibitors. Samples were sonicated to solubilize kinetochore proteins (12 s, three times with 1-min intervals on ice). 12 μg of mouse monoclonal anti-GFP antibodies (11814460001; Roche), preconjugated with 60 μl slurry of Protein-G magnetic beads (10004D; Thermo Fisher Scientific) with dimethyl pimelimidate (Unnikrishnan et al., 2012), was incubated with the extracts for 2.5 h with constant rotation, followed by four washes with modified BH0.15 containing protease inhibitors, phosphatase inhibitors, and 2 mM DTT. Beads were further washed three times with pre-elution buffer (50 mM Tris-HCl, pH 8.3, 75 mM KCl, and 1 mM EGTA). Bound proteins were eluted from the beads by agitation in 60 μl of elution buffer (50 mM Tris-HCl, 0.3% SDS, pH 8.3) for 25 min at RT. Reduction of disulfide bridges in cysteine-containing proteins was performed with 10 mM DTT dissolved in 50 mM HEPES, pH 8.5 (56°C, 30 min). Reduced cysteines were alkylated with 20 mM 2-chloroacetamide dissolved in 50 mM HEPES, pH 8.5 (room temperature, in the dark, 30 min). Samples were prepared using the SP3 protocol (Hughes et al., 2019), and trypsin (Promega) was added in the 1:50 enzyme-to-protein ratio for overnight digestion at 37°C. The next day, peptide recovery was done by collecting the supernatant on a magnet and combining it with the second elution of beads with 50 mM HEPES, pH 8.5. For a further sample clean up, an OASIS HLB μElution Plate (Waters) was used. The samples were dissolved in 10 μl of reconstitution buffer (96:4 water: acetonitrile, 1% formic acid and analyzed by LC-MS/MS using QExactive (Thermo Fisher Scientific) in the proteomics core facility at EMBL Heidelberg (https://www.embl.org/groups/proteomics/).

Peptides were identified by searching tandem mass spectrometry spectra against the *T. brucei* protein database with MaxQuant (version 2.0.1) with carbamidomethyl cysteine set as a fixed modification and oxidization (Met), phosphorylation (Ser, Thr, and Tyr), and acetylation (Lys) set as variable modifications. Up to two missed cleavages were allowed. The first peptide tolerance was set to 10 ppm. Results were filtered to remove contaminants and reverse hits. Differential enrichment analysis of phosphopeptides was performed using the DEP package in R (Zhang et al., 2018). Reverse hits and contaminants were removed, and the results were filtered for peptides that

were identified in all replicates of at least one condition. The data was background corrected and normalized by variance stabilizing transformation (vsn). Missing values were imputed using the k-nearest neighbor approach (knn). Raw mass spectrometry files and the custom database file used in this study have been deposited to the ProteomeXchange Consortium via the PRIDE partner repository (Perez-Riverol et al., 2019; Deutsch et al., 2023) with the dataset identifier PXD047806 (GFP-KKT14N_MG132_rep1/2 and GFP-KKT14N_1NM-PP1_rep1/2).

## Expression and purification of recombinant proteins from *E. coli* and insect cells

Recombinant 6HIS-tagged KKT4 fragments (pBA1413: KKT4$^{2-114}$, pBA1065: KKT4$^{115-343}$, pBA2714: KKT4$^{115-343}$ PD, pBA2715: KKT4$^{115-343}$ PM, pBA1641: KKT4$^{300-488}$, pBA1513: KKT4$^{463-645}$), KKT14 fragments (pBA2704: KKT14$^{2-357}$, pBA2353: KKT14$^{358-685}$), and CPC fragments (pBA2519: KIN-A$^{2-309}$, pBA2513: KIN-B$^{2-316}$, pBA2574) were expressed in *E. coli* BL21(DE3) cells and purified and eluted from TALON beads as described previously (Llauró et al., 2018; Ishii and Akiyoshi, 2020). Briefly, cells were grown in 2xTY media at 37°C to an OD600 of ~0.8. Protein expression was induced by the addition of 0.1 mM IPTG overnight at 16–20°C. Cells were pelleted and resuspended in lysis buffer (50 mM sodium phosphate, pH 7.5, 500 mM NaCl, 5 mM imidazole, and 10% glycerol) supplemented with protease inhibitors (20 µg/ml leupeptin, 20 µg/ml pepstatin, 20 µg/ml E-64, and 2 mM benzamidine) and 1 mM TCEP, and sonicated on ice. Lysed material was spun at 48,000 $g$ at 4°C for 30 min. The supernatant was incubated with TALON beads (Takara Clontech) for 1 h at 4°C. We extensively washed the beads with lysis buffer, followed by elution of bound proteins with elution buffer (50 mM sodium phosphate, pH 7.5, 500 mM NaCl, 250 mM imidazole, 10% glycerol, 1 mM TCEP). Recombinant 6HIS-KKT10 (pBA234: [Ishii and Akiyoshi, 2020]), 6HIS-KKT8/9/11/12 (pBA457: [Ishii and Akiyoshi, 2020]), and 6HIS-KKT16/17/18 (pBA202: [Tromer et al., 2021]) were expressed in Rosetta 2(DE3)pLys *E. coli* cells (Novagen) and purified using the same protocol. pACEBac plasmids (Bieniossek et al., 2009) containing 3Flag-KKT3 (pBA315), 3Flag-KKT2 (pBA314), KKT3 (pBA882), 3Flag-KKT6/1 (pBA819), SNAP-6HIS-3FLAG-KKT4 (pBA925), 3FLAG-KKT7 (pBA1531), 3Flag-KKT14 (pBA334), 3Flag-Aurora B$^{AUK1}$/INCENP$^{CPC1}$ (pBA1084), and 3Flag-Aurora B$^{AUK1\ K58R}$/INCENP$^{CPC1}$ (pBA2396) were transformed into DH10EmBacY cells to make bacmids, which were purified and used to transfect Sf9 cells using Cellfectin II transfection reagent (Thermo Fisher Scientific). Sf9 cells were grown in Sf-900 II SFM media (Thermo Fisher Scientific). Baculovirus was amplified through three rounds of amplification. Recombinant proteins were expressed and purified from Sf9 cells using a protocol described previously (Llauró et al., 2018). Protein concentration was determined by protein assay (Bio-Rad).

## In vitro kinase assay

Recombinant kinetochore proteins mixed with active or kinase-dead 3Flag-Aurora B$^{AUK1}$/INCENP$^{CPC1}$ complexes in kinase buffer (50 mM Tris-HCl pH 7.4, 1 mM DTT, 25 mM β-glycerophosphate, 5 mM MgCl$_2$, 5 µCi [$^{32}$P]ATP, and 10 µM ATP) were incubated at 30°C for 30 min. The reaction was stopped by the addition of the LDS sample buffer (Thermo Fisher Scientific). The samples were run on an SDS-PAGE gel, which was stained with Coomassie Brilliant Blue R-250 (Bio-Rad) and subsequently dried and used for autoradiography using a Phosphorimager Screen. The signal was detected by an FLA 7000 scanner (GE Healthcare). The $^{32}$P signal intensity for each protein was quantified in ImageJ/FIJI and normalized to the total protein amount (estimated from measuring the intensity of Coomassie-stained bands). To correct for non-Aurora B$^{AUK1}$-dependent phosphorylation, the normalized intensities from the kinase-dead controls were subtracted from these values. Normalized and corrected signal intensities are presented relative to Aurora B$^{AUK1}$ auto-phosphorylation in Fig. 3 F.

To identify AUK1-dependent phosphorylation sites on KKT14 NTR, [$^{32}$P]ATP in the kinase assay was replaced with non-labeled ATP. Gels were stained using SimplyBlue (Invitrogen) and bands corresponding to the KKT14 NTR were cut out and subjected to in-gel digestion with trypsin. Peptides were extracted from the gel pieces by sonication for 15 min, followed by centrifugation and supernatant collection. A solution of 50:50 water:acetonitrile and 1% formic acid (2x the volume of the gel pieces) was added for a second extraction and the samples were again sonicated for 15 min, centrifuged, and the supernatant pooled with the first extract. The pooled supernatants were processed using speed vacuum centrifugation. The samples were dissolved in 10 µl of reconstitution buffer (96:4 water:acetonitrile, 1% formic acid) and analyzed by LC-MS/MS using an Orbitrap Fusion Lumos mass spectrometer (Thermo Fisher Scientific) at the proteomics core facility at EMBL Heidelberg. Peptides were identified by searching tandem mass spectrometry spectra against the *T. brucei* protein database with MaxQuant as described above. Following S/T sites were mutated to A or D/E in KKT14$^{NTR}$ PD1 and PM1 constructs, respectively: S25, T104, S107, S113, T115, S173/S174, T299/S300/S301/S302/S303, T332/T333, S347/T348, whereby those sites separated by a slash (/) could not be assigned unequivocally based on the mass spectrometry data. Raw mass spectrometry files and the custom database file used in this study have been deposited to the ProteomeXchange Consortium via the PRIDE partner repository (Perez-Riverol et al., 2019; Deutsch et al., 2023) with the dataset identifier PXD048677.

## Positional scanning peptide array analysis

AUK1 phosphorylation site specificity was analyzed using a positional scanning peptide array consisting of 198 peptide mixtures with the general sequence Y-A-x-x-x-x-x-S/T-x-x-x-x-A-G-K-K(biotin). For each mixture, eight of the "x" positions were a random mixture of 17 amino acids (all were natural amino acids except Ser, Thr, and Cys), with the remaining one fixed as one of the 20 unmodified amino acids, phosphothreonine, or phosphotyrosine. Peptides (50 µM) were arrayed in 1,536-well plates in 2 µl of library buffer (50 mM HEPES, pH 7.4, 1 mM EGTA, 0.4 mM EDTA, 5 mM MgCl$_2$, 2.5 mM β-glycerophosphate, 1 mM DTT, 0.1% Tween 20) containing 0.1 mg/ml BSA, 0.67 µM PKI, and 50 µM [γ-$^{32}$P]ATP (10 µCi/ml). Reactions were initiated by adding AUK1/CPC1 to 37.5 µg/ml, and the plates were sealed and incubated at 30°C for 2 h. Aliquots were then transferred to SAM2 biotin capture membrane (Promega), which was washed, dried,

and exposed to a phosphor screen as described (Hutti et al., 2004). Radiolabel incorporation was visualized on a Molecular Imager FX phosphorimager (Bio-Rad) and quantified using QuantityOne Software (version 11.0.5; Bio-Rad). Quantified data are the average normalized data from two independent experiments. The sequence logo for Aurora B[AUK1] was generated using the logomaker package in Python (Tareen and Kinney, 2020). The height of every letter is the ratio of its value to the median value of that position. The serine and threonine heights in position "0" were set to the ratio between their favorability. For improved readability, negative values were adjusted so that the sum does not exceed –10.

### Immunoblotting

Cells were harvested by centrifugation (800 $g$, 5 min) and washed with 1 ml PBS. The pellet was resuspended in 1× LDS sample buffer (Thermo Fisher Scientific) with 0.1 M DTT. Denaturation of proteins was performed for 5 min at 95°C. SDS-PAGE and immunoblots were performed by standard methods using a Trans-Blot Turbo Transfer System with nitrocellulose membranes (Bio-Rad) and the following antibodies: rabbit polyclonal anti-GFP (TP401, 1:5,000) and mouse monoclonal TAT1 (anti-trypanosomal-alpha-tubulin, 1:5,000, a kind gift from Keith Gull) (Woods et al., 1989). Secondary antibodies used were IRDye 680RD goat anti-mouse (926-68070; LI-COR) and IRDye 800CW goat anti-rabbit (926-32211; LI-COR). Bands were visualized on an ODYSSEY Fc Imaging System (LI-COR).

### Microtubule cosedimentation assay

Microtubule cosedimentation assays were performed as described previously (Ludzia et al., 2021). Briefly, taxol-stabilized microtubules were prepared by mixing 2.5 ml of 100 μM porcine tubulin (Cytoskeleton) resuspended in BRB80 with 1 mM GTP (Cytoskeleton), 1.25 μl BRB80, 0.5 μl of 40 mM MgCl$_2$, 0.5 μl of 10 mM GTP, and 0.25 μl DMSO, and incubated for 20 min at 37°C. 120 μl of prewarmed BRB80 containing 12.5 μM Taxol (paclitaxel; Sigma-Aldrich) was added to the sample to bring the microtubule concentration to ~2 μM. 20 μl of 6HIS-KKT4[115–343] proteins (WT, PD, or PM) in BRB80 with 100 mM KCl were mixed with 20 μl of microtubules (final, 1 μM) and incubated for 45 min at room temperature. As a control, we incubated the 6HIS-KKT4[115–343] proteins with BRB80 (with 12.5 μM Taxol). The samples were spun at 20,000 $g$ at room temperature for 10 min, and the supernatant was collected. To the tubes containing pelleted fractions, we added 40 μl of chilled BRB80 with 5 mM CaCl$_2$ and incubated on ice for 5 min to depolymerize microtubules. Following incubation, samples were boiled for 5 min before SDS-PAGE. Gels were stained with SimplyBlue Safe Stain (Invitrogen).

### Statistical analysis

Statistical tests were performed using scipy.stats module in Python. For the $t$ tests used, data distribution was assumed to be normal, but this was not formally tested. A minimum of two independent biological replicates were performed in all experiments.

### Online supplemental material

Fig. S1 shows that Aurora B[AUK1] activity is required for mitotic exit and spindle stability. Fig. S2 illustrates that Aurora B[AUK1] activity is not required for the recruitment of inner and outer kinetochore proteins. Fig. S3 shows in vitro phosphorylation of CPC and KKT4 fragments by Aurora B[AUK1]. Fig. S4 characterizes the effect of ectopic expression of KKT14[NTR] constructs. Fig. S5 reveals kinetochore–microtubule attachment defects upon expression of full-length KKT14 phospho-mutants. Table S1 lists the trypanosome cell lines, plasmids, primers, and synthetic DNA used in this study. Table S2 lists phosphorylation sites on KKT14[NTR], Aurora B[AUK1], and CPC1 detected by mass spectrometry. Table S3 shows the quantification of phosphorylation sites detected in the IP-MS analysis of GFP-KKT14[NTR] from Aurora B[AUK1-as1] cells treated with 2 μM 1NM-PP1 or 10 μM MG132 as a control for 4 h. Data S1 is a custom ImageJ macro to calculate the mean intensity of segmented kinetochore foci for each cell.

### Data availability

Data are available in the article itself and its supplementary materials. Raw data, cell lines, and plasmids generated in this study are available upon request from the corresponding author.

## Acknowledgments

We thank Sam Taylor and Dipika Mishra for comments on the manuscript. We also thank Patryk Ludzia for providing recombinant KKT proteins, Sam Dean (Warwick Medical School, University of Warwick, UK) for pPOT plasmids, and Keith Gull for advice on electron microscopy and for providing TAT1 antibodies. We thank Errin Johnson and Charlotte Melia at the Dunn School Electron Microscopy Facility for providing support, reagents, and training. We thank the Micron Advanced Bioimaging Unit at the University of Oxford, the Centre Optical Instrumentation Laboratory (COIL) at the University of Edinburgh, and the Proteomics Core Facility at the EMBL in Heidelberg, especially Mandy Rettel and Jennifer Schwarz, for their support.

D. Ballmer was supported by the Berrow Foundation and the Department of Biochemistry at the University of Oxford. B.E. Turk was supported by the National Institutes of Health grant R01 GM135331. B. Akiyoshi was supported by a Wellcome Trust Senior Research Fellowship (grant 210622/Z/18/Z), a Wellcome Discovery Award (grant 227243/Z/23/Z), and a Centre Core Grant to the Wellcome Trust Centre for Cell Biology (203149). Open Access funding provided by University of Edinburgh.

Author contributions: D. Ballmer: Conceptualization, Formal analysis, Investigation, Methodology, Project administration, Software, Validation, Visualization, Writing—original draft, Writing—review & editing, H.J. Lou: Investigation, Methodology, M. Ishii: Formal analysis, Software, B.E. Turk: Investigation, Supervision, Visualization, B. Akiyoshi: Conceptualization, Data curation, Formal analysis, Funding acquisition, Investigation, Methodology, Project administration, Resources, Supervision, Validation, Visualization, Writing—review & editing.

Disclosures: The authors declare no competing interests exist.

Submitted: 31 January 2024

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

# Supplemental material

**JCB**

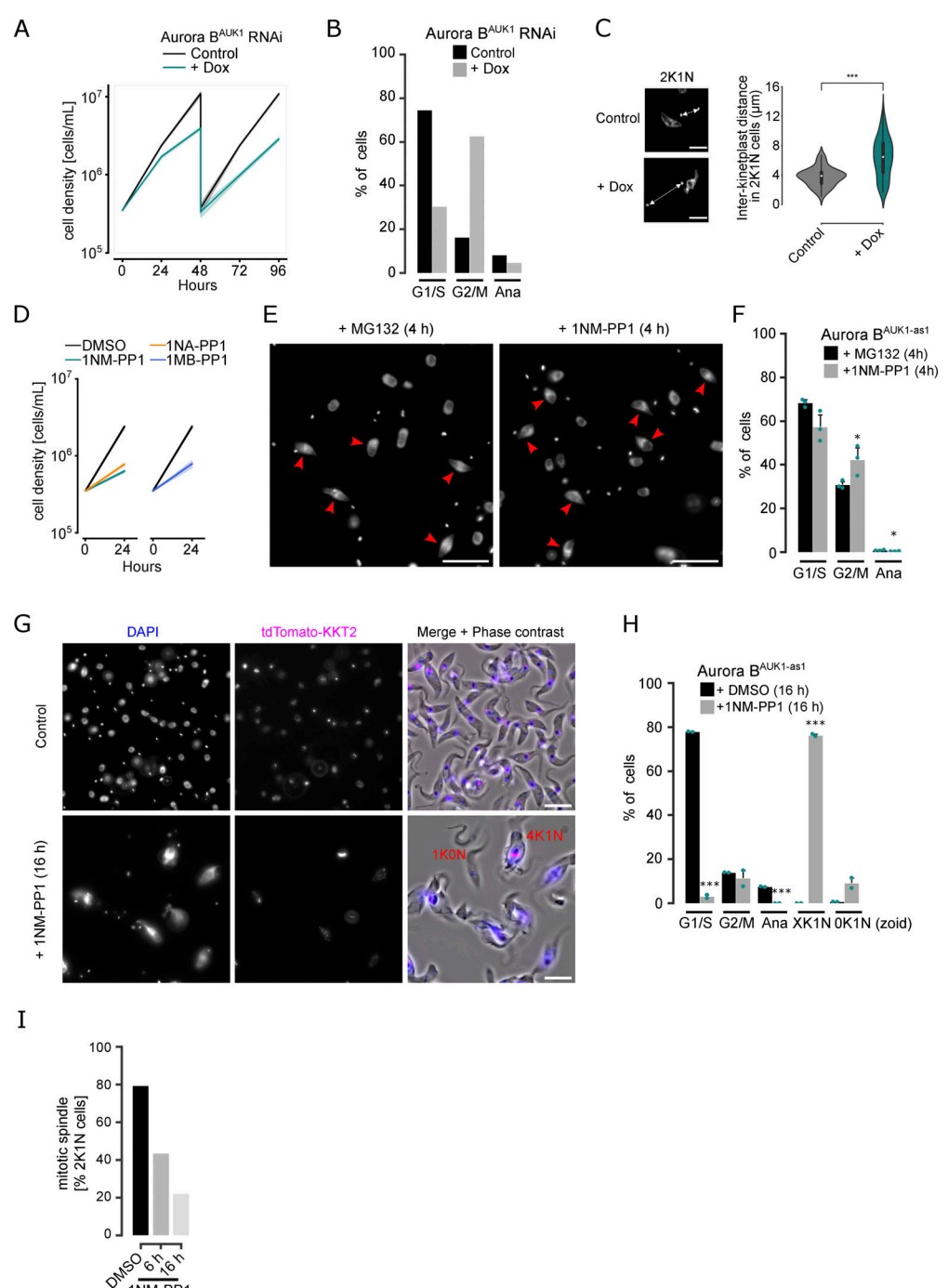

Figure S1. **Aurora B^AUK1 activity is required for mitotic exit and spindle stability. (A)** Growth curves upon RNAi-mediated knockdown of Aurora B^AUK1. RNAi was induced with 1 µg/ml doxycycline. Cell densities were measured at 0, 24, 48, 72, and 96 h. Cultures were diluted at 48 h. Data are presented as the mean ± SD of three replicates. Cell line: BAP941. **(B)** Cell cycle profile upon knockdown of Aurora B^AUK1. RNAi was induced with 1 µg/ml doxycycline and cells were fixed at 16 h. A minimum of 350 cells per condition was quantified. **(C)** Quantification of the distance between kinetoplasts in 2K1N cells upon depletion of Aurora B^AUK1 for 16 h. A minimum of 50 cells per condition was quantified. Cell line: BAP2129. *P < 0.05, **P ≤ 0.01, ***P ≤ 0.001 (Mann–Whitney U). **(D)** Growth curves upon treatment of Aurora B^AUK1-as1 cells with 2 µM 1NM-PP1, 1NA-PP1, or 1MB-PP1. Cell densities were measured at 0 and 24 h. Data are presented as the mean ± SD of three replicates. Cell line: BAP2198. **(E)** Representative fluorescence micrographs showing cell cycle distribution upon treatment of Aurora B^AUK1-as1 cells with 10 µM MG132 or 2 µM 1NM-PP1 for 4 h. DNA was stained with DAPI. Red arrowheads indicate 2K1N cells. Cell line: BAP2357. Scale bars, 10 µm. **(F)** Cell cycle profile for indicated conditions as in E. All graphs depict the means (bar) ± SD of three replicates (dots). A minimum of 500 cells per replicate were quantified. *P < 0.05, **P ≤ 0.01, ***P ≤ 0.001 (two-sided, unpaired t test). **(G)** Representative fluorescence micrographs showing an overview of Aurora B^AUK1-as1 cells treated with DMSO (control) or 2 µM 1NM-PP1 for 16 h. Examples of a 1K0N (zoid) and a 4K1N cell are labeled in red. Cell lines: BAP2924. Scale bars, 10 µm. **(H)** Cell cycle profile for indicated conditions as in G. All graphs depict the means (bar) ± SD of at least two replicates (dots). A minimum of 200 cells per condition was quantified. *P < 0.05, **P ≤ 0.01, ***P ≤ 0.001 (two-sided, unpaired t test). **(I)** Quantification of 2K1N Aurora B^AUK1-as1 cells that possess a mitotic spindle upon treatment with DMSO or 2 µM 1NM-PP1 for 6 and 16 h. A minimum of 50 cells per condition was quantified.

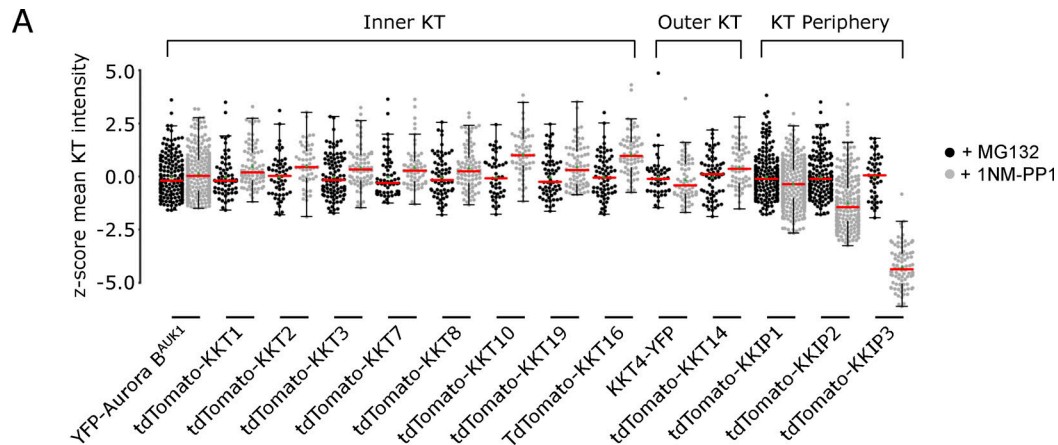

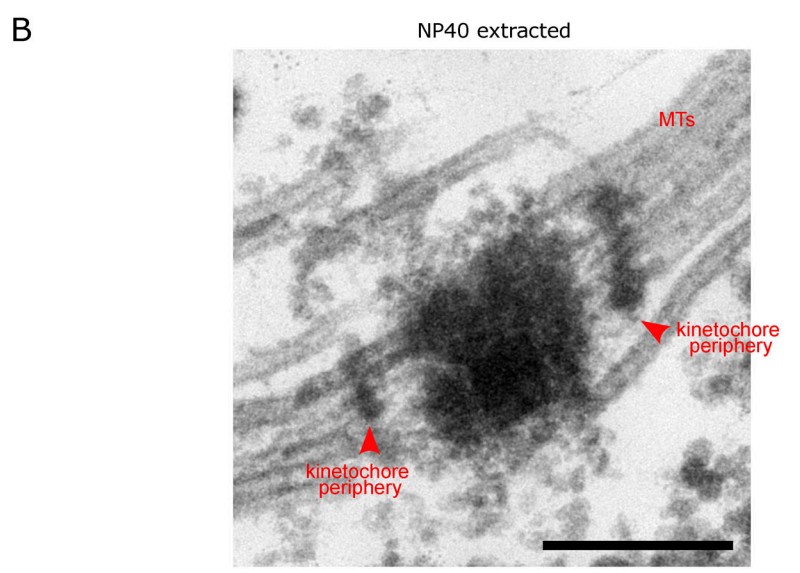

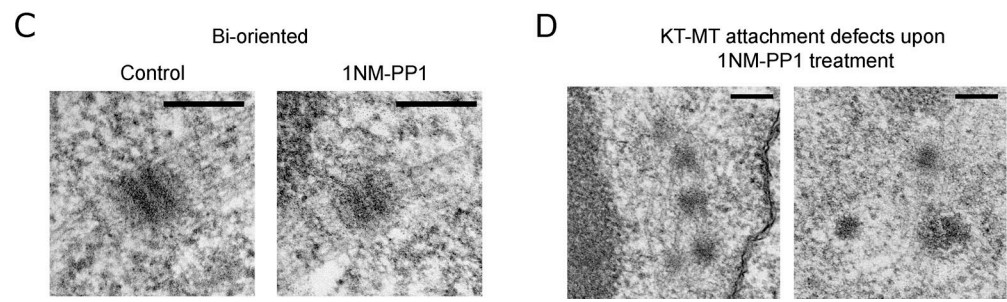

Figure S2. **Aurora B^AUK1 activity not required for recruitment of inner and outer kinetochore proteins. (A)** Quantification of mean signal intensities presented as z-scores of indicated inner and outer kinetochore and kinetochore periphery components in Aurora B^AUK1-as1 cells arrested in metaphase by treatment with 10 µM MG132 (control) or 2 µM 1NM-PP1 for 4 h. Kinetochores were segmented in ImageJ/FIJI using a custom macro (See Materials and methods). At least 45 cells (shown as dots) were analyzed per condition. The median is indicated in red. *P < 0.05, **P ≤ 0.01, ***P ≤ 0.001 (Mann–Whitney U). **(B)** Transmission electron micrograph of an NP40-extracted sample fixed with a combination of glutaraldehyde and tannic acid, which improves contrast of certain subcellular structures such as microtubules (Ogbadoyi et al., 2000; Fujiwara and Linck, 1982). Kinetochore peripheries are indicated with red arrowheads. Scale bar, 200 nm. **(C and D)** Representative transmission electron micrographs showing bioriented (C) and improperly attached (D) kinetochores in Aurora B^AUK1-as1 cells arrested in metaphase upon treatment with 10 µM MG132 or 2 µM 1NM-PP1 for 4 h as shown in Fig. 2, D and E, but without microtubules highlighted. Scale bars, 200 nm. Cell line: BAP2198.

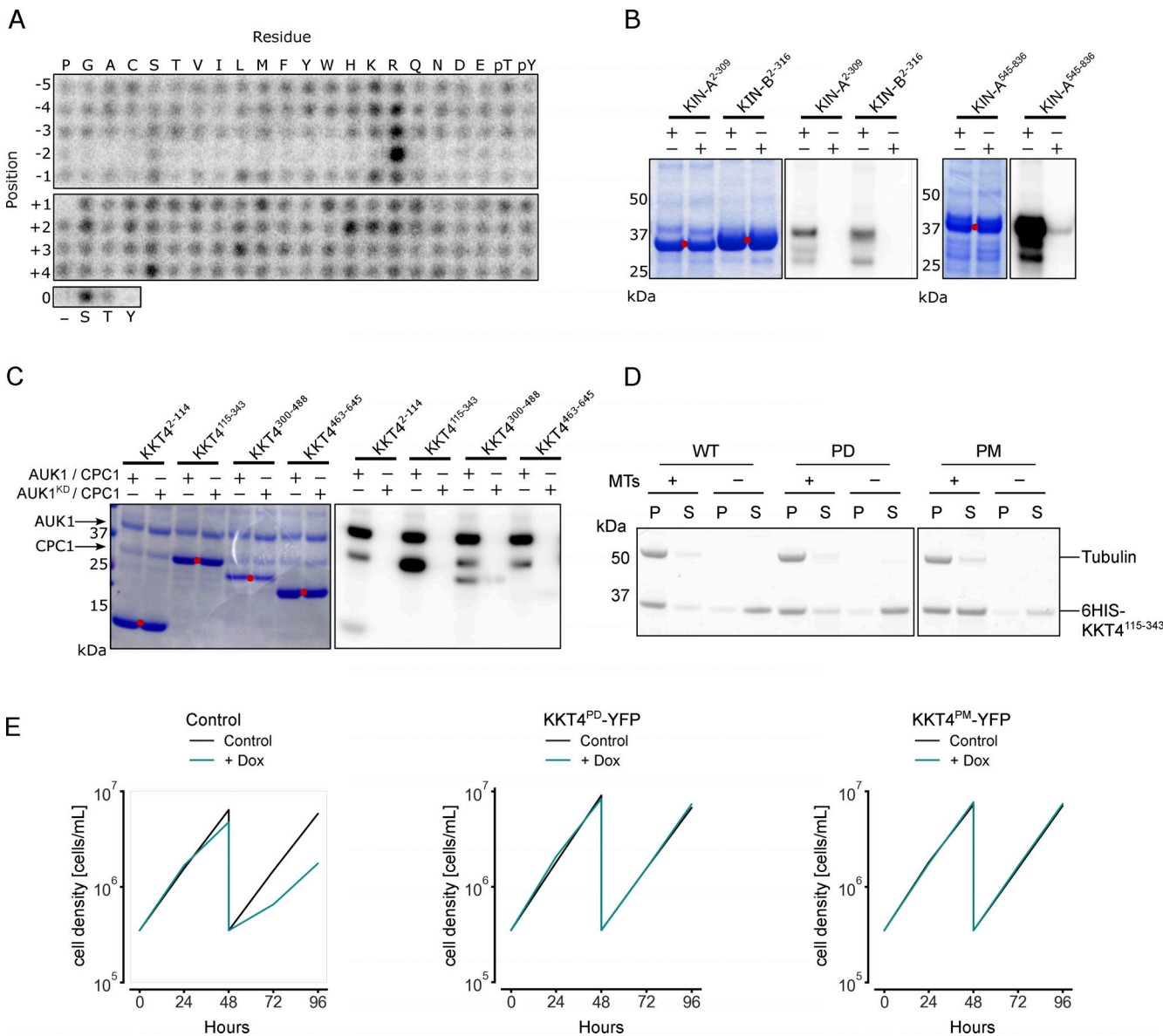

**Figure S3.  In vitro phosphorylation of CPC and KKT4 fragments by Aurora B^AUK1.** **(A)** Positional scanning peptide array image of recombinant 3FLAG-Aurora B^AUK1/INCENP^CPC1. Darker spots indicate preferred residues. **(B and C)** Aurora B^AUK1 in vitro kinase assay using the indicated recombinant CPC (B) and KKT4 (C) constructs as substrates. The left panel (input) shows the Coomassie Brilliant Blue staining. Substrates are marked with red dots. Phosphorylation was detected by autoradiography. **(D)** Microtubule cosedimentation assay with 6HIS-KKT4^115–343 WT (left), 6HIS-KKT4^115–343 PD (center), and 6HIS-KKT4^115–343 PM (right) (WT: wild-type, PD: phosphodeficient, PM: phosphomimetic). S and P correspond to supernatant and pellet fractions, respectively. Following S/T sites were mutated to A (PD) or D (PM): T234, T266, T267, T268, T316, T319, T320, S334. **(E)** Growth curves for indicated cell lines and conditions. RNAi was induced with 1 µg/ml doxycycline to deplete the untagged KKT4 allele and cultures were diluted at day 2. Cell lines: BAP2508, BAP2507, BAP2354. Source data are available for this figure: SourceData FS3.

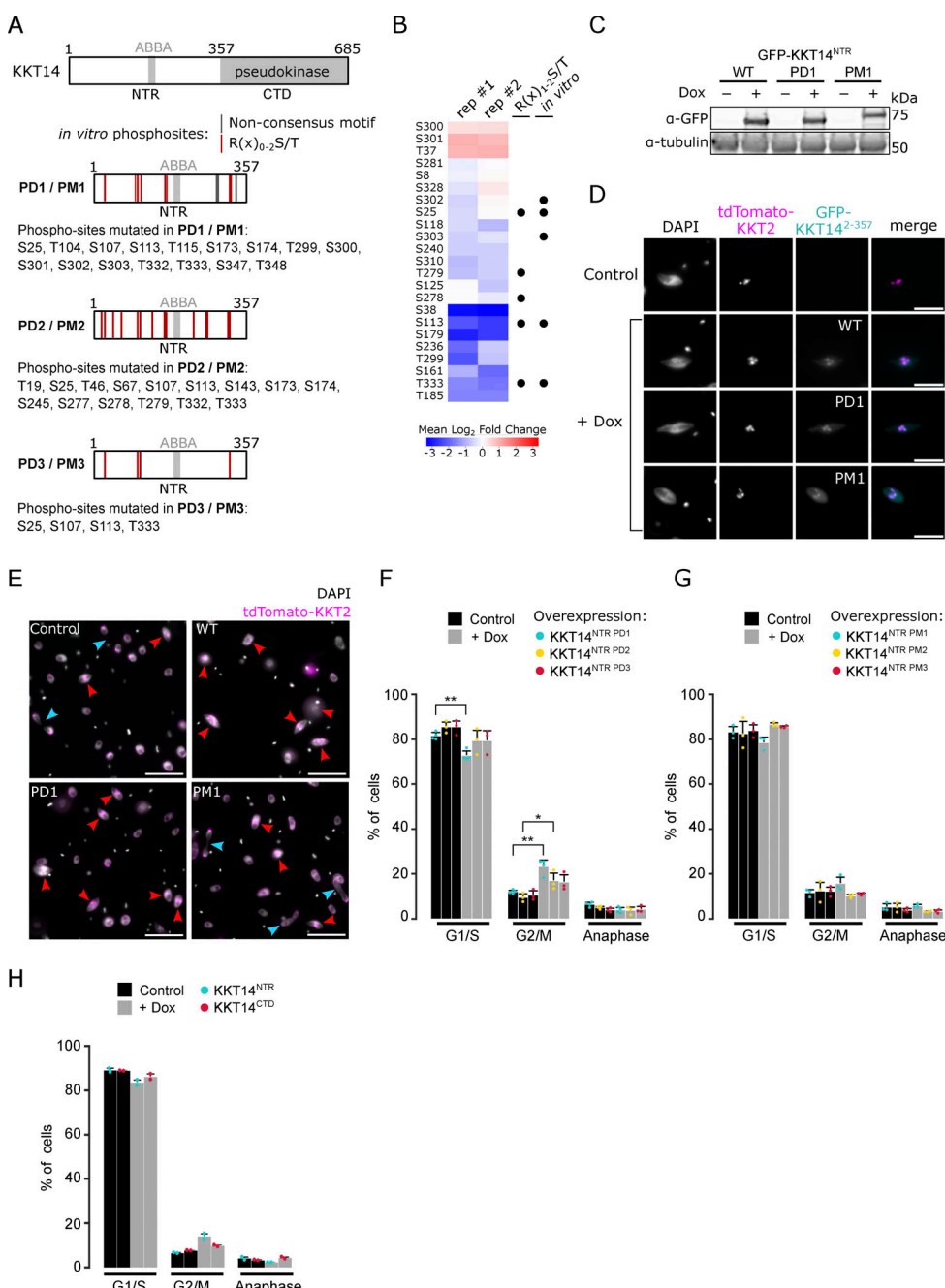

Figure S4. **Ectopic expression of KKT14^NTR constructs. (A)** Schematic representation of KKT14 showing NTR and C-terminal pseudokinase domain, ABBA motif, and S/T sites mutated to A in phosphodeficient (PD) or to D/E in phosphomimetic (PM) constructs, respectively. PD1/PM1: NTR phosphorylation sites detected by mass spectrometry are indicated by lines (gray: Non-consensus motif, red: $R(x)_{0-2}S/T$) (Table S2). PD2/PM2: All S/T sites in the NTR corresponding to the AUK1 consensus motif ($R(x)_{0-2}S/T$) mutated. PD3/PM3: Only S24, S107, S113, and T333 mutated, which are highly conserved among kinetoplastids (Ballmer et al., 2024). **(B)** Quantification of phosphosites detected in IP-MS analysis of GFP-KKT14^NTR from Aurora B^AUK1-as1 cells treated with 2 µM 1NM-PP1 or 10 µM MG132 as a control for 4 h (two replicates each, rep #1 and #2) (Table S3). The heatmap shows the $\log_2$ fold change the 1NM-PP1–treated samples compared to the control, with positive values shown in red and negative values shown in blue. Black dots indicate whether phospho-sites match the $R(x)_{1-2}S/T$ consensus motif and whether they were detected in vitro. Cell line: BAP2505. **(C)** Western blot showing protein levels of indicated GFP-KKT14^NTR constructs (WT: wild-type, PD1, PM1), induced with 1 µg/ml doxycycline for 24 h. Tubulin was used as a loading control. Cell lines: BAP2924, BAP2925, BAP2928. **(D)** Representative fluorescence micrographs showing localization of indicated GFP-KKT14^NTR constructs and tdTomato-KKT2 (kinetochore marker). Expression of fusion proteins was induced with 1 µg/ml doxycycline and cells were fixed at 24 h. Cell lines: BAP2924, BAP2925, BAP2928. Scale bars, 2 µm. **(E)** Representative fluorescence micrographs showing cell cycle distribution upon expression of indicated KKT14^NTR constructs, induced with 1 µg/ml doxycycline for 24 h. TdTomato-KKT2 marks kinetochores and DNA was stained with DAPI. Arrowheads indicate 2K1N (red) and 2K2N (light blue) cells. Cell lines: BAP2924, BAP2925, BAP2928. Scale bars, 10 µm. **(F–H)** Cell cycle profiles upon expression of indicated GFP-KKT14^NTR constructs. All graphs depict the means (bar) ± SD of at least two replicates (dots). A minimum of 300 cells per replicate were quantified. Cell lines: BAP3206, BAP3207, BAP3208, BAP3209, BAP2924, BAP2925, BAP2928, BAP2386, BAP2387. For each condition, doxycycline-treated cells were compared to the untreated isogenic cell line. *P < 0.05, **P ≤ 0.01, ***P ≤ 0.001 (two-sided, unpaired t test). Source data are available for this figure: SourceData FS4.

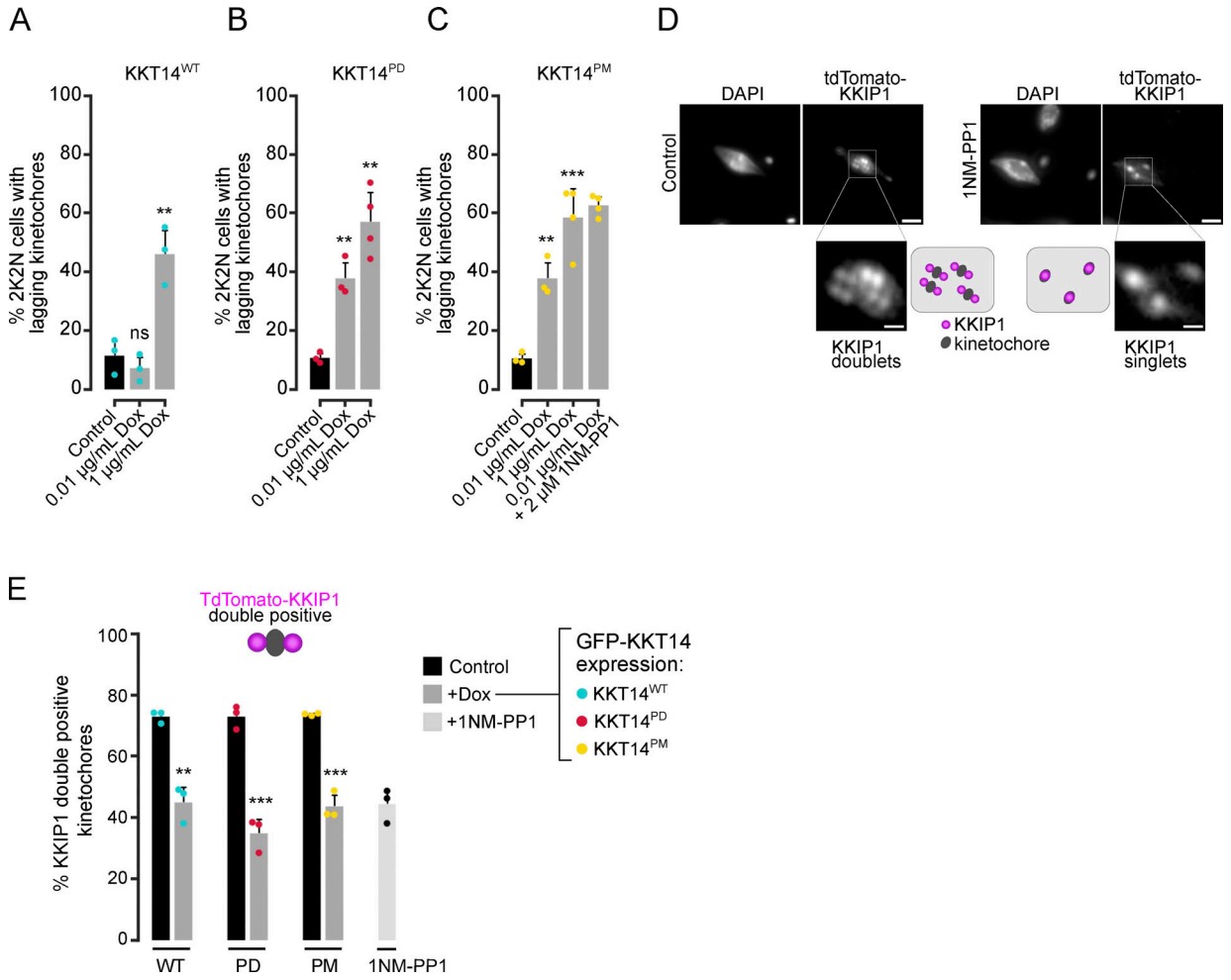

Figure S5. **Kinetochore-microtubule attachment defects upon expression of KKT14 phospho-mutants. (A–C)** Quantification of lagging kinetochores in 2K2N cells under indicated conditions. Expression of fusion proteins was induced with 0.01 or 1 µg/ml doxycycline and cells were fixed at 24 h. In C, cells were treated with 2 µM 1NM-PP1 for 4 h. All graphs depict the means (bar) ± SD of at least three replicates (dots). A minimum of 15 anaphase cells per replicate were quantified. Cell lines: BAP3228, BAP3212, BAP3213. *P < 0.05, **P ≤ 0.01, ***P ≤ 0.001 (two-sided, unpaired $t$ test). **(D)** Representative fluorescence micrographs showing the configuration of tdTomato-KKIP1 (kinetochore periphery, magenta) in Aurora B[AUK1-as1] cells arrested in metaphase upon treatment with 2 µM 1NM-PP1 or 10 µM MG132 (control) for 4 h, with a schematic guide for each configuration. Note that the kinetochore periphery component KKIP1 undergoes displacement upon biorientation and forms two foci across the inter-sister kinetochore axis ("KKIP1 doublet"). Scale bars, 2 µm. The insets show the magnification of the boxed region (scale bars, 0.5 µm). Cell line: BAP3227. **(E)** Quantification of bioriented kinetochores in Aurora B[AUK1-as1] cells under indicated conditions in metaphase. Note that the kinetochore periphery component KKIP1 undergoes displacement upon biorientation and forms two foci across the inter-sister kinetochore axis ("double positive"). Expression of fusion proteins was induced with 1 µg/ml doxycycline and cells were fixed at 24 h. Where indicated, cells were treated with 2 µM 1NM-PP1 for 4 h. For each replicate (dots), at least 80 kinetochores from ~20 metaphase cells were analyzed. Cell lines: BAP3227, BAP3218, BAP3219. *P < 0.05, **P ≤ 0.01, ***P ≤ 0.001 (two-sided, unpaired $t$ test).

Provided online are Table S1, Table S2, Table S3, and Data S1. Table S1 lists the trypanosome cell lines, plasmids, primers, and synthetic DNA used in this study. Table S2 lists phosphorylation sites on KKT14[NTR], Aurora B[AUK1], and CPC1 detected by mass spectrometry. Table S3 shows quantification of phosphorylation sites detected in the IP-MS analysis of GFP-KKT14[NTR] from Aurora B[AUK1-as1] cells treated with 2 µM 1NM-PP1 or 10 µM MG132 as a control for 4 h. Data S1 is a custom ImageJ macro to calculate the mean intensity of segmented kinetochore foci.

