## [Peer Review File · The Journal of Cell Biology]

Aurora B controls anaphase onset and error-free chromosome segregation in trypanosomes

Daniel Ballmer, Hua Lou, Midori Ishii, Benjamin Turk, and Bungo Akiyoshi

Corresponding Author(s): Bungo Akiyoshi, University of Edinburgh

Review Timeline:

Submission Date:	2024-01-31
Editorial Decision:	2024-03-06
Revision Received:	2024-06-12
Editorial Decision:	2024-07-17
Revision Received:	2024-07-23

Monitoring Editor: Arshad Desai

Scientific Editor: Dan Simon

Transaction Report:

DOI: <https://doi.org/10.1083/jcb.202401169>

March 6, 2024

Re: JCB manuscript #202401169

Dr. Bungo Akiyoshi
University of Edinburgh
Max Born Crescent
Edinburgh EH9 3BF
United Kingdom

Dear Dr. Akiyoshi,

Thank you for submitting your manuscript entitled "An unconventional regulatory circuitry involving Aurora B controls anaphase onset and error-free chromosome segregation in trypanosomes." Your manuscript has been assessed by expert reviewers, whose comments are appended below. Although the reviewers express potential interest in this work, significant concerns unfortunately preclude publication of the current version of the manuscript in JCB. Based on their feedback, we are willing to consider a revised version that will be subject to re-assessment by the reviewers.

As you will see, the reviewers all appreciate your analysis of Aurora B kinase in kinetoplastids and the surprising finding that it is essential for the metaphase-anaphase transition. However, they raise a number of concerns that will require additional experimental effort to address. To help guide your revision, we have highlighted key points below:

- 1) As indicated by Reviewer 3, the analysis of KKT14 phosphorylation needs to be improved to reach a robust conclusion. Please note the specific feedback from the reviewer on this important point.
- 2) We recommend focusing on reviewer comments targeted at strengthening the major finding linking control of metaphase-anaphase transition to Aurora B phosphorylation of KKT14. While a detailed mechanism is not expected, whether CDC20 binding by KKT14 is relevant or not would certainly help constrain mechanistic models and may be feasible to test given the tools you already have at hand.
- 3) While two reviewers highlight statement of data not shown on KKT4 (which is not permissible by the journal), in our view we do not believe analysis of KKT4 is necessary for this effort. However, you are welcome to include analysis of KKT4 if you believe it strengthens the manuscript (specifically the kinetochore-biorientation defects). You should, however, address Reviewer 1's point on whether KKT14 mutants tested exhibit biorientation defects.
- 4) As Reviewer 1 suggested, the ectopic tethering experiment would benefit from a catalytically inactive Aurora B control. This would help address whether the effects observed relate to target phosphorylation. Depending on the technical effort involved, we highly recommend such a control but will also be open to text changes acknowledging that tethering may have effects independently of target phosphorylation.
- 5) Reviewer 2 pointed out the need for improved images and other reviewer comments highlight need for text and figure changes.

Please let us know if you are able to address the major issues outlined above and wish to submit a revised manuscript to JCB. Note that a substantial amount of additional experimental data likely would be needed to satisfactorily address the concerns of the reviewers. The typical timeframe for revisions is three to four months. We at JCB realize that the lingering effects of the COVID-19 pandemic may still be impacting some aspects of your work, including the acquisition of equipment and reagents. Therefore, if you anticipate any difficulties in meeting this aforementioned revision time limit, please contact us and we can work with you to find an appropriate time frame for resubmission. Please note that papers are generally considered through only one revision cycle, so any revised manuscript will likely be either accepted or rejected.

If you choose to revise and resubmit your manuscript, please also attend to the following editorial points. Please direct any editorial questions to the journal office.

GENERAL GUIDELINES:

Text limits: Character count is < 40,000, not including spaces. Count includes title page, abstract, introduction, results, discussion, and acknowledgments. Count does not include materials and methods, figure legends, references, tables, or supplemental legends.

Figures: Your manuscript may have up to 10 main text figures. To avoid delays in production, figures must be prepared according to the policies outlined in our Instructions to Authors, under Data Presentation,

<https://jcb.rupress.org/site/misc/ifora.xhtml>. All figures in accepted manuscripts will be screened prior to publication.

IMPORTANT: It is JCB policy that if requested, original data images must be made available. Failure to provide original images upon request will result in unavoidable delays in publication. Please ensure that you have access to all original microscopy and blot data images before submitting your revision.

Supplemental information: There are strict limits on the allowable amount of supplemental data. Your manuscript may have up to 5 supplemental figures. Up to 10 supplemental videos or flash animations are allowed. A summary of all supplemental material should appear at the end of the Materials and methods section.

Please note that JCB now requires authors to submit Source Data used to generate figures containing gels and Western blots with all revised manuscripts. This Source Data consists of fully uncropped and unprocessed images for each gel/blot displayed in the main and supplemental figures. Since your paper includes cropped gel and/or blot images, please be sure to provide one Source Data file for each figure that contains gels and/or blots along with your revised manuscript files. File names for Source Data figures should be alphanumeric without any spaces or special characters (i.e., SourceDataF#, where F# refers to the associated main figure number or SourceDataFS# for those associated with Supplementary figures). The lanes of the gels/blots should be labeled as they are in the associated figure, the place where cropping was applied should be marked (with a box), and molecular weight/size standards should be labeled wherever possible.

If you choose to resubmit, please include a cover letter addressing the reviewers' comments point by point. Please also highlight all changes in the text of the manuscript.

Regardless of how you choose to proceed, we hope that the comments below will prove constructive as your work progresses. We would be happy to discuss them further once you've had a chance to consider the points raised. You can contact the journal office with any questions at cellbio@rockefeller.edu.

Thank you for thinking of JCB as an appropriate place to publish your work.

Sincerely,

Arshad Desai, PhD
Monitoring Editor
Journal of Cell Biology

Dan Simon, PhD
Scientific Editor
Journal of Cell Biology

Reviewer #1 (Comments to the Authors (Required)):

In this work, the authors studied the roles of Aurora B kinase in the kinetoplastid parasite, *T. brucei*. They developed analogue-sensitive Aurora B and demonstrated that Aurora B kinase activity is required for kinetochore biorientation and metaphase-to-anaphase transition. Aurora B phosphorylates several kinetochore proteins and it is suggested that phosphorylation of KKT14 is important for metaphase-to-anaphase transition. Moreover, they showed that targeting the catalytic module of the CPC (including Aurora B and INCENP) to the outer kinetochore promotes the metaphase-to-anaphase transition but does not prevent chromosome missegregation.

The roles of Aurora B have been studied well in human cells and various model organisms, but little is known about them in kinetoplastids. The current study suggests the evolutionarily conserved and non-conserved roles of Aurora B in *T. brucei*, which I find very interesting. In principle, I support the publication of this paper in JCB. However, additional experiments would improve the paper, as follows:

Major points:

1) Fig 4F-H: The effect of KKT14 phosphorylation by Aurora B has been analysed after KKT14 wild-type and PD/PM mutants were overexpressed. I wonder whether they can do a 'cleaner' experiment with the (almost) normal expression level of KKT14 if

feasible. It will be ideal if PD, but not WT, (partially) phenocopies Aurora B inhibition in the 'cleaner' experiment.

2) Fig 4: If feasible, they can address whether KKT14 phosphorylation is important for kinetochore biorientation in addition to the metaphase-to-anaphase transition. Other kinetochore proteins may be important substrates of Aurora B for biorientation. But it is possible that KKT14 phosphorylation is also involved in ensuring biorientation.

3) Fig 5I: They interpret that lagging kinetochores in anaphase occur due to defects in biorientation defects. However, lagging kinetochores in anaphase could also happen due to other defects. If feasible, they should check defects in biorientation in metaphase, for example, using the assay shown in Fig 2A-C.

4) Fig 5I: It will be interesting whether the increase of lagging kinetochores with tethering to the outer kinetochore requires the Aurora B kinase activity. If feasible, the authors can tether catalytically inactive Aurora B to the outer kinetochore to study the outcome.

Some of these experiments may not be technically feasible and I would understand it if they cannot be carried out due to technical difficulty.

Minor points:

1) Fig 4H: It is not clear which experimental groups are compared in the unpaired t-test. This should be clarified.

2) Line 279-281: A recent paper (Li et al *Curr Biol* 33, 4557-69, 2023) tested the spatial separation model more directly in budding yeast. This paper can be cited if the authors agree with its relevance.

Reviewer #2 (Comments to the Authors (Required)):

In this work, Ballmer and colleagues have studied the function of Aurora kinase in mitotic chromosome separation in the parasite *Trypanosoma brucei*, which represents a highly divergent organism compared to the model systems that have been employed to study this process. *T. brucei* appears to lack a conventional spindle assembly checkpoint and has uniquely configured kinetochores, which suggests that mitotic regulation may employ distinct mechanisms in these organisms. Aurora kinase and components of the chromosomal passenger complex are conserved and may have taken on additional functions. The authors employ an analog-sensitive allele of AUK coupled with fluorescent labeling of kinetoplastid-specific kinetochore proteins to show that loss of AUK activity decreases tension across the kinetochore, which can also cause kinetochore-microtubule attachment defects. Biochemistry using recombinant AUK-CPC1 subunit active complex and recombinant KKTs showed that KKT14, a potentially divergent Bub1R homolog, is strongly phosphorylated by AUK. Mutation of putative AUK sites on KKT14 to phosphomimetic AAs allows *T. brucei* to compensate for AUK inhibition and continue on to anaphase, which is a striking result. Building on this, the authors propose that tethering the CPC1-AUK1 active complex to the inner kinetochore using a GFP nanobody is sufficient to allow anaphase progression, while tethering to the outer kinetochore allows progress but with lagging chromosomes.

Overall, this work is an impressive mix of advanced genetics/cell biology and biochemistry that provides novel insights into the function of a divergent CPC and kinetochore structure. To my knowledge, no one has used analog sensitive methods and RNAi simultaneously in *T. brucei*, which makes this work a significant technical advance in the field. I have three main concerns about the manuscript in its current form:

1. I do not think that the localization and functional data for the recruitment of the in Figure 5D-I is compelling. The tdTomato channel images do not convincingly show that the nanobody-YFP interaction is recruiting the AUK-CPC1-nanobody fusion to the kinetochore. The lookup tables on this channel look like they were set very very low. I think the authors need to provide more compelling imaging to show that the nanobody is recruiting the complex to the kinetochore via the KKT-YFP fusion. The functional result of being able to reach anaphase is compelling, but the imaging needs to be conclusive. Also, why are there still elevated levels of cells in metaphase in all the cases?

2. The authors refer to two key pieces of data in their discussion- that a phosphomimetic KKT4 mutant has reduced affinity for MTs and that KKT4 phosphorylation outside of the MT-binding domain stabilizes the KKT4-MT interaction. They list these as "data not shown" or "unpublished observations". As these results would provide important mechanistic insight, I think this data must be included in the paper so that it can be evaluated, or mention of it should be omitted from the discussion.

3. Can the authors explain why overexpression of the phosphomimetic KKT14 allows a bypass of AUK1 activity without prematurely triggering anaphase progression? Does overexpression of any of these KKT14 constructs have an effect on cell division?

More minor points:

Overall, I found the microscopy images in the figures to be very small, even as larger figures for review. I think that journal-page sized figures will be hard to interpret. Figure 1F was reasonable to me, and the use of magnified boxes such as in 2A also helped significantly with interpreting the data.

Considering the number of different cell lines generated in this work, including complex analog-sensitive and RNAi lines, there is a significant lack of detail about how they were assembled. Is the AS line a dual allele mutant, or was one allele knocked out? This information and more general details about cell line construction should be included as a supplemental.

I think versions of the EM in 2D-E without the artificial enhancement should be included as supplemental figures.

Fig 1A and Fig S1A- It would be preferable to have clear markers that show when cell densities were measured. It's difficult to tell in these graphs as currently depicted.

Figure 3D: There is a fair amount of phosphorylation in the KD lanes for the KKT3, 2/3, and 1/3/6 complex. Clearly some other kinase activities are in the CPC1/AUK1 prep, which is concerning. This should be mentioned in the main text, along with the normalization strategy.

Figure 3F- Does the quantitation account for the large differences in size of the different substrates? The experimental says that the signal was normalized for "amount", but is that just weight or Mol? Because with substrates that are this different in size the molar amount in the substrate can be quite different, which would affect the expected amount of signal during the kinase assay.

Figure S3: The MBP-CPC2 kinase assay experiments are hard to interpret due to the presence of MBP fusion as a potential substrate. I would cleave the MBP or omit this data.

Line 307-309: confusingly phrased.

Figure 5H: What's the "Negative" control?

Reviewer #3 (Comments to the Authors (Required)):

In this manuscript from the Akiyoshi lab by Ballmer and colleagues, the authors essentially explore the role of the Aurora B homologue (AUK1) in kinetoplastid mitosis using the trypanosome model *T. brucei*. The authors demonstrate that compared to the other common model organisms used to study the CPC and kinetochores, *T. brucei* AUK1 has both conserved and divergent functions. In particular, using an analogue sensitive allele of AUK1, the authors demonstrate a role for the kinase activity in mediating the metaphase-anaphase transition. After screening a panel of substrates, the authors find strong phosphorylation of the potential BUB1 orthologue KKT14 by AUK1. Phospho-mimetic mutants of the AUK1 sites on KKT14 promote Met-ana transition, whereas the non-phosphorylatable versions did not, rather mimicking AUK1 inhibition. In addition, the authors demonstrated that AUK1 activity supports proper spindle formation and biorientation and finally that the position of AUK1 at the outer-kinetochore destabilizes attachments resulting in massive errors in chromosome segregation but that inner or outer kinetochore AUK1 positioning supports met-ana transition.

Overall, I feel that there are several interesting discoveries outlined in this manuscript, but that their impact is limited by lack of more in-depth exploration which unfortunately has made for a rather disjointed story of two phenotypes and in this reviewer's opinion, not suitable as an article in JCB.

Major points

My main concern is that none of the discoveries are thoroughly followed up in any depth. Below are a few examples:

1. In the analysis of KKT14 phospho-sites, the authors are unclear as to which sites are mutated. In the methods section, they list a number of sites some separated by comma others by a slash (/) meaning that the site could not be assigned, presumably. In the figure legend, there's 2 fewer sites mentioned, so why were the 2 additional sites mutated if there was no evidence they were phosphorylated? The authors should clarify their nomenclature and the exact sites tested.

2. Overall the authors need to narrow down their phospho-site analysis of KKT14 to the most relevant sites or provide convincing evidence that ALL the sites mutated are functionally relevant.

The authors presumably mutated all sites in KKT14 that they identified as AUK1 targets. This is a brute force method that is somewhat crude. Given the number of sites mutated, how can the authors rule out that what they seeing is a result of disrupting the global structure of the N-terminus of KKT14 rather than the phosphor-sites? The authors need to prioritize sites based on some criteria (eg. Conservation amongst trypan sp. and/or adherence to the AUK1 consensus motif). At minimum, the authors need to generate the individual and triple phosphor-mimetic/non-phosphorylatable mutants covering the three most promising AUK1 sites they mentioned and test the function of these collectively or individually before moving on to more encompassing mutagenesis experiments.

3. The effect of KKT14 depletion in Fig 4B is only partial suggesting that other AUK1 targets may be more important than KKT14, despite KKT14 being an excellent substrate. The authors should discuss this and other alternative explanations for the

data. Are there any other candidates amongst their identified targets that could rescue the AUK1 inhibition phenotype when depleted? This lab has a unique access to resources to test this idea and they should push the analysis further to really explore the role of AUK1 in Met-ana transition.

4. Where are the KTT14 phospho-sites relative to the putative ABBA motif, and is this ABBA motif important for the Met-Ana transition? At the very least, the authors need to check the effect of the ABBA mutant on Met-ana transition and whether there is cooperativity with the phosphorylation sites on this process.

5. In fig 4h, the authors need to include an RNAi only control without the rescue of the KKT14.

6. The effect of AUK1 kinase activity on biorientation shown in fig 1 and the last figure is exciting, but there are no follow-up experiments. The authors speculate and report some unpublished observations in the discussion on the potential role of KKT4 in this but provide no experimental data. At the very least to make a more complete story the authors need to perform the same type of phospho-mutants analysis with KKT4 as they have with KKT14 . In particular they should demonstrate the effect of the mutations in the biorientation assay and also on the microtubule binding.

Minor points:

1. Figure 4 D is too small and the position of the phosphosites should be shown relative to other known motifs such as the ABBA.

2. Fig S1K is an important observation the deserves to be part of a main figure

3. There is no evidence that the RNAi used throughout the manuscript is functional. The authors should show WB, IF or even pPCR data that it is functioning as expected. If these were demonstrated in a previous manuscript, this needs to be referenced.

Editor's summary:

Thank you for submitting your manuscript entitled "An unconventional regulatory circuitry involving Aurora B controls anaphase onset and error-free chromosome segregation in trypanosomes." Your manuscript has been assessed by expert reviewers, whose comments are appended below. Although the reviewers express potential interest in this work, significant concerns unfortunately preclude publication of the current version of the manuscript in JCB. Based on their feedback, we are willing to consider a revised version that will be subject to re-assessment by the reviewers.

As you will see, the reviewers all appreciate your analysis of Aurora B kinase in kinetoplastids and the surprising finding that it is essential for the metaphase-anaphase transition. However, they raise a number of concerns that will require additional experimental effort to address. To help guide your revision, we have highlighted key points below:

We thank the Editor and Reviewers for their positive assessment of our work and the constructive feedback to further improve our manuscript. We have addressed their comments in our response to Reviewers' comments below. Changes in the main text are highlighted in red in the revised manuscript. Our brief response to the comments in the Editor's summary is as follows:

1) As indicated by Reviewer 3, the analysis of KKT14 phosphorylation needs to be improved to reach a robust conclusion. Please note the specific feedback from the reviewer on this important point.

We have significantly extended our analysis of the Aurora B-dependent phosphorylation of KKT14 by creating and testing >10 new cell lines. These data are now presented in a new main (Fig. 5) and two supplementary figures (Fig. S4, S5).

Briefly, we created additional KKT14^{NTR} PD/PM constructs: PD2/PM2, in which we targeted all Aurora B consensus sites (including those that were not found by mass spectrometry), and PD3/PM3, in which we targeted four highly conserved R(x)₀₋₂S/T sites (S25, S107, S113, T333), three of which correspond to those mentioned by the reviewer (e.g. which showed reduced phosphorylation upon Aurora B^{AUK1} inhibition in our IP-MS experiment). Overexpression of KKT14^{NTR} PD3 caused a similar delay in the metaphase-anaphase transition as PD1 and PD2, whereas PM3 did not, suggesting that S25, S107, S113, T333 are indeed important sites.

In addition to our KKT14^{NTR} ectopic expression analysis, we generated new cell lines allowing inducible expression of full-length KKT14 PD/PM constructs. Inspired by the Reviewer's comments, we followed up the role of KKT14 phosphorylation in more detail by assessing cell proliferation, cell cycle progression and lagging kinetochores upon induction of KKT14 PD/PM with 0.01 (~physiological KKT14 levels) or 1 µg/ml doxycycline (overexpression), with or without Aurora B^{AUK1} inhibition. We found that overexpression of KKT14^{PD} and KKT14^{PM} and to a lesser extent KKT14^{WT} caused growth defect and lagging kinetochores in anaphase. Overall, we found that the effect of KKT14^{PD} and KKT14^{PM} were more severe compared to KKT14^{WT}, with growth defects and lagging kinetochores observed even when expression was induced with 0.01 µg/ml doxycycline. Importantly, expression of full-length KKT14^{PD} but not KKT14^{WT} caused a prominent delay in the metaphase-anaphase transition. Furthermore, expression of KKT14^{PM} rescued the cell cycle arrest caused by Aurora B^{AUK1} inhibition to a similar extent as observed upon KKT14 depletion. Taken together, these new results support our conclusion that phosphorylation of KKT14 by Aurora B promotes the metaphase-to-anaphase transition in trypanosomes.

2) We recommend focusing on reviewer comments targeted at strengthening the major finding linking control of metaphase-anaphase transition to Aurora B phosphorylation of KKT14. While a detailed mechanism is not expected, whether CDC20 binding by KKT14 is relevant or not would certainly help constrain mechanistic models and may be feasible to test given the tools you already have at hand.

We agree that this is a very important point. Unfortunately, we have been unable to purify a recombinant *T. brucei* CDC20 protein to test whether it binds KKT14 in an ABBA-dependent manner. When we overexpressed KKT14^{NTR} with the ABBA motif mutated to 'AAAAAA' (ABBA 6A), we found that the mutant had the same effect on cell cycle progression as the wild-type control, suggesting that the delay is not dependent on the functional ABBA motif at least in this assay (although we cannot exclude the possibility that the mutant KKT14 protein still binds CDC20 via different motifs). We failed to derive an ABBA 6A line targeting the endogenous locus, which could indicate that this motif is essential for cell division. We will need to establish additional tools (both in vivo and in vitro) to study the functional importance of this motif. Given that our preliminary results were negative, and we currently do not have any robust data connecting Aurora B^{AUK1} activity to CDC20-binding through the KKT14 ABBA motif, we did not include any experiments characterizing the function of the ABBA motif in the manuscript.

3) While two reviewers highlight statement of data not shown on KKT4 (which is not

permissible by the journal), in our view we do not believe analysis of KKT4 is necessary for this effort. However, you are welcome to include analysis of KKT4 if you believe it strengthens the manuscript (specifically the kinetochore-biorientation defects). You should, however, address Reviewer 1's point on whether KKT14 mutants tested exhibit biorientation defects.

In the revised manuscript, we have included the data in Fig. S3. In brief, our microtubule co-sedimentation assays using recombinant phosphodeficient and phosphomimetic 6HIS-KKT4¹¹⁵⁻³⁴³ constructs show that the phosphomimetic mutant has reduced microtubule binding activity *in vitro*. To examine the function of these mutants in trypanosome cells, we performed rescue experiments and found that both mutants can support cell growth, suggesting that Aurora B^{AUK1}-dependent phosphorylation of KKT4 does not significantly affect its microtubule-binding activity *in vivo* or that loss of its MT-binding activity can be compensated through other, yet to be identified, MT-binding proteins at the trypanosome kinetochore.

To address Reviewer 1's point, we tested the effect of KKT14 PD/PM expression on kinetochore-microtubule attachments and chromosome segregation using the kinetochore periphery protein KKIP1 as bi-orientation marker. As mentioned above, overexpression of KKT14^{PD} and KKT14^{PM} and to a lesser extent KKT14^{WT} caused growth defect and lagging kinetochores in anaphase. Overall, we found that the effect of KKT14^{PD} and KKT14^{PM} were more severe compared to KKT14^{WT}, with growth defects and lagging kinetochores observed even when expression was induced with 0.01 µg/ml doxycycline. These results show the importance of phospho-regulation for a proper control of KKT14. We also found bi-orientation defects in metaphase cells upon expression of KKT14 proteins, suggesting that the lagging kinetochores observed in anaphase indeed stem from a failure to form stable bioriented attachments. These results show that phosphorylation of KKT14 plays important roles in both bi-orientation and cell cycle regulation.

4) As Reviewer 1 suggested, the ectopic tethering experiment would benefit from a catalytically inactive Aurora B control. This would help address whether the effects observed relate to target phosphorylation. Depending on the technical effort involved, we highly recommend such a control but will also be open to text changes acknowledging that tethering may have effects independently of target phosphorylation.

It is indeed possible that tethering of the CPC^{cat}-tdTomato-VhhGFP4 fusion protein to the outer kinetochore causes defects in KT-MT attachments in an Aurora B^{AUK1} activity-

independent manner. However, the fact that a robust increase in lagging kinetochores upon targeting CPC^{cat}-tdTomato-VhhGFP4 to the outer kinetochore is only observed when we simultaneously inhibit the endogenous Aurora B^{AUK1} kinase, suggests that this phenotype is caused by shifting the bulk of Aurora B^{AUK1} activity away from the inner kinetochore to the outer kinetochore. Due to the time-consuming nature for testing the catalytically inactive version of Aurora B^{AUK1} in this assay, we have changed the texts acknowledging this possible caveat of our tethering assays, as suggested by the Editor.

5) Reviewer 2 pointed out the need for improved images and other reviewer comments highlight need for text and figure changes.

We have enlarged all the fluorescence micrographs in the manuscript. Note that for some images (e.g. Figs. 1D, 4A and 5G), our goal is to provide a representative overview of the cell cycle distribution rather than to highlight the subcellular localization of particular proteins. In those cases, we aimed at showing ~15-20 cells/nuclei, which means that some cells/nuclei may nevertheless still appear rather small. However, to guide the reader through these images, we use arrowheads to mark cells in metaphase and anaphase.

Regarding the nanobody-based tethering assays (Fig. 6 in the revised manuscript), we have improved the contrast for the tdTomato channel and have enlarged all the images. We hope that the nanobody-dependent recruitment of AUK1-CPC1 to YFP-tagged kinetochore proteins is clearer now. Additional changes to the text/figures as suggested by the reviewers include: Moving panel S1K to main figure 1, improving the schematics of KKT14 to include the position of the ABBA motif and further highlight the exact S/T sites targeted in the phosphomutants, and adding more information on how the data from the kinase assays were quantified in the legend.

Reviewer #1 (Comments to the Authors (Required)):

In this work, the authors studied the roles of Aurora B kinase in the kinetoplastid parasite, *T. brucei*. They developed analogue-sensitive Aurora B and demonstrated that Aurora B kinase activity is required for kinetochore biorientation and metaphase-to-anaphase transition. Aurora B phosphorylates several kinetochore proteins and it is suggested that phosphorylation of KKT14 is important for metaphase-to-anaphase transition. Moreover, they showed that targeting the catalytic module of the CPC (including Aurora B and INCENP) to the outer kinetochore promotes the metaphase-to-anaphase transition but does not prevent chromosome missegregation.

The roles of Aurora B have been studied well in human cells and various model organisms,

but little is known about them in kinetoplastids. The current study suggests the evolutionarily conserved and non-conserved roles of Aurora B in *T. brucei*, which I find very interesting. In principle, I support the publication of this paper in JCB. However, additional experiments would improve the paper, as follows:

We thank the reviewer for their positive assessment of our work and the constructive feedback and suggestions to further improve our manuscript. We are pleased that the reviewer found our study of high quality and the potential for a publication at JCB.

Major points:

1) Fig 4F-H: The effect of KKT14 phosphorylation by Aurora B has been analysed after KKT14 wild-type and PD/PM mutants were overexpressed. I wonder whether they can do a 'cleaner' experiment with the (almost) normal expression level of KKT14 if feasible. It will be ideal if PD, but not WT, (partially) phenocopies Aurora B inhibition in the 'cleaner' experiment.

Although it would be ideal to perform a 'cleaner' rescue experiment in which RNAi-resistant KKT14 mutants are expressed from one of the endogenous alleles and the other allele expressing wild-type KKT14 is conditionally depleted by RNAi or other methods, we failed to establish such a system thus far. To address the reviewer's Major points 1 and 2, we took advantage of currently available tools to extend our analysis on the role of KKT14 phosphorylation by Aurora B^{AUK1}, which we present in a new main and two supplementary figures (Fig. 5, S4, S5). These new results support our conclusion that KKT14 phospho-deficient mutants partially phenocopy Aurora B inhibition and that phospho-mimetic mutants partially alleviate the cell cycle delay caused by Aurora B inhibition. Below, we explain the details.

We acknowledge that our previous analysis was limited to overexpression of KKT14^{NTR} constructs with PD/PM mutations corresponding to in vitro phosphosites detected by mass spectrometry. Of note, some of these sites do not conform to the Aurora B^{AUK1} consensus motif, while other potentially relevant sites may not have been detected for technical reasons. To provide a more comprehensive analysis, we generated new cell lines for inducible expression of KKT14^{NTR} PD/PM constructs, in which we mutated either all or only four specific R(x)₀₋₂S/T sites (S25, S107, S113, T333), which we found were highly conserved among kinetoplastids (Ballmer et al., 2024). In addition to our KKT14^{NTR} overexpression analysis, we also generated new cell lines allowing inducible expression of full-length KKT14 PD/PM constructs (with all R(x)₀₋₂S/T sites targeted in the NTR). As

suggested by the reviewer, we followed up the role of KKT14 phosphorylation in more detail by assessing cell proliferation, cell cycle progression and lagging kinetochores upon induction of KKT14 PD/PM with 0.01 (~physiological KKT14 levels) or 1 $\mu\text{g/ml}$ doxycycline (overexpression), in the presence/absence of Aurora B^{AUK1} inhibition. We describe our new results as follows:

To examine the importance of the CPC-dependent phosphorylation of KKT14, we ectopically expressed GFP-KKT14^{NTR} constructs containing wild-type (WT), phosphodeficient (PD; S/T to A) or phosphomimetic (PM; S/T to D/E) with 1 $\mu\text{g/ml}$ doxycycline and monitored cell cycle distribution. We first mutated the phosphorylation sites detected by mass spectrometry (referred to as PD1/PM1). These GFP-KKT14^{NTR} constructs were expressed at similar levels (Fig. S4C) and localized to kinetochores (Fig. S4D). We then created additional PD and PM constructs for which either all or only highly conserved R(x)₀₋₂S/T sites (S25, S107, S113, T333) (Ballmer et al., 2024) were targeted (referred to as PD2/PM2 and PD3/PM3, respectively) (Fig. S4A). We found that expression of WT, PD1, PD2 and to a lesser extent PD3 constructs caused an increase in 2K1N cells (Fig. S4, E, F and H), and thus partially phenocopied Aurora B^{AUK1} inhibition. In contrast, expression of phosphomimetic KKT14^{NTR} or the C-terminal pseudokinase domain did not delay anaphase onset (Fig. S4, G and H). These results suggest that expression of KKT14^{NTR} affects cell cycle progression, which is regulated by Aurora B^{AUK1}.

We next extended our analysis by expressing GFP-fusion of full-length KKT14 PD and PM constructs that mutated all R(x)₀₋₂S/T sites in the NTR with either 0.01 or 1 $\mu\text{g/ml}$ doxycycline (Fig. 5, B, C), and assessed cell growth, cell cycle progression and chromosome segregation fidelity. Expression of KKT14^{PD} and KKT14^{PM} and to a lesser extent KKT14^{WT} slowed down cell growth and caused lagging kinetochores in anaphase (Figs. 5, D–F, and S5, A–C). Overall, we found that the effect of KKT14^{PD} and KKT14^{PM} were more severe compared to KKT14^{WT}, with growth defects and lagging kinetochores observed even when expression was induced with 0.01 $\mu\text{g/ml}$ doxycycline. To test whether the failure to accurately segregate chromosomes upon KKT14 expression was preceded by a defect in the formation of stable KT-MT attachments in (pro)metaphase, we counted the number of kinetochores double positive for the kinetochore periphery protein KKIP1 in cells treated with MG132. Like KKIP2, KKIP1 undergoes tension-dependent displacement upon proper bi-orientation (Llauró et al., 2018). Similarly to Aurora B^{AUK1} inhibition, expression of KKT14 was accompanied by a significant reduction in tdTomato-KKIP1 double positive kinetochores (Fig. S5, D and E), suggesting that the lagging kinetochores observed in anaphase indeed stem from a failure to form stable KT-MT attachments and/or properly bi-orient kinetochores in (pro)metaphase.

Importantly, expression of full-length KKT14^{PD} but not KKT14^{WT} caused a prominent delay in the metaphase-anaphase transition (Fig. 5, G–I). However, when combined with Aurora B^{AUK1} inhibition, KKT14^{PD} did not further impair cell cycle progression, indicating that KKT14 operates downstream of Aurora B^{AUK1} in regulating the timing of anaphase onset. In line with this, expression of KKT14^{PM} rescued the cell cycle arrest caused by Aurora B^{AUK1} inhibition to a similar extent as observed upon KKT14 depletion (Figs. 5, G and J; 4, A and B). Many cells were able to enter an early stage of anaphase, albeit in the presence of lagging kinetochores (Figs. 5G and S5C). The absence of a full rescue upon depletion of KKT14 or expression of the phosphomimetic mutant suggests that Aurora B^{AUK1} phosphorylates additional kinetochore proteins to promote error-free entry into anaphase and/or that continued activity of Aurora B^{AUK1} at the anaphase central spindle is required for completion of nuclear division.

2) Fig 4: If feasible, they can address whether KKT14 phosphorylation is important for kinetochore biorientation in addition to the metaphase-to-anaphase transition. Other kinetochore proteins may be important substrates of Aurora B for biorientation. But it is possible that KKT14 phosphorylation is also involved in ensuring biorientation.

Thank you for this suggestion. We made new cell lines to test the effect of KKT14 PD/PM overexpression on kinetochore-microtubule attachments using the kinetochore periphery protein KKIP1 as 'bi-orientation' marker (similarly to our assay using KKIP2 in Fig. 2). In this case, we used tdTomato-KKIP1, since the corresponding cell line was straightforward to derive, whereas it would have taken additional steps to generate Aurora B^{AUK1}-as1/tdTomato-KKIP2/GFP-KKT14 (WT/PD/PM) lines due to availability of selection markers. We describe our novel experimental results as follows:

To test whether the failure to accurately segregate chromosomes upon KKT14 expression was preceded by a defect in the formation of stable KT-MT attachments in (pro)metaphase, we counted the number of kinetochores double positive for the kinetochore periphery protein KKIP1 in cells treated with MG132. Like KKIP2, KKIP1 undergoes tension-dependent displacement upon proper bi-orientation (Llauró et al., 2018). Similarly to Aurora B^{AUK1} inhibition, expression of KKT14 was accompanied by a significant reduction in tdTomato-KKIP1 double positive kinetochores (Fig. S5, D and E), suggesting that the lagging kinetochores observed in anaphase indeed stem from a failure to form stable KT-MT attachments and/or properly bi-orient kinetochores in (pro)metaphase.

3) Fig 5I: They interpret that lagging kinetochores in anaphase occur due to defects in

biorientation defects. However, lagging kinetochores in anaphase could also happen due to other defects. If feasible, they should check defects in biorientation in metaphase, for example, using the assay shown in Fig 2A-C.

We agree that it would be interesting if we could test this hypothesis using our bi-orientation assay shown in Fig. 2. However, as the reviewer hints at, such an experiment is technically challenging with our current set-up. We use the blue channel for DAPI staining (needed to identify 2K1N cells in metaphase), the green channel for YFP-tagged kinetochore proteins (receptors for the nanobody-based tethering) and the red channel to verify expression of the CPC^{cat}-tdTomato-VhhGFP4 fusion protein. In theory, it would be possible to tag either KKIP1 or KKIP2 (e.g. with an 3xHA or 3xFLAG tag) and perform immunofluorescence staining using far-red secondary antibodies as a fourth color, but this would need to be first established. The current trypanosome strains used for the nanobody-based tethering assays are already selected based on five different antibiotics (Aurora B^{AUK1-as1} alleles: Hygromycin/G418, YFP-tagged kinetochore protein: Blasticidin, CPC^{cat}-tdTomato-VhhGFP4: Phleomycin, Tet-On system: Puromycin, see Supplemental Table S1). In order to additionally tag KKIP1/2, we would need to establish a sixth antibiotic selection regime, which is not available in the lab. We feel that this is beyond the scope of this revision and kindly ask the reviewer to excuse us for not investing more time and resources in addressing this question.

4) Fig 5I: It will be interesting whether the increase of lagging kinetochores with tethering to the outer kinetochore requires the Aurora B kinase activity. If feasible, the authors can tether catalytically inactive Aurora B to the outer kinetochore to study the outcome.

Some of these experiments may not be technically feasible and I would understand it if they cannot be carried out due to technical difficulty.

It is indeed possible that tethering of CPC^{cat}-tdTomato-VhhGFP4 fusion protein to the outer kinetochore causes defects in KT-MT attachments, which are independent of Aurora B^{AUK1} activity (such as the fusion protein impairing MT-binding activity of KKT4). However, the fact that a robust increase in lagging kinetochores upon targeting CPC^{cat}-tdTomato-VhhGFP4 to the outer kinetochore is only observed when we simultaneously inhibit the endogenous Aurora B^{AUK1} kinase, suggests that this phenotype is caused by shifting the bulk of Aurora B^{AUK1} activity away from the inner kinetochore to the outer kinetochore. As the reviewer suggests, we could address this issue more accurately by employing a catalytically inactive Aurora B^{AUK1}. This would require us to test CPC^{cat}-tdTomato-VhhGFP4 fusions with a

kinase-dead Aurora B^{AUK1} in vivo and derive the corresponding cell lines in the Aurora B^{AUK1-as1} background, which would be possible albeit time consuming. Instead, for this revision, we focused on extending our analysis of KKT14 phosphorylation (see above), to which end we generated >10 new cell lines. We hope that the reviewer appreciates this effort and is content with us discussing this possible caveat of our tethering assays in the text as suggested by the editor. We have added the following statement in the results section:

Intriguingly, targeting CPC^{cat}-tdTomato-VhhGFP4 to the outer kinetochore components KKT4 or KKT14 partially restored cell cycle progression but resulted in a massive increase of lagging chromosomes in anaphase (Figs. 5, H and I). This is unlikely to be an artifact of impairing KT-MT attachments due to steric hindrance (e.g. by physically blocking access of MTs to binding sites at the kinetochore), because tethering of CPC^{cat}-tdTomato-VhhGFP4 to these outer kinetochore proteins without inhibiting the endogenous Aurora B^{AUK1} did not affect cell cycle progression and sustained error-free chromosome segregation (Figs. 5, H and I). Nevertheless, we cannot exclude that tethering may have additional effects independently of target phosphorylation.

In summary, our results imply that localization of Aurora B^{AUK1} to either the inner or outer kinetochore is sufficient to promote anaphase entry in trypanosomes.

Minor points:

1) Fig 4H: It is not clear which experimental groups are compared in the unpaired t-test. This should be clarified.

Thank you for pointing this out. We have now added brackets to highlight which groups are compared in our statistical analysis. Note that this data has been moved to Fig. S4 in the revised manuscript.

2) Line 279-281: A recent paper (Li et al Curr Biol 33, 4557-69, 2023) tested the spatial separation model more directly in budding yeast. This paper can be cited if the authors agree with its relevance.

Thank you for mentioning this paper. We have now added this citation as suggested by the reviewer.

Reviewer #2 (Comments to the Authors (Required)):

In this work, Ballmer and colleagues have studied the function of Aurora kinase in mitotic chromosome separation in the parasite *Trypanosoma brucei*, which represents a highly divergent organism compared to the model systems that have been employed to study this process. *T. brucei* appears to lack a conventional spindle assembly checkpoint and has uniquely configured kinetochores, which suggests that mitotic regulation may employ distinct mechanisms in these organisms. Aurora kinase and components of the chromosomal passenger complex are conserved and may have taken on additional functions. The authors employ an analog-sensitive allele of AUK coupled with fluorescent labeling of kinetoplastid-specific kinetochore proteins to show that loss of AUK activity decreases tension across the kinetochore, which can also cause kinetochore-microtubule attachment defects.

Biochemistry using recombinant AUK-CPC1 subunit active complex and recombinant KKTs showed that KKT14, a potentially divergent Bub1R homolog, is strongly phosphorylated by AUK. Mutation of putative AUK sites on KKT14 to phosphomimetic AAs allows *T. brucei* to compensate for AUK inhibition and continue on to anaphase, which is a striking result.

Building on this, the authors propose that tethering the CPC1-AUK1 active complex to the inner kinetochore using a GFP nanobody is sufficient to allow anaphase progression, while tethering to the outer kinetochore allows progress but with lagging chromosomes.

Overall, this work is an impressive mix of advanced genetics/cell biology and biochemistry that provides novel insights into the function of a divergent CPC and kinetochore structure. To my knowledge, no one has used analog sensitive methods and RNAi simultaneously in *T. brucei*, which makes this work a significant technical advance in the field.

We thank the reviewer for their thorough and favourable assessment, and for appreciating the novelty of our work.

I have three main concerns about the manuscript in its current form:

1. I do not think that the localization and functional data for the recruitment of the in Figure 5D-I is compelling. The tdTomato channel images do not convincingly show that the nanobody-YFP interaction is recruiting the AUK-CPC1-nanobody fusion to the kinetochore. The lookup tables on this channel look like they were set very very low. I think the authors need to provide more compelling imaging to show that the nanobody is recruiting the complex to the kinetochore via the KKT-YFP fusion. The functional result of being able to

reach anaphase is compelling, but the imaging needs to be conclusive. Also, why are there still elevated levels of cells in metaphase in all the cases?

Thank you for pointing this out. We have improved the contrast for the tdTomato channel and have enlarged all the images. We hope that the nanobody-dependent recruitment of AUK1-CPC1 to YFP-tagged kinetochore proteins is clearer now. Note that the data in Fig. 5 has been moved to Fig. 6 in the revised manuscript.

Regarding the second question: We used a very low concentration of doxycycline (7.5 ng/mL) to induce expression of CPC^{cat}-tdTomato-VhhGFP4, as we found that with higher expression levels most of the fusion protein mis-localized to the nucleolus (most likely due to the available YFP receptor sites at the kinetochore reaching saturation). As shown in the graph below, overexpression of CPC^{cat}-tdTomato-VhhGFP4 using doxycycline concentrations of 10 ng/mL (High Dox) or more is sufficient to fully rescue the cell cycle arrest caused by inhibition of the endogenous Aurora B^{AUK1}. This again indicates that Aurora B^{AUK1} is able to promote metaphase-anaphase transitioning regardless of whether it is targeted to the inner or outer kinetochore, although it does so less efficiently and causes lagging kinetochores in anaphase when targeted to the outer kinetochore. We did not include the High Dox (10 ng/mL) condition in the manuscript due to the mis-localization of CPC^{cat}-tdTomato-VhhGFP4 to the nucleolus, which made it impossible to disentangle the effects of inner vs. outer kinetochore targeting.

2. The authors refer to two key pieces of data in their discussion- that a phosphomimetic KKT4 mutant has reduced affinity for MTs and that KKT4 phosphorylation outside of the MT-binding domain stabilizes the KKT4-MT interaction. They list these as "data not shown" or "unpublished observations". As these results would provide important mechanistic insight, I

think this data must be included in the paper so that it can be evaluated, or mention of it should be omitted from the discussion.

Thank you for this comment. We have now included the data in Fig. S3. We performed microtubule co-sedimentation assays using recombinant phosphodeficient and phosphomimetic 6HIS-KKT4¹¹⁵⁻³⁴³ constructs, which show that the phosphomimetic mutations disrupt microtubule binding in vitro. To follow up this finding in vivo, we derived rescue cell lines, allowing expression of C-terminally YFP-tagged, RNAi-resistant complementation constructs, while depleting the untagged allele using a previously validated shRNA construct directed against the 5'UTR (Llauró et al., 2018). The results from these experiments are described below. Note that we were unable to employ the same complementation system for KKT14 as this strategy requires RNAi constructs directed against the 5' or 3'UTR of the tagged gene, which were inefficient for depleting KKT14. The KKT14 shRNA construct used in this study targets the coding region of KKT14.

The fact that KKT4¹¹⁵⁻³⁴³ contains the microtubule-binding domain hints at a potential involvement of the CPC in modulating the interaction of the outer kinetochore with microtubules. Indeed, MT co-sedimentation assays using a 6HIS-KKT4¹¹⁵⁻³⁴³ construct containing phosphomimetic (PM; S/T to D) mutations for putative Aurora B^{AUK1} sites (T234, T266, T267, T268, T316, S334) had reduced affinity for MTs, whereas the corresponding phosphodeficient (PD; S/T to A) mutant behaved similar to the wild-type control (Fig. S3D). To test whether phosphorylation of these residues is important for chromosome segregation, we replaced one allele of KKT4 with C-terminally YFP-tagged KKT4^{PD} or KKT4^{PM} constructs and used a previously validated RNAi construct directed against the 3'UTR of KKT4 to deplete the untagged allele (Llauró et al., 2018). Both KKT4^{PD}-YFP and KKT4^{PM}-YFP were able to rescue the growth defect caused by KKT4 depletion (Fig. S3E), suggesting that Aurora B^{AUK1}-dependent phosphorylation of KKT4 does not affect its MT-binding activity in vivo or that loss of its MT-binding activity can be compensated through other, yet to be identified, MT-binding proteins in trypanosomes.

3. Can the authors explain why overexpression of the phosphomimetic KKT14 allows a bypass of AUK1 activity without prematurely triggering anaphase progression? Does overexpression of any of these KKT14 constructs have an effect on cell division?

When AUK1 activity is not inhibited, overexpression of phosphomimetic (PM) KKT14, unlike phosphodeficient (PD) KKT14, does not noticeably affect cell cycle progression (although it causes lagging kinetochores in anaphase, see below). What we found is that induction of

KKT14 PM only partially rescues the cell cycle defect caused by AUK1 inhibition (see new Fig. 5J). It is likely that phosphorylation of additional targets at the kinetochore/centromere by AUK1 is required for cells to progress into anaphase. We acknowledge this possibility in the results section as follows:

The absence of a full rescue upon depletion of KKT14 or expression of the phosphomimetic mutant suggests that Aurora B^{AUK1} phosphorylates additional kinetochore proteins to promote error-free entry into anaphase and/or that continued activity of Aurora B^{AUK1} at the anaphase central spindle is required for completion of nuclear division.

Regarding the reviewer's second question: Overexpression of both phosphodeficient and phosphomimetic KKT14 clearly impairs cell division (see new Fig. 5, E – F). In addition to measuring cell growth, we have provided a more in-depth analysis on the role of KKT14 phosphorylation by assessing cell cycle progression and lagging kinetochores upon expression KKT14 PD/PM constructs induced with 0.01 (~physiological KKT14 levels) or 1 µg/ml doxycycline (overexpression), in the presence/absence of AUK1 inhibition. Moreover, we tested the effect of KKT14 PD/PM overexpression on kinetochore-microtubule attachments using the kinetochore periphery protein KKIP1 as 'bi-orientation' marker (similarly to our assay using KKIP2 in Fig. 2). The observed growth defect upon overexpression of KKT14 correlates with defects in chromosome bi-orientation and lagging kinetochores in anaphase. We describe our novel experimental results as follows:

To examine the importance of the CPC-dependent phosphorylation of KKT14, we ectopically expressed GFP-KKT14^{NTR} constructs containing wild-type (WT), phosphodeficient (PD; S/T to A) or phosphomimetic (PM; S/T to D/E) with 1 µg/ml doxycycline and monitored cell cycle distribution. We first mutated the phosphorylation sites detected by mass spectrometry (referred to as PD1/PM1). These GFP-KKT14^{NTR} constructs were expressed at similar levels (Fig. S4C) and localized to kinetochores (Fig. S4D). We then created additional PD and PM constructs for which either all or only highly conserved R(x)₀₋₂S/T sites (S25, S107, S113, T333) (Ballmer et al., 2024) were targeted (referred to as PD2/PM2 and PD3/PM3, respectively) (Fig. S4A). We found that expression of WT, PD1, PD2 and to a lesser extent PD3 constructs caused an increase in 2K1N cells (Fig. S4, E, F and H), and thus partially phenocopied Aurora B^{AUK1} inhibition. In contrast, expression of phosphomimetic KKT14^{NTR} or the C-terminal pseudokinase domain did not delay anaphase onset (Fig. S4, G and H). These results suggest that expression of KKT14^{NTR} affects cell cycle progression, which is regulated by Aurora B^{AUK1}.

We next extended our analysis by expressing GFP-fusion of full-length KKT14 PD and PM constructs that mutated all R(x)₀₋₂S/T sites in the NTR with either 0.01 or 1 µg/ml doxycycline (Fig. 5, B, C), and assessed cell growth, cell cycle progression and chromosome segregation fidelity. Expression of KKT14^{PD} and KKT14^{PM} and to a lesser extent KKT14^{WT} slowed down cell growth and caused lagging kinetochores in anaphase (Figs. 5, D–F, and S5, A–C). Overall, we found that the effect of KKT14^{PD} and KKT14^{PM} were more severe compared to KKT14^{WT}, with growth defects and lagging kinetochores observed even when expression was induced with 0.01 µg/ml doxycycline. To test whether the failure to accurately segregate chromosomes upon KKT14 expression was preceded by a defect in the formation of stable KT-MT attachments in (pro)metaphase, we counted the number of kinetochores double positive for the kinetochore periphery protein KKIP1 in cells treated with MG132. Like KKIP2, KKIP1 undergoes tension-dependent displacement upon proper bi-orientation (Llauró et al., 2018). Similarly to Aurora B^{AUK1} inhibition, expression of KKT14 was accompanied by a significant reduction in tdTomato-KKIP1 double positive kinetochores (Fig. S5, D and E), suggesting that the lagging kinetochores observed in anaphase indeed stem from a failure to form stable KT-MT attachments and/or properly bi-orient kinetochores in (pro)metaphase.

Importantly, expression of full-length KKT14^{PD} but not KKT14^{WT} caused a prominent delay in the metaphase-anaphase transition (Fig. 5, G–I). However, when combined with Aurora B^{AUK1} inhibition, KKT14^{PD} did not further impair cell cycle progression, indicating that KKT14 operates downstream of Aurora B^{AUK1} in regulating the timing of anaphase onset. In line with this, expression of KKT14^{PM} rescued the cell cycle arrest caused by Aurora B^{AUK1} inhibition to a similar extent as observed upon KKT14 depletion (Figs. 5, G and J; 4, A and B). Many cells were able to enter an early stage of anaphase, albeit in the presence of lagging kinetochores (Figs. 5G and S5C). The absence of a full rescue upon depletion of KKT14 or expression of the phosphomimetic mutant suggests that Aurora B^{AUK1} phosphorylates additional kinetochore proteins to promote error-free entry into anaphase and/or that continued activity of Aurora B^{AUK1} at the anaphase central spindle is required for completion of nuclear division.

More minor points:

Overall, I found the microscopy images in the figures to be very small, even as larger figures for review. I think that journal-page sized figures will be hard to interpret. Figure 1F was reasonable to me, and the use of magnified boxes such as in 2A also helped significantly with interpreting the data.

We have now increased the sizes of the fluorescence micrographs in all figures. Note that for some images (e.g. Figs. 1D, 4A, 5G), our objective is to provide an overview of the cell cycle distribution rather than to highlight the subcellular localization of particular proteins. In those cases, we aimed at showing ~15-20 cells/nuclei to provide a representative picture of cell cycle distribution for each condition. This means that some cells/nuclei may nevertheless still appear rather small. To guide the reader through these images, we therefore used arrowheads to mark cells in metaphase and anaphase.

Considering the number of different cell lines generated in this work, including complex analog-sensitive and RNAi lines, there is a significant lack of detail about how they were assembled. Is the AS line a dual allele mutant, or was one allele knocked out? This information and more general details about cell line construction should be included as a supplemental.

The AUK1-as1 cell line is indeed a dual allele mutant. We would like to draw the attention of the reviewer to Supplementary Table S1 that was part of our first submission. This table contains detailed descriptions of how all the cell lines and plasmids used in this study (including sequences of oligos and synthetic genes) were generated.

I think versions of the EM in 2D-E without the artificial enhancement should be included as supplemental figures.

We did not perform any artificial enhancement/contrast improvement on the transmission electron micrographs shown in Fig. 2. The raw images were directly exported as .tif files from the imaging software. To conform to the figure size restrictions of the journal, we reduced the image sizes from 16.8 MB to 4.2 MB using ImageJ.

Fig 1A and Fig S1A- It would be preferable to have clear markers that show when cell densities were measured. It's difficult to tell in these graphs as currently depicted.

Thank you for this point. For all the growth curves shown, cell densities were measured in triplicates at 24, 48, 72 and 96 hours. We have now added the following statement in the corresponding figure legends:

Cell densities were measured at 0 h, 24 h, 48 h, 72 h and 96 h. Cultures were diluted at 48 h.

Figure 3D: There is a fair amount of phosphorylation in the KD lanes for the KKT3, 2/3, and 1/3/6 complex. Clearly some other kinase activities are in the CPC1/AUK1 prep, which is concerning. This should be mentioned in the main text, along with the normalization strategy.

The reason for this is that KKT2 and KKT3 are themselves protein kinases (Akiyoshi and Gull, 2014; Nerusheva and Akiyoshi, 2016; Marciano et al., 2019; Ishii et al., 2022). This is also true for KKT10^{CLK1}, shown in Fig. 3D ((Ishii and Akiyoshi, 2020). To make this clearer we have added the following sentence in the corresponding figure legend:

Note that KKT2, KKT3 and KKT10^{CLK1} are kinases themselves and display auto-phosphorylation.

Moreover, we have added a short paragraph on KKT2 and KKT3 in the introduction, since both proteins are used in various experiments throughout this study. TdTomato-KKT2 is used as a kinetochore marker throughout the manuscript, and KKT3-YFP is used as a receptor in the nanobody-based targeting assay (Fig. 6 in the revised manuscript).

We therefore suggest that the term 'outer kinetochore' is used for the microtubule-binding KKT4 protein and KKT14/15, 'kinetochore periphery' for those (e.g. KKIP1, KKIP2, KKIP3, KKT24) whose N- or C-terminal ends are located farther away from KKT4, and 'inner kinetochore' for those proteins that appear as single dots at the resolution of 3D-SIM (Brusini et al., 2021; D'Archivio and Wickstead, 2017; Nerusheva et al., 2019) (Hayashi and Akiyoshi, unpublished data). The latter includes the kinases KKT2 and KKT3, which possess kinase domains classified as unique among known eukaryotic kinase subfamilies (Parsons et al., 2005; Akiyoshi and Gull, 2014). KKT2 and KKT3 localize to centromeres throughout the cell cycle using unique zinc-binding central domains and have divergent polo boxes required for the localization of other kinetochore proteins (Marciano et al., 2019; Ishii et al., 2022). Hence, they are thought to form the base of the kinetochore assembly hierarchy in *T. brucei*.

Figure 3F- Does the quantitation account for the large differences in size of the different substrates? The experimental says that the signal was normalized for "amount", but is that just weight or Mol? Because with substrates that are this different in size the molar amount in the substrate can be quite different, which would affect the expected amount of signal during the kinase assay.

Our quantification of substrate phosphorylation in the kinase assay does indeed account for the differences in size/molecular weight of the substrates, since we normalize/divide the ^{32}P signal by the total weight of that protein loaded onto the gel, estimated by measuring the intensity of the Coomassie-stained bands. We are aware that small differences may exist between substrates in the number of basic residues that can be bound by the dye. Nevertheless, this metric should still serve as reasonable estimation. Furthermore, since there may be a small amount of co-purifying kinases in some of the preps and/or some substrates are themselves protein kinases, we performed each phosphorylation experiment in parallel using a kinase-dead AUK1 control. This allowed us to subtract the normalized intensities from the kinase-dead controls for each sample. Hence our quantification can be thought to reflect the relative amount/frequency of sites phosphorylated by AUK1 per substrate.

We added a description of how the normalized ^{32}P signal intensities were quantified in the Fig. 3 legend and modified the existing paragraph in the Materials & Methods section as follows:

Normalized ^{32}P signal intensities for indicated proteins relative to Aurora B^{AUK1} auto-phosphorylation. The ^{32}P signal intensity for each substrate was normalized to the total protein amount (estimated from measuring the intensity of Coomassie-stained bands). To correct for non-Aurora B^{AUK1}-dependent phosphorylation, the normalized intensities from the kinase-dead controls were subtracted from these values.

Figure S3: The MBP-CPC2 kinase assay experiments are hard to interpret due to the presence of MBP fusion as a potential substrate. I would cleave the MBP or omit this data.

We agree with the reviewer on this point and removed this data from Fig. S3.

Line 307-309: confusingly phrased.

We have now simplified the conclusion statement of this results section. Furthermore, as discussed by the editor and reviewer 1, we added a sentence acknowledging that tethering of CPC^{cat}-tdTomato-VhhGFP4 to kinetochore proteins may cause effects independently of target phosphorylation. This paragraph now reads as follows:

Intriguingly, targeting CPC^{cat}-tdTomato-VhhGFP4 to the outer kinetochore components KKT4 or KKT14 partially restored cell cycle progression but resulted in a massive increase of lagging

chromosomes in anaphase (Figs. 5, H and I). This is unlikely to be an artifact of impairing KT-MT attachments due to steric hindrance (e.g. by physically blocking access of MTs to binding sites at the kinetochore), because tethering of CPC^{cat}-tdTomato-VhhGFP4 to these outer kinetochore proteins without inhibiting the endogenous Aurora B^{AUK1} did not affect cell cycle progression and sustained error-free chromosome segregation (Figs. 5, H and I). Nevertheless, we cannot exclude that tethering may have additional effects independently of target phosphorylation. In summary, our results imply that localization of Aurora B^{AUK1} to either the inner or outer kinetochore is sufficient to promote anaphase entry in trypanosomes.

Figure 5H: What's the "Negative" control?

In the 'negative' control, the experiment is performed in the absence of any YFP-tagged protein. Hence, the CPC^{cat}-tdTomato-VhhGFP4 fusion protein remains in the nucleoplasm and is not targeted to kinetochores (as shown in Fig. 6C). To clarify this, we have labelled Fig. 6C accordingly, and added the following statement in the legend:

(H) Cell cycle profiles for indicated treatment regimes. 'Control' cells were treated with DMSO for 4 h. 'Negative' control corresponds to a cell line that does not express any YFP-tagged protein (as shown in (C)).

Reviewer #3 (Comments to the Authors (Required)):

In this manuscript from the Akiyoshi lab by Ballmer and colleagues, the authors essentially explore the role of the Aurora B homologue (AUK1) in kinetoplastid mitosis using the trypanosome model *T. brucei*. The authors demonstrate that compared to the other common model organisms used to study the CPC and kinetochores, *T. brucei* AUK1 has both conserved and divergent functions. In particular, using an analogue sensitive allele of AUK1, the authors demonstrate a role for the kinase activity in mediating the metaphase-anaphase transition. After screening a panel of substrates, the authors find strong phosphorylation of the potential BUB1 orthologue KKT14 by AUK1. Phospho-mimetic mutants of the AUK1 sites on KKT14 promote Met-ana transition, whereas the non-phosphorylatable versions did not, rather mimicking AUK1 inhibition. In addition, the authors demonstrated that AUK1 activity supports proper spindle formation and biorientation and finally that the position of AUK1 at the outer-kinetochore destabilizes attachments resulting in massive errors in chromosome segregation but that inner or outer kinetochore AUK1 positioning supports met-ana transition.

Overall, I feel that there are several interesting discoveries outlined in this manuscript, but that their impact is limited by lack of more in-depth exploration which unfortunately has made for a rather disjointed story of two phenotypes and in this reviewer's opinion, not suitable as an article in JCB.

We thank the reviewer for their thorough assessment of our manuscript and the insightful feedback provided to further improve our study.

Major points

My main concern is that none of the discoveries are thoroughly followed up in any depth.

Below are a few examples:

1. In the analysis of KKT14 phospho-sites, the authors are unclear as to which sites are mutated. In the methods section, they list a number of sites some separated by comma others by a slash (/) meaning that the site could not be assigned, presumably. In the figure legend, there's 2 fewer sites mentioned, so why were the 2 additional sites mutated if there was no evidence they were phosphorylated? The authors should clarify their nomenclature and the exact sites tested.

Thank you for this comment. As the reviewer correctly points out, those sites separated by a slash (/) could not be clearly assigned based on the mass spectrometry data. We have now clarified this in the Materials & Methods section as follows:

Following S/T sites were mutated to A or D/E in KKT14^{NTR} PD1 and PM1 constructs, respectively: S25, T104, S107, S113, T115, S173/S174, T299/S300/S301/S302/S303, T332/T333, S347/T348, whereby those sites separated by a slash (/) could not be assigned unequivocally based on the mass spectrometry data.

In addition, we moved the list of phosphosites from the legend to the actual figure (previously Fig. 4, now Fig. S4 in the revised manuscript). This is now presented alongside a larger, improved schematic of KKT14, which also includes the position of the ABBA motif (as suggested by the reviewer in Minor points).

2. Overall the authors need to narrow down their phospho-site analysis of KKT14 to the most relevant sites or provide convincing evidence that ALL the sites mutated are functionally relevant.

The authors presumably mutated all sites in KKT14 that they identified as AUK1 targets.

This is a brute force method that is somewhat crude. Given the number of sites mutated,

how can the authors rule out that what they seeing is a result of disrupting the global structure of the N-terminus of KKT14 rather than the phosphor-sites? The authors need to prioritize sites based on some criteria (eg. Conservation amongst tryp sp. and/or adherence to the AUK1 consensus motif). At minimum, the authors need to generate the individual and triple phosphor-mimetic/non-phosphorylatable mutants covering the three most promising AUK1 sites they mentioned and test the function of these collectively or individually before moving on to more encompassing mutagenesis experiments.

We thank the reviewer for these suggestions. We have extended our analysis of the Aurora B-dependent phosphorylation of KKT14, which is now presented in a new main and two supplementary figures (Fig. S4, 5, S5). These new results support our conclusion that KKT14 phospho-deficient mutants partially phenocopy Aurora B inhibition and that phospho-mimetic mutants partially alleviate the cell cycle delay caused by Aurora B inhibition.

We are aware that our previous analysis was limited to overexpression of phosphodeficient (PD) and phosphomimetic (PM) KKT14^{NTR} constructs, in which we mutated in vitro phosphosites detected by mass spectrometry (now referred to as PD1/PM1 in the revised manuscript). Some of these sites do not conform to the Aurora B^{AUK1} consensus motif, while other potentially interesting phosphosites may not have been detected for technical reasons. To provide a more comprehensive assessment, we have now tested additional KKT14^{NTR} PD/PM constructs: PD2/PM2, in which we targeted all R(x)₀₋₂S/T sites (including those that were not found by mass spectrometry), and PD3/PM3, in which we targeted four highly conserved R(x)₀₋₂S/T sites (S25, S107, S113, T333) (Ballmer et al., 2024), three of which correspond to those mentioned by the reviewer (e.g. which showed reduced phosphorylation in our IP-MS experiment). Overexpression of KKT14^{NTR} PD3 caused a similar delay in the metaphase-anaphase transition as PD1 and PD2, whereas PM3 did not, suggesting that S25, S107, S113, T333 are indeed important sites. Of note, the KKT14 NTR is predicted to be largely unstructured based on AlphaFold2 (Ballmer et al., 2024). We deem it unlikely that mutating these four conserved sites or even the full complement of R(x)₀₋₂S/T sites in the NTR disrupts KKT14 structure and/or causes protein instability, as all constructs were expressed (according to fluorescence microscopy and Western blotting) and were still able to localize to kinetochores.

In addition to our KKT14^{NTR} overexpression analysis, we also generated new cell lines allowing inducible expression of full-length KKT14 PD/PM constructs. Inspired by the reviewers' comments, we followed up the role of KKT14 phosphorylation in more detail by assessing cell proliferation, cell cycle progression and lagging kinetochores upon induction

of KKT14 PD/PM with 0.01 (~physiological KKT14 levels) or 1 µg/ml doxycycline (overexpression), with or without Aurora B^{AUK1} inhibition. Moreover, as suggested by reviewer 1, we tested the effect of KKT14 PD/PM overexpression on kinetochore-microtubule attachments using the kinetochore periphery protein KKIP1 as bi-orientation marker (similarly to our assay using KKIP2 in Fig. 2). In this case, we have used tdTomato-KKIP1, since the corresponding cell line was straightforward to derive, whereas it would have taken additional steps to generate Aurora B^{AUK1}-as1/tdTomato-KKIP2/GFP-KKT14 (WT/PD/PM) lines. We describe our new experimental data as follows:

To examine the importance of the CPC-dependent phosphorylation of KKT14, we ectopically expressed GFP-KKT14^{NTR} constructs containing wild-type (WT), phosphodeficient (PD; S/T to A) or phosphomimetic (PM; S/T to D/E) with 1 µg/ml doxycycline and monitored cell cycle distribution. We first mutated the phosphorylation sites detected by mass spectrometry (referred to as PD1/PM1). These GFP-KKT14^{NTR} constructs were expressed at similar levels (Fig. S4C) and localized to kinetochores (Fig. S4D). We then created additional PD and PM constructs for which either all or only highly conserved R(x)₀₋₂S/T sites (S25, S107, S113, T333) (Ballmer et al., 2024) were targeted (referred to as PD2/PM2 and PD3/PM3, respectively) (Fig. S4A). We found that expression of WT, PD1, PD2 and to a lesser extent PD3 constructs caused an increase in 2K1N cells (Fig. S4, E, F and H), and thus partially phenocopied Aurora B^{AUK1} inhibition. In contrast, expression of phosphomimetic KKT14^{NTR} or the C-terminal pseudokinase domain did not delay anaphase onset (Fig. S4, G and H). These results suggest that expression of KKT14^{NTR} affects cell cycle progression, which is regulated by Aurora B^{AUK1}.

We next extended our analysis by expressing GFP-fusion of full-length KKT14 PD and PM constructs that mutated all R(x)₀₋₂S/T sites in the NTR with either 0.01 or 1 µg/ml doxycycline (Fig. 5, B, C), and assessed cell growth, cell cycle progression and chromosome segregation fidelity. Expression of KKT14^{PD} and KKT14^{PM} and to a lesser extent KKT14^{WT} slowed down cell growth and caused lagging kinetochores in anaphase (Figs. 5, D–F, and S5, A–C). Overall, we found that the effect of KKT14^{PD} and KKT14^{PM} were more severe compared to KKT14^{WT}, with growth defects and lagging kinetochores observed even when expression was induced with 0.01 µg/ml doxycycline. To test whether the failure to accurately segregate chromosomes upon KKT14 expression was preceded by a defect in the formation of stable KT-MT attachments in (pro)metaphase, we counted the number of kinetochores double positive for the kinetochore periphery protein KKIP1 in cells treated with MG132. Like KKIP2, KKIP1 undergoes tension-dependent displacement upon proper bi-orientation (Llauró et al., 2018). Similarly to Aurora B^{AUK1} inhibition, expression of KKT14 was accompanied by a significant reduction in tdTomato-KKIP1 double positive kinetochores (Fig. S5, D and E),

suggesting that the lagging kinetochores observed in anaphase indeed stem from a failure to form stable KT-MT attachments and/or properly bi-orient kinetochores in (pro)metaphase.

Importantly, expression of full-length KKT14^{PD} but not KKT14^{WT} caused a prominent delay in the metaphase-anaphase transition (Fig. 5, G–I). However, when combined with Aurora B^{AUK1} inhibition, KKT14^{PD} did not further impair cell cycle progression, indicating that KKT14 operates downstream of Aurora B^{AUK1} in regulating the timing of anaphase onset. In line with this, expression of KKT14^{PM} rescued the cell cycle arrest caused by Aurora B^{AUK1} inhibition to a similar extent as observed upon KKT14 depletion (Figs. 5, G and J; 4, A and B). Many cells were able to enter an early stage of anaphase, albeit in the presence of lagging kinetochores (Figs. 5G and S5C). The absence of a full rescue upon depletion of KKT14 or expression of the phosphomimetic mutant suggests that Aurora B^{AUK1} phosphorylates additional kinetochore proteins to promote error-free entry into anaphase and/or that continued activity of Aurora B^{AUK1} at the anaphase central spindle is required for completion of nuclear division.

3. The effect of KKT14 depletion in Fig 4B is only partial suggesting that other AUK1 targets may be more important than KKT14, despite KKT14 being an excellent substrate. The authors should discuss this and other alternative explanations for the data. Are there any other candidates amongst their identified targets that could rescue the AUK1 inhibition phenotype when depleted? This lab has a unique access to resources to test this idea and they should push the analysis further to really explore the role of AUK1 in Met-ana transition.

Indeed, we found that depletion of KKT14 (or expression of the phosphomimetic KKT14 mutant) only partially rescues the cell cycle defect caused by Aurora B^{AUK1} inhibition. Here, we decided to focus on KKT14, since it was the best substrate of Aurora B^{AUK1} in our in vitro kinase assay and its homology to Bub1/BubR1 proteins, which in other eukaryotes are key components of the spindle checkpoint. Nevertheless, as the reviewer points out, it is likely that the CPC needs to phosphorylate additional targets at the kinetochore/centromere for cells to progress into anaphase. These may include other KKT proteins, such as KKT1, KKT7, KKT8 (which were also robustly phosphorylated by Aurora B^{AUK1}), CPC subunits, components of the APC/C, cohesin etc. We have now started exploring such possibilities in more detail in our group – we are determined to elucidate the mechanism controlling anaphase onset in trypanosomes, which appears to be quite different from the spindle checkpoint-based system described in traditional model eukaryotes. However, we believe it is beyond the scope of this particular manuscript to extend our analysis of Aurora B^{AUK1} targets beyond KKT14 and KKT4. Note that this is first study characterizing the role of the

CPC in regulating chromosome segregation in kinetoplastids and the first attempt at dissecting the regulatory network controlling the metaphase-anaphase transition in this organism. We kindly ask the reviewer to excuse us for not investing more time and resources in testing the importance of other potential Aurora B^{AUK1} substrates in this work.

We acknowledge the possibility that CPC-dependent phosphorylation of additional targets is required for metaphase-anaphase transition in the results section as follows:

The absence of a full rescue upon depletion of KKT14 or expression of the phosphomimetic mutant suggests that Aurora B^{AUK1} phosphorylates additional kinetochore proteins to promote error-free entry into anaphase and/or that continued activity of Aurora B^{AUK1} at the anaphase central spindle is required for completion of nuclear division.

In addition, we extended our discussion to speculate on how KKT14 phosphorylation may regulate APC/C activation and how phosphorylation of other CPC and kinetochore components may play a role in the regulation of mitotic exit:

Remarkably, inhibition of Aurora B^{AUK1} in *T. brucei* arrests cells in metaphase, a phenotype that has not been reported in traditional model eukaryotes (Biggins and Murray, 2001; Hauf et al., 2003). Our data suggests that the divergent Bub1-homologue KKT14 is a main target of Aurora B^{AUK1}. The KKT14 NTR contains an ABBA motif (a conserved CDC20-interaction motif) (Ballmer et al., 2024), suggesting that this domain might be involved in regulating APC/C activity. Although direct binding to CDC20 and/or APC/C subunits remains to be demonstrated, it is possible that the KKT14 NTR in its unphosphorylated state might prevent premature APC/C activation by sequestering CDC20 or certain subunits of the APC/C, and that this inhibition could be relieved by Aurora B^{AUK1} activity. Alternatively, the NTR in its phosphorylated state may act as a scaffold that promotes APC/C-CDC20 interaction and its subsequent activation. A third possibility is that kinetoplastids evolved an APC/C- or CDC20-independent mechanism to regulate the metaphase-anaphase transition.

Apart from KKT14, Aurora B^{AUK1} also phosphorylated several CPC subunits in vitro, including its activator INCENP^{CPC1} and the C-terminal tail of KIN-A. We recently demonstrated that the C-terminal unstructured tail of KIN-A plays a key role in targeting the CPC to kinetochores by interacting with the KKT8 complex at the inner kinetochore. We speculate that Aurora B^{AUK1}-dependent phosphorylation of KIN-A may modulate its kinetochore-binding affinity. Importantly, inhibition of Aurora B^{AUK1} using our analogue sensitive approach did not impair kinetochore localization of the CPC. Rather, phosphorylation of the KIN-A C-terminal tail could serve to weaken the association of KIN-A with its kinetochore receptors and facilitate

the release of the CPC from kinetochores onto spindle microtubules upon metaphase-anaphase transition. An intriguing possibility is that the interaction of the KIN-A motor domain with microtubules in (pro)metaphase, coupled to the establishment of proper kinetochore-microtubule attachments, may trigger conformational changes within the CPC (such as tension-dependent stretch of the KIN-A C-terminus) and/or full activation of Aurora B^{AUK1}, thereby allowing temporally and spatially regulated phosphorylation of the KIN-A C-terminal tail. Whether timely release of KIN-A from kinetochores may be a prerequisite for mitotic exit in addition to KKT14 phosphorylation will require further experimental testing in the future.

Further potential targets of the trypanosome CPC identified through our in vitro kinase assays include the inner kinetochore components KKT1, KKT7 and KKT8, which are also substrates of the two functionally redundant KKT10/19 kinases (also called CLK1/2) (Ishii and Akiyoshi, 2020). Interestingly, KKT10 and KKT19 localize to the inner kinetochore by binding to the N-terminus of KKT7, which in conjuncture with the KKT8 complex also serves as the main recruitment arm for the trypanosome CPC (Ishii and Akiyoshi, 2020; Ballmer et al., 2024). Moreover, co-depletion of KKT10/19 causes a delay in the metaphase-anaphase progression and lagging kinetochores in anaphase, raising the possibility that these kinases may be involved in some form of error correction process and/or form part of the regulatory circuitry controlling anaphase onset. Contrary to Aurora B^{AUK1}, however, KKT10/19 phosphorylate the C-terminal domain of KKT4 rather than its MT-binding domain (Ishi and Akiyoshi 2020), and the mitotic spindle appears to be hyper-stabilized rather than destabilized in these mutants (unpublished observations). Thus, it is conceivable that KKT10/19 and Aurora B^{AUK1} may play opposing roles for spindle stability, despite their close spatial association.

4. Where are the KTT14 phospho-sites relative to the putative ABBA motif, and is this ABBA motif important for the Met-Ana transition? At the very least, the authors need to check the effect of the ABBA mutant on Met-ana transition and whether there is cooperativity with the phosphorylation sites on this process.

We thank the reviewer for this important point. Overexpression of KKT14^{NTR} with the ABBA motif mutated to 'AAAAAA' (ABBA 6A) had the same effect on cell cycle progression as overexpression of the wild-type control (see below).

We failed to derive an ABBA 6A line targeting the endogenous locus, which could indicate that this motif is essential for cell division. We will need to establish additional tools (both in vivo and in vitro) to study the functional importance of this motif. Given that our preliminary results were negative, and we currently do not have any robust data connecting Aurora B^{AUK1} activity to CDC20-binding through the KKT14 ABBA motif, we would prefer to leave out any experiments focusing on the ABBA motif in this manuscript.

5. In fig 4h, the authors need to include an RNAi only control without the rescue of the KKT14.

Cell cycle progression is only modestly affected by depletion of KKT14 as shown in Fig. 4, A and B. Note that these results are consistent with our previously published characterization of the KKT14 RNAi phenotype (see (Ballmer et al., 2024), Fig. 7).

6. The effect of AUK1 kinase activity on biorientation shown in fig 1 and the last figure is exciting, but there are no follow-up experiments. The authors speculate and report some unpublished observations in the discussion on the potential role of KKT4 in this but provide no experimental data. At the very least to make a more complete story the authors need to perform the same type of phospho-mutants analysis with KKT4 as they have with KKT14. In particular they should demonstrate the effect of the mutations in the biorientation assay and also on the microtubule binding.

We have included the microtubule co-sedimentation assays using recombinant phosphodeficient and phosphomimetic KKT4¹¹⁵⁻³⁴³ constructs in Fig. S3, which show that the phosphomimetic mutations disrupt microtubule binding in vitro. To follow this up in vivo, we derived rescue cell lines, allowing expression of C-terminally YFP-tagged, RNAi-resistant complementation constructs, while depleting the untagged allele using a previously validated

shRNA construct directed against the 5'UTR (Llauró et al., 2018). The results from these experiments are described below. Note that we were unable to employ the same complementation system for KKT14 as this strategy requires RNAi constructs directed against the 5' or 3'UTR of the tagged gene, which turned out to be inefficient for depleting KKT14. The KKT14 RNAi construct used in this study is an shRNA directed against the coding region of KKT14.

The fact that KKT4¹¹⁵⁻³⁴³ contains the microtubule-binding domain hints at a potential involvement of the CPC in modulating the interaction of the outer kinetochore with microtubules. Indeed, MT co-sedimentation assays using a 6HIS-KKT4¹¹⁵⁻³⁴³ construct containing phosphomimetic (PM; S/T to D) mutations for putative Aurora B^{AUK1} sites (T234, T266, T267, T268, T316, S334) had reduced affinity for MTs, whereas the corresponding phosphodeficient (PD; S/T to A) mutant behaved similar to the wild-type control (Fig. S3D). To test whether phosphorylation of these residues is important for chromosome segregation, we replaced one allele of KKT4 with C-terminally YFP-tagged KKT4^{PD} or KKT4^{PM} constructs and used a previously validated RNAi construct directed against the 3'UTR of KKT4 to deplete the untagged allele (Llauró et al., 2018). Both KKT4^{PD}-YFP and KKT4^{PM}-YFP were able to rescue the growth defect caused by KKT4 depletion (Fig. S3E), suggesting that Aurora B^{AUK1}-dependent phosphorylation of KKT4 does not affect its MT-binding activity in vivo or that loss of its MT-binding activity can be compensated through other, yet to be identified, MT-binding proteins in trypanosomes.

Minor points:

1. Figure 4 D is too small and the position of the phosphosites should be shown relative to other known motifs such as the ABBA.

Thank you for pointing this out. We have enlarged the schematics (Fig. S4 and 5 in the revised manuscript) showing the organisation and position of the Aurora B^{AUK1} phosphosites as well as the ABBA motif.

2. Fig S1K is an important observation the deserves to be part of a main figure

We thank the reviewer for this suggestion. We have now moved Fig. S1K (alongside Fig. S1G) to the main Fig. 1.

3. There is no evidence that the RNAi used throughout the manuscript is functional. The

authors should show WB, IF or even pPCR data that it is functioning as expected. If these were demonstrated in a previous manuscript, this needs to be referenced.

All the RNAi constructs used in this study have been previously published and validated (Aurora B^{AUK1}: (Ballmer and Akiyoshi, 2024), KKT14: (Marcianò et al., 2021; Ballmer et al., 2024), KKT4: (Llauró et al., 2018)). We now made sure that relevant papers were cited where necessary.

References

- Akiyoshi, B., and K. Gull. 2014. Discovery of Unconventional Kinetochores in Kinetoplastids. *Cell*. 156. doi:10.1016/j.cell.2014.01.049.
- Ballmer, D., and B. Akiyoshi. 2024. Dynamic localization of the chromosomal passenger complex in trypanosomes is controlled by the orphan kinesins KIN-A and KIN-B. *Elife*. 13. doi:10.7554/ELIFE.93522.
- Ballmer, D., W. Carter, J.J.E. van Hooff, E.C. Tromer, M. Ishii, P. Ludzia, and B. Akiyoshi. 2024b. Kinetoplastid kinetochore proteins KKT14–KKT15 are divergent Bub1/BubR1–Bub3 proteins. *Open Biol*. 14. doi:10.1098/RSOB.240025.
- Biggins, S., and A.W. Murray. 2001. The budding yeast protein kinase Ipl1/Aurora allows the absence of tension to activate the spindle checkpoint. *Genes Dev*. 15:3118. doi:10.1101/GAD.934801.
- Brusini, L., S. D'Archivio, J. McDonald, and B. Wickstead. 2021. Trypanosome KKIP1 Dynamically Links the Inner Kinetochore to a Kinetoplastid Outer Kinetochore Complex. *Front Cell Infect Microbiol*. 11. doi:10.3389/fcimb.2021.641174.
- D'Archivio, S., and B. Wickstead. 2017. Trypanosome outer kinetochore proteins suggest conservation of chromosome segregation machinery across eukaryotes. *Journal of Cell Biology*. 216. doi:10.1083/jcb.201608043.
- Hauf, S., R.W. Cole, S. LaTerra, C. Zimmer, G. Schnapp, R. Walter, A. Heckel, J. van Meel, C.L. Rieder, and J.-M. Peters. 2003. The small molecule Hesperadin reveals a role for Aurora B in correcting kinetochore–microtubule attachment and in maintaining the spindle assembly checkpoint. *Journal of Cell Biology*. 161. doi:10.1083/jcb.200208092.
- Ishii, M., and B. Akiyoshi. 2020. Characterization of unconventional kinetochore kinases KKT10/19 in *Trypanosoma brucei*. *J Cell Sci*. doi:10.1242/jcs.240978.
- Ishii, M., P. Ludzia, G. Marcianò, W. Allen, O.O. Nerusheva, and B. Akiyoshi. 2022. Divergent polo boxes in KKT2 bind KKT1 to initiate the kinetochore assembly cascade in *Trypanosoma brucei*. *Mol Biol Cell*. 33. doi:10.1091/MBE.E22-07-0269-T.
- Llauró, A., H. Hayashi, M.E. Bailey, A. Wilson, P. Ludzia, C.L. Asbury, and B. Akiyoshi. 2018. The kinetoplastid kinetochore protein KKT4 is an unconventional microtubule tip–coupling protein. *Journal of Cell Biology*. 217. doi:10.1083/jcb.201711181.

- Marciano, G., M. Ishii, O. Nerusheva, and B. Akiyoshi. 2019. Unconventional kinetochore kinases KKT2 and KKT3 have unique centromere localization domains. *bioRxiv*.
- Marcianò, G., M. Ishii, O.O. Nerusheva, and B. Akiyoshi. 2021. Kinetoplastid kinetochore proteins KKT2 and KKT3 have unique centromere localization domains. *J Cell Biol.* 220. doi:10.1083/JCB.202101022.
- Nerusheva, O.O., and B. Akiyoshi. 2016. Divergent polo box domains underpin the unique kinetoplastid kinetochore. *Open Biol.* 6. doi:10.1098/rsob.150206.
- Nerusheva, O.O., P. Ludzia, and B. Akiyoshi. 2019. Identification of four unconventional kinetoplastid kinetochore proteins KKT22–25 in *Trypanosoma brucei*. *Open Biol.* 9. doi:10.1098/rsob.190236.
- Parsons, M., E.A. Worthey, P.N. Ward, and J.C. Mottram. 2005. Comparative analysis of the kinomes of three pathogenic trypanosomatids: *Leishmania major*, *Trypanosoma brucei* and *Trypanosoma cruzi*. *BMC Genomics.* 6. doi:10.1186/1471-2164-6-127.

July 17, 2024

RE: JCB Manuscript #202401169R

Dr. Bungo Akiyoshi
University of Edinburgh
Max Born Crescent
Edinburgh EH9 3BF
United Kingdom

Dear Dr. Akiyoshi,

Thank you for submitting your revised manuscript entitled "Aurora B controls anaphase onset and error-free chromosome segregation in trypanosomes." The manuscript has been re-reviewed by two of the original reviewers. Reviewer #3 was not available for re-review. We would be happy to publish your paper in JCB pending the minor changes recommended by the reviewers as well as final revisions necessary to meet our formatting guidelines (see details below).

A. MANUSCRIPT ORGANIZATION AND FORMATTING:

1) Text limits: Character count for Articles is < 40,000, not including spaces. Count includes title page, abstract, introduction, results, discussion, and acknowledgments. Count does not include materials and methods, figure legends, references, tables, or supplemental legends.

2) Figure formatting: Articles may have up to 10 main text figures. Scale bars must be present on all microscopy images, including inset magnifications. Molecular weight or nucleic acid size markers must be included on all gel electrophoresis. Please add scale bars to Figure 2F and the magnifications images in Figures 2A & S5D.

Also, please avoid pairing red and green for images and graphs to ensure legibility for color-blind readers. If red and green are paired for images, please ensure that the particular red and green hues used in micrographs are distinctive with any of the colorblind types. If not, please modify colors accordingly or provide separate images of the individual channels.

3) Statistical analysis: Error bars on graphic representations of numerical data must be clearly described in the figure legend. The number of independent data points (n) represented in a graph must be indicated in the legend. Please, indicate whether 'n' refers to technical or biological replicates (i.e. number of analyzed cells, samples or animals, number of independent experiments). If independent experiments with multiple biological replicates have been performed, we recommend using distribution-reproducibility SuperPlots (please see Lord et al., JCB 2020) to better display the distribution of the entire dataset, and report statistics (such as means, error bars, and P values) that address the reproducibility of the findings.

Statistical methods should be explained in full in the materials and methods. For figures presenting pooled data the statistical measure should be defined in the figure legends. Please also be sure to indicate the statistical tests used in each of your experiments (both in the figure legend itself and in a separate methods section) as well as the parameters of the test (for example, if you ran a t-test, please indicate if it was one- or two-sided, etc.). Also, if you used parametric tests, please indicate if the data distribution was tested for normality (and if so, how). If not, you must state something to the effect that "Data distribution was assumed to be normal but this was not formally tested."

4) Materials and methods: Should be comprehensive and not simply reference a previous publication for details on how an experiment was performed. Please provide full descriptions (at least in brief) in the text for readers who may not have access to referenced manuscripts. The text should not refer to methods "...as previously described." Please also indicate the type of membrane used for immunoblotting.

5) For all cell lines, vectors, constructs/cDNAs, etc. - all genetic material: please include database / vendor ID (e.g., Addgene, ATCC, etc.) or if unavailable, please briefly describe their basic genetic features, even if described in other published work or gifted to you by other investigators (and provide references where appropriate). Please be sure to provide the sequences for all of your oligos: primers, si/shRNA, RNAi, gRNAs, etc. in the materials and methods. You must also indicate in the methods the source, species, and catalog numbers/vendor identifiers (where appropriate) for all of your antibodies, including secondary. If antibodies are not commercial, please add a reference citation if possible.

6) Microscope image acquisition: The following information must be provided about the acquisition and processing of images:

- a. Make and model of microscope
- b. Type, magnification, and numerical aperture of the objective lenses
- c. Temperature
- d. Imaging medium
- e. Fluorochromes
- f. Camera make and model
- g. Acquisition software
- h. Any software used for image processing subsequent to data acquisition. Please include details and types of operations involved (e.g., type of deconvolution, 3D reconstitutions, surface or volume rendering, gamma adjustments, etc.).

7) References: There is no limit to the number of references cited in a manuscript. References should be cited parenthetically in the text by author and year of publication. Abbreviate the names of journals according to PubMed.

8) Supplemental materials: Articles may have up to 5 supplemental figures and 10 videos. Please also note that tables, like figures, should be provided as individual, editable files. A summary of all supplemental material should appear at the end of the Materials and methods section. Please include one brief sentence per item.

9) eTOC summary: A ~40-50 word summary that describes the context and significance of the findings for a general readership should be included on the title page. The statement should be written in the present tense and refer to the work in the third person. It should begin with "First author name(s) et al..." to match our preferred style.

10) Conflict of interest statement: JCB requires inclusion of a statement in the acknowledgements regarding competing financial interests. If no competing financial interests exist, please include the following statement: "The authors declare no competing financial interests." If competing interests are declared, please follow your statement of these competing interests with the following statement: "The authors declare no further competing financial interests."

11) A separate author contribution section is required following the Acknowledgments in all research manuscripts. All authors should be mentioned and designated by their first and middle initials and full surnames. We encourage use of the CRediT nomenclature (<https://casrai.org/credit/>).

12) ORCID IDs: ORCID IDs are unique identifiers allowing researchers to create a record of their various scholarly contributions in a single place. Please note that ORCID IDs are required for all authors. At resubmission of your final files, please be sure to provide your ORCID ID and those of all co-authors.

13) JCB requires authors to submit Source Data used to generate figures containing gels and Western blots with all revised manuscripts. This Source Data consists of fully uncropped and unprocessed images for each gel/blot displayed in the main and supplemental figures. Since your paper includes cropped gel and/or blot images, please be sure to provide one Source Data file for each figure that contains gels and/or blots along with your revised manuscript files. File names for Source Data figures should be alphanumeric without any spaces or special characters (i.e., SourceDataF#, where F# refers to the associated main figure number or SourceDataFS# for those associated with Supplementary figures). The lanes of the gels/blots should be labeled as they are in the associated figure, the place where cropping was applied should be marked (with a box), and molecular weight/size standards should be labeled wherever possible. Source Data files will be directly linked to specific figures in the published article.

14) Journal of Cell Biology now requires a data availability statement for all research article submissions. These statements will be published in the article directly above the Acknowledgments. The statement should address all data underlying the research presented in the manuscript. Please visit the JCB instructions for authors for guidelines and examples of statements at (<https://rupress.org/jcb/pages/editorial-policies#data-availability-statement>).

B. FINAL FILES:

Thank you for your attention to these final processing requirements. Please revise and format the manuscript and upload materials within 7 days. If you need an extension for whatever reason, please let us know and we can work with you to determine a suitable revision period.

Thank you for this interesting contribution, we look forward to publishing your paper in Journal of Cell Biology.

Sincerely,

Arshad Desai, PhD
Monitoring Editor
Journal of Cell Biology

Dan Simon, PhD
Scientific Editor
Journal of Cell Biology

Reviewer #1 (Comments to the Authors (Required)):

The authors satisfactorily addressed all the points I raised in the first review. I recommend the revised manuscript for publication in jCB.

Reviewer #2 (Comments to the Authors (Required)):

The resubmission has addressed my main criticisms of the first version of the manuscript. I think the updated work on KKT14, with finer control on the number of phosphosites targeted for mutation, has improved and focused the results. It is a little unfortunate that the in vivo phosphomutants of KKT4 do not appear to have any phenotypes, but that is still useful information. The increased lookups for the nanobody fusion make the localization of these constructs much easier to interpret.

Is the "High dox" condition truly 10 ng/mL, compared to "low dox" at 7.5 ng/mL. I'm surprised that such a small difference leads to a very big change in the localization of the nanobody fusion, considering that full-bore induction is usually considered to be around 100 ng/mL.

Per the authors' response to my point about the artificial enhancement of their EM, I was referring to the brown lines used to enhance the MTs in the figure 2 insets. I would like to see those images without the lines, so that we can draw our own conclusions about the positioning and calls on the MTs.

Reviewer #1 (Comments to the Authors (Required)):

The authors satisfactorily addressed all the points I raised in the first review. I recommend the revised manuscript for publication in JCB.

We thank the reviewer for his/her favourable assessment of our revised manuscript and for recommending our work for publication in JCB.

Reviewer #2 (Comments to the Authors (Required)):

The resubmission has addressed my main criticisms of the first version of the manuscript. I think the updated work on KKT14, with finer control on the number of phosphosites targeted for mutation, has improved and focused the results. It is a little unfortunate that the in vivo phosphomutants of KKT4 do not appear to have any phenotypes, but that is still useful information. The increased lookups for the nanobody fusion make the localization of these constructs much easier to interpret.

We thank the reviewer again for his/her comments and for acknowledging the additional work performed within the frame of this revision.

Is the "High dox" condition truly 10 ng/mL, compared to "low dox" at 7.5 ng/mL. I'm surprised that such a small difference leads to a very big change in the localization of the nanobody fusion, considering that full-bore induction is usually considered to be around 100 ng/mL.

It is indeed the case that the induction of this construct behaves very sensitively to small changes in doxycycline concentration and is easily 'overexpressed'. We were surprised ourselves, and hence had to carefully evaluate different concentrations of doxycycline before arriving at 7.5 ng/mL as an ideal condition.

Per the authors' response to my point about the artificial enhancement of their EM, I was referring to the brown lines used to enhance the MTs in the figure 2 insets. I would like to see those images without the lines, so that we can draw our own conclusions about the positioning and calls on the MTs.

We apologize for having mis-understood this point in the first round of revision. We now present the figure 2 insets without the brown lines to highlight MTs in supplementary figure S2, C and D.